# LLM as an Algorithmist: Enhancing Anomaly Detectors via Programmatic Synthesis

**Hangting Ye**[1,2], **Jinmeng Li**[1], **He Zhao**[3,4], **Mingchen Zhuge**[5], **Dandan Guo**[1,5*],
**Yi Chang**[1,6,7*], **Hongyuan Zha**[2]

School of Artificial Intelligence, Jilin University[1]; CUHK-Shenzhen[2]
CSIRO's Data61[3]; Department of DSAI, Monash University[4]; KAUST[5]
International Center of Future Science, Jilin University[6]
Engineering Research Center of Knowledge-Driven Human-Machine Intelligence, MOE, China[7]
`{yeht22,lijm9921}@mails.jlu.edu.cn,`
`he.zhao@data61.csiro.au, mingchen.zhuge@kaust.edu.sa,`
`{guodandan,yichang}@jlu.edu.cn, zhahy@cuhk.edu.cn`

## Abstract

Existing anomaly detection (AD) methods for tabular data usually rely on some assumptions about anomaly patterns, leading to inconsistent performance in real-world scenarios. While Large Language Models (LLMs) show remarkable reasoning capabilities, their direct application to tabular AD is impeded by fundamental challenges, including difficulties in processing heterogeneous data and significant privacy risks. To address these limitations, we propose LLM-DAS, a novel framework that repositions the LLM from a "data processor" to an "algorithmist". Instead of being exposed to raw data, our framework leverages the LLM's ability to reason about algorithms. It analyzes a high-level description of a given detector to understand its intrinsic weaknesses and then generates detector-specific, data-agnostic Python code to synthesize "hard-to-detect" anomalies that exploit these vulnerabilities. This generated synthesis program, which is reusable across diverse datasets, is then instantiated to augment training data, systematically enhancing the detector's robustness by transforming the problem into a more discriminative two-class classification task. Extensive experiments on 36 TAD benchmarks show that LLM-DAS consistently boosts the performance of mainstream detectors. By bridging LLM reasoning with classic AD algorithms via programmatic synthesis, LLM-DAS offers a scalable, effective, and privacy-preserving approach to patching the logical blind spots of existing detectors. The source code is available at `https://github.com/HangtingYe/LLM_DAS#`.

## 1 Introduction

Anomaly detection (AD) is a fundamental machine learning task that identifies instances significantly deviating from normal data. Tabular data, typically represented as heterogeneous feature vectors (e.g. numerical and categorical features) (Gorishniy et al., 2021; Borisov et al., 2022a; Ye et al., 2024), is a crucial modality for AD (Han et al., 2022; Yin et al., 2024), with applications spanning cyber-security (Ahmad et al., 2021), rare disease diagnosis (Fernando et al., 2021; Ye et al., 2023a), and financial fraud detection (Al-Hashedi & Magalingam, 2021). However, labeled anomalies are scarce due to the costly and time-consuming annotation by domain experts (Chandola et al., 2009; Ye et al., 2023b; Guo et al., 2023). In tabular anomaly detection, many recent works adopt a one-class classification paradigm, where only normal samples are available for training (Schölkopf et al., 1999; Sohn et al., 2021; Yin et al., 2024; Ye et al., 2025). These TAD methods usually model how anomalies differ from normal data (Ahmed et al., 2016) by making specific assumptions, e.g., reconstruction-based approaches like Principal Component Analysis (PCA) (Shyu et al., 2003) assume anomalies are harder to reconstruct. Despite the success of these methods, it has been shown that assumptions made by a method may not hold when facing to heterogeneous tabular data in real-world scenarios (Wolpert & Macready, 1997; Han et al., 2022).

---

*Corresponding authors.

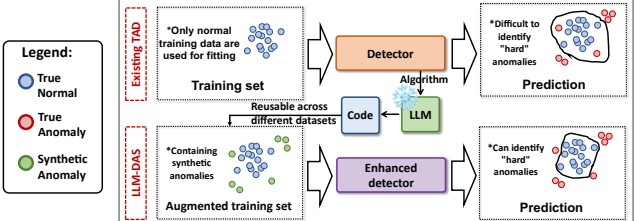

Figure 1: Comparison between traditional TAD methods and our LLM-DAS. By synthesizing "hard" anomalies, LLM-DAS effectively transforms the original one-class classification problem into a more discriminative two-class classification problem, thereby strengthening the detector and yielding a more nuanced decision boundary.

Rather than designing yet another detector with its own new set of fragile assumptions, we ask a more fundamental question: can we systematically enhance existing detectors to be more robust against the violation of their own core logic? Our core idea is to strengthen detectors by exposing them to their "blind spots" through synthesized "hard" anomalies. Here, "hard" anomalies are synthetic anomalous samples that the detector struggles to tell apart from normal data., i.e., cases where the detector is most likely to fail. As illustrated in Fig. 1, this targeted augmentation effectively transforms the learning problem from a limited one-class setting into a more powerful and discriminative two-class setting, allowing a model to learn a more nuanced and robust decision boundary. The pivotal challenge, however, lies in generating anomalies of sufficient quality to be truly "hard". This requires targeting the detector's core algorithmic logic, demanding a paradigm shift from data-level optimization to a logic-level strategic attack. This is a task for which the unique algorithmic reasoning capabilities of Large Language Models (LLMs) are exceptionally well-suited. With their remarkable capabilities in language understanding (Brown et al., 2020), complex planning (Qin et al., 2023), and code generation (Zheng et al., 2024), LLMs possess an unparalleled ability to reason about abstract algorithmic mechanisms.

In this paper, we introduce LLM-DAS (LLM-Guided Detector-Aware Anomaly Synthesis), a framework that positions the LLM as an "algorithm strategist" rather than a "data processor". In the first stage, LLM-DAS utilizes an LLM's reasoning capabilities to analyze the underlying mechanism of a given detector and generate detector-specific, data-agnostic Python code for anomaly synthesis. As no data is input into the LLM, this process elegantly sidesteps challenges related to tabular data processing and privacy concerns. In the second stage, the generated code is instantiated on a specific dataset to synthesize "hard-to-detect" anomalies. These anomalies are then used to augment the training data, enhancing the detector's robustness against violated assumptions. Crucially, the detector-specific synthesis code is reusable across different TAD datasets, enabling both scalability and high effectiveness of LLM utilization. In brief, our contributions are summarized as follows: **1)** We identify the fragile assumptions of existing TAD methods as a core limitation and propose a novel direction of using LLM-generated code for targeted anomaly synthesis to enhance detector robustness. This approach simultaneously addresses inherent issues related to data privacy and processing for LLMs in the tabular domain. **2)** We propose the LLM-DAS framework, which leverages an LLM to reason about detectors and generate reusable, data-agnostic synthesis code. This code is then instantiated to create synthetic hard anomalies that enhance detection performance. **3)** We conduct extensive experiments on numerous benchmarks, where results demonstrate that LLM-DAS consistently and significantly boosts the performance of mainstream TAD methods.

## 2 RELATED WORK

**Tabular Anomaly Detection.** One of the main settings for tabular anomaly detection is one-class classification, where a model learns from only normal samples (Sohn et al., 2021; Yin et al., 2024). The success of existing methods, however, relies on specific and often fragile assumptions. *(1) Reconstruction-based methods* like PCA (Shyu et al., 2003) and autoencoders (Principi et al., 2017; Kim et al., 2019) assume anomalies yield high reconstruction error (Shyu et al., 2003; Kim et al., 2019; Ye et al., 2025). This premise fails for anomalies that lie deceptively close to the learned manifold of normal data. *(2) Density estimation methods* posit that anomalies occupy low-density regions (Parzen, 1962; Li et al., 2022). This makes them vulnerable to local anomalies hidden within otherwise dense data clusters. *(3) Classification-based approaches* such as OCSVM learn a boundary around normal data (Schölkopf et al., 1999; Tax & Duin, 2004), an assumption that is easily

violated by complex, non-convex data distributions. *(4) Isolation-based methods* like IForest assume anomalies are easily isolated (Liu et al., 2008; Cao et al., 2025). Their effectiveness diminishes when anomalies form small clusters or are adjacent to normal ones. In these cases, the isolation paths of anomalies are no longer much shorter than those of normal data, compromising the core principle of isolation. The inherent brittleness of these varied assumptions motivates our work to directly enhance detectors against the violation of their own core logic. For a comprehensive overview, we refer readers to several surveys (Pang et al., 2021; Ruff et al., 2021; Chandola et al., 2009).

**Large Language Models for Tabular Data Learning.** Recent advances in LLMs have sparked growing interest in their application to tabular data tasks, including prediction (Dinh et al., 2022; Hegselmann et al., 2023), synthesis (Borisov et al., 2022b; Zhang et al., 2023), and feature engineering (Han et al., 2024; Nam et al., 2024). A comprehensive overview can be found in the survey by Fang et al. (2024). However, these approaches mostly treat the LLM as a *data processor*, requiring direct access to raw feature values and treating each row as natural language descriptions. This paradigm faces fundamental challenges: LLMs inherently struggle with heterogeneous numerical features (Fang et al., 2024; Yan et al., 2024), and direct data exposure creates significant privacy risks. Even recent attempts (Tsai et al., 2025) to apply LLMs for tabular anomaly detection—by fine-tuning them on normal samples—are still bound by these data-access and cost limitations, hindering their practicality. In this work, we depart from this paradigm by proposing a new role for the LLM: an *algorithmist*. Instead of processing data, the LLM reasons about the high-level, data-agnostic logic of a given anomaly detector. Its task is to understand the detector's intrinsic weaknesses and programmatically generate a synthesis strategy in the form of executable code. This "logic-level" approach inherently preserves data privacy, leverages the LLM's core strengths in reasoning and code generation, and produces a reusable synthesis logic that is both detector-aware and data-agnostic.

# 3 METHOD

## 3.1 PROBLEM FORMULATION AND MOTIVATION

Following previous works (Yin et al., 2024; Ye et al., 2025), we formulate tabular anomaly detection (TAD) in a one-class classification setting, where training set $D_{\text{train}}$ only contains normal samples with $\mathbf{x} \in \mathcal{X}$ and test set $D_{\text{test}}$ contains both normal and anomalous samples. The goal of TAD is to learn a score function $f : \mathcal{X} \to \mathbb{R}$ from $D_{\text{train}}$, which assigns an anomaly score $f(\mathbf{x})$ to a given test sample, with a higher score indicating a higher likelihood of being an anomaly. A variety of methods have been developed for this task, such as reconstruction-based (e.g., PCA (Shyu et al., 2003)), isolation-based (e.g., IForest (Liu et al., 2008)), and density-based (e.g., ECOD (Li et al., 2022)) approaches. To make our exposition precise, we denote $T = \{\text{"IForest"}, \text{"PCA"}, ...\}$ as the set of algorithm names, where $t \in T$ is a symbolic label specifying the chosen algorithm type. Each $t$ corresponds to an abstract, parameter-free algorithmic description, $A_t$. Applying the algorithm $A_t$ to the training set $D_{\text{train}}$ yields a concrete score function $f_t$ with learned parameters.

The efficacy of these classical TAD algorithms, however, hinges on their underlying assumptions about the nature of anomalies. For instance, IForest (Liu et al., 2008) assumes anomalies are more easily isolated, while PCA (Shyu et al., 2003) assumes they have large reconstruction errors from a learned low-dimensional subspace. Details of these detectors are in Appendix 6.4. These assumptions are often brittle and fail when confronted with heterogeneous, real-world data, leading to degraded performance (Wolpert & Macready, 1997; Han et al., 2022). It raises a critical question: how can we systematically enhance existing detectors to be more robust against the violation of their own core logic?

Our core approach is to strengthen detectors by exposing them to their own blind spots through the synthesis of "hard" anomalies—samples specifically crafted to challenge the algorithmic assumptions of a given detector. By augmenting the training data with these targeted negative examples, we transform the learning problem from a limited one-class setting into a more discriminative two-class setting. The success of this approach, however, hinges on generating anomalies of sufficient quality to be truly "hard". Naive, data-level methods, such as applying small perturbations to input features, often fail to target the core algorithmic logic of a detector. To overcome this, we introduce a paradigm shift from data-level optimization to a logic-level strategic attack, leveraging the unique ability of Large Language Models (LLMs) to reason about abstract algorithmic mechanisms. The LLM's role is not to process data, but to analyze a detector's high-level logic, infer its weaknesses, and then

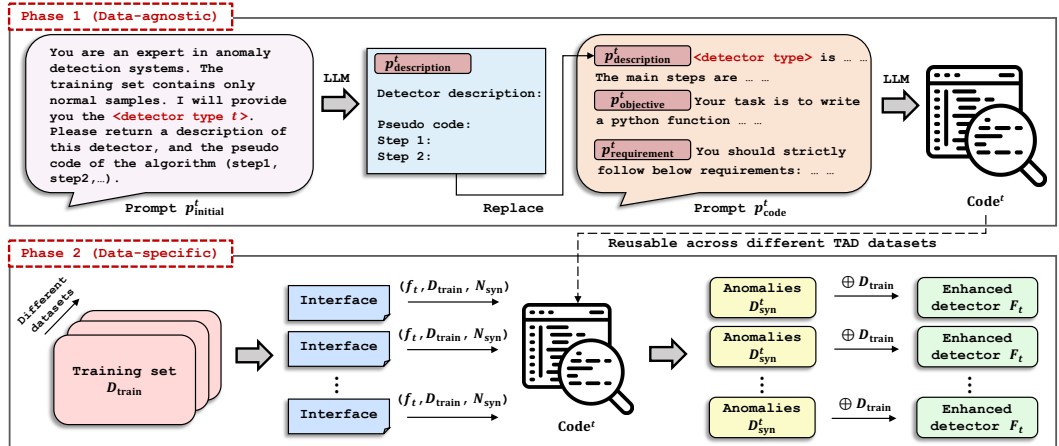

Figure 2: The LLM-DAS framework consists of two phases: (1) a data-agnostic reasoning phase, where an LLM generates a reusable anomaly synthesis code for one type of detector, and (2) a data-specific phase, where this code is applied to generate challenging anomalies for detector enhancement.

generate data-agnostic Python code that programmatically synthesizes anomalies tailored to exploit these specific vulnerabilities.

## 3.2 PROPOSED METHOD: LLM-DAS

This work proposes LLM-DAS (LLM-Guided Detector-Aware Anomaly Synthesis), a two-stage framework. The core of LLM-DAS is to first leverage an LLM's reasoning to generate a data-agnostic synthesis code tailored for a detector algorithm $A_t$ (Stage 1), and then execute this code to create the "hard" anomalies needed to enhance $f_t$ on a specific dataset (Stage 2). Notably, the detector-specific synthesis code is reusable across different TAD datasets. Fig. 2 illustrates the overall pipeline.

### 3.2.1 DETECTOR-AWARE CODE GENERATION VIA LLM

This stage aims to translate the LLM's abstract algorithmic knowledge regarding a detector type $t$ (represented by its algorithm $A_t$) into a concrete, executable synthesis code. As the LLM analyzes the detector not the data, this process is entirely data-agnostic to preserve privacy and ensure the reusability of the generated code. Specifically, given a detector type $t$, we construct a prompt $p_{\text{code}}^t$ that guides the LLM to capture the underlying assumptions and mechanisms of algorithm $A_t$, and then produce anomalies synthesis code that captures the kinds of anomalies that are inherently difficult for $A_t$ to detect. To fully exploit the reasoning capability of LLMs, the prompt $p_{\text{code}}^t$ is defined as:

$$p_{\text{code}}^t = p_{\text{description}}^t + p_{\text{objective}}^t + p_{\text{requirements}}^t. \tag{1}$$

Below we introduce the three key components (see Appendix 6.5 for details).

(i) Detector description $p_{\text{description}}^t$, a summary and pseudo-code of the detector algorithm $A_t$. This information is generated by the LLM itself:

$$p_{\text{description}}^t = \text{LLM}(p_{\text{initial}}^t), \tag{2}$$

where prompt $p_{\text{initial}}^t$ is the input for LLM with details provided in Appendix 6.5. This step provides a structured context to prime the LLM for generating detector-aware synthesis code.

(ii) Objective specification $p_{\text{objective}}^t$, which instructs the LLM to generate code that synthesizes "hard" anomalies. A core tenet of our framework is the use of symbolic interfaces, which are defined in this prompt section. For example, we state: "... *After the Python function is completed, users can provide the function with: A trained PCA model (`model`) that exposes `predict_score()`, The training samples (`X_train`).*" This interaction paradigm is crucial: by defining interfaces as symbolic placeholders rather than concrete values, the LLM can write a general-purpose program without ever accessing the real data. It effectively provides the LLM with a standard API to program against, positioning it as an "algorithm strategist" rather than a "data processor." This enables the

generated code to be both powerful–accessing training set statistics and querying the detector's scoring behavior–and universally applicable across different datasets.

**(iii) Generation requirements** $p^t_{\text{requirements}}$, which impose constraints on the code format and functionality. Notably, we encourage the code to identify "borderline" normal training samples lying near the decision boundary of the detector, and transform them into anomalies leveraging LLM reasoning. This approach is effective for producing informative anomalous samples. Nevertheless, if the LLM discovers a superior strategy, it is allowed to adopt it.

The LLM takes the prompt $p^t_{\text{code}}$ as input and leverages its prior knowledge to understand the mechanism of algorithm $A_t$ (summarized as $p^t_{\text{description}}$ in Eq. 2). Based on this understanding, it reasons about which types of anomalies are most challenging for $A_t$ to detect while being most informative for improving detection performance. The output strictly follows a Python code format:

$$\text{Code}^t = \text{LLM}(p^t_{\text{code}}),$$
$$\{\mathcal{S}^t_{\text{policy}}, \mathcal{S}^t_{\text{program}}, \mathcal{S}^t_{\text{explanation}}\} \subseteq \text{Code}^t, \tag{3}$$

where the code consists of three structured components: the policy $\mathcal{S}^t_{\text{policy}}$ that specifies how anomalies should be synthesized in detail, the executable program $\mathcal{S}^t_{\text{program}}$ that implements the policy $\mathcal{S}^t_{\text{policy}}$ in Python, and the explanation $\mathcal{S}^t_{\text{explanation}}$ with clear comments explaining key steps in program $\mathcal{S}^t_{\text{program}}$. This structured output, comprising not just a program but also its underlying policy and explanation, significantly enhances the interpretability and trustworthiness of our approach, allowing users to verify the logic behind the LLM's synthesis strategy. We provide an example template of the generated code in List. 3.2.1, with detailed generated code provided in Appendix 6.6.

**The Generated Python Code Template**

```python
def generate_hard_anomalies(n_samples: int, model
    , X_train: np.ndarray) -> np.ndarray:
    """
    # === Policy placeholder ===
    # The anomaly synthesis logic is applied here
    # (details omitted for brevity)
    """
    # === Program and explanation placeholder ===
    # (details omitted for brevity)
    # Explanation for step 1
    Step 1 xxx
    ...
    return np.array(generated_anomalies)
```

Crucially, this LLM-driven code generation is data-agnostic. The LLM reasons about a detector's algorithmic weaknesses based solely on its description—never observing raw data or model parameters, thus preserving privacy. This process yields a reusable, detector-specific synthesis code without requiring any LLM fine-tuning. The resulting structured code is instantiated with dataset-specific values in the next stage to synthesize anomalies.

### 3.2.2 DATASET-WISE HARD ANOMALY INSTANTIATION

Once the detector-aware code is generated, the next step is to instantiate it on a specific dataset. To do so, the generated code, denoted as $\text{Code}^t$, requires a well-defined set of input interfaces to execute. We define this complete set of interfaces as $\mathcal{I}$, which comprises three core elements presented in their logical order: (1) Training set $D_{\text{train}}$, which serves as the foundation for the instantiation process. The code requires this data not only to train the score function in the next step but also to access its distribution and statistical properties (such as the decision boundary) during the synthesis phase to generate more challenging anomalies. (2) A fitted score function $f_t$: Using the training set $D_{\text{train}}$ described above, we obtain a fitted score function $f_t$ by applying algorithm $A_t$ to $D_{\text{train}}$. This score function provides the key standardized interface that maps any given input sample to an anomaly score. (3) The number of anomalies to synthesize $N_{\text{syn}}$, a hyperparameter specifying the desired number of anomaly samples to be generated. Collectively, these three components constitute the complete set of information required for the code's execution. The complete interface set $\mathcal{I}$ is therefore defined as their union:

$$\mathcal{I}_t = D_{\text{train}} \cup f_t \cup N_{\text{syn}}. \tag{4}$$

With this set of interfaces $\mathcal{I}$ available for a specific dataset, we can execute the LLM-generated code. The code operates as a function that accepts $\mathcal{I}$ as input and returns a set of synthesized anomalies,

$$\text{Code}^t : \mathcal{I}_t \mapsto D^t_{\text{syn}}, D^t_{\text{syn}} = \{\mathbf{x}^t_{\text{syn}}\}. \tag{5}$$

This formulation makes explicit that the code consumes both the score function and the training set at instantiation time, while the LLM that produced the code itself remained data-agnostic during

generation. The resulting $\text{Code}^t$, generated in a single forward pass for a specific detector type $t$, can be applied universally across diverse datasets.

**Case Study**. To make this instantiation process concrete, we now present a case study for the IForest detector (Liu et al., 2008). The LLM-generated code, $\text{Code}^{\text{IForest}}$, operates based on a specific synthesis policy designed to exploit the core mechanics of IForest. This policy $\mathcal{S}_{\text{policy}}^{\text{IForest}}$, which the LLM articulates as part of its output (Eq. 3), is detailed below in Policy 3.2.2.

---

**Policy $\mathcal{S}_{\text{policy}}^t$ contained in $\text{Code}^t$ (Eq. 3) for Generating Hard Anomalies for IForest**

The **core weakness** of IForest is that it relies on axis-aligned splits to isolate points. A sample is considered anomalous if it can be isolated in a few splits (short path length). Conversely, a sample is normal if it is "deep" within the data cloud, requiring many splits to be isolated (long path length). Our strategy exploits this by creating new points that are conceptually anomalous but are geometrically located in a way that maximizes their path length, thereby fooling the IForest. The policy consists of two main steps:
**1. Identify 'Borderline' Normal Samples:** These are the normal training samples that lie on the fringe or edge of the dense data cloud. The IForest model is already less certain about these points, assigning them the highest anomaly scores among all normal samples. They serve as perfect "seeds" for our hard anomalies because they are already close to the decision boundary. We identify these by finding the training samples in the top percentile of anomaly scores.
**2. Transform Seeds into Hard Anomalies via 'Controlled Extrapolation':** We transform these borderline seeds into anomalies. A naive transformation (e.g., adding large random noise) would create an obvious outlier with a short path length, which is an *easy* anomaly. To create a *hard* anomaly, we must move the seed point in a way that doesn't significantly shorten its path length.

---

In practice, the implementation of this policy relies on the interfaces provided to the code. Specifically, to identify "Borderline" Normal Samples (Step 1), the code uses the fitted $f_t$ to compute anomaly scores for all samples in the training set. It then selects a subset of these samples as "seeds" from the top percentile of scores. For Step 2, the code applies targeted modifications to these seed points. The objective is to push them into a sparser region of the feature space, making them factually anomalous, while algorithmically ensuring the transformation does not significantly reduce their path length as computed by the IForest score function $f_t$. This process effectively creates anomalies that are "camouflaged" to appear normal to IForest, thus representing a significant challenge for the detector.

**Detector Enhancement with Hard Anomalies.** Finally, the synthesized anomalies are integrated with the original training set:

$$D_{\text{aug}}^t = D_{\text{train}} \cup D_{\text{syn}}^t, \tag{6}$$

and a binary classifier $\tilde{f}_t$ (e.g., a decision tree) is trained on $D_{\text{aug}}^t$ to distinguish between normal and anomalous samples. By explicitly framing the task as a supervised classification problem and exposing the classifier to these challenging synthetic anomalies, the learning process encourages the $\tilde{f}_t$ to capture more discriminative patterns between the two classes, thereby improving its ability to generalize and detect real-world anomalies. We fuse the original ($f_t$) and enhancement ($\tilde{f}_t$) score functions by summing their min-max normalized scores with the following final score:

$$F_t(x) = \text{Norm}_{\text{min-max}}(f_t(x)) + \text{Norm}_{\text{min-max}}(\tilde{f}_t(x)). \tag{7}$$

This approach preserves the original detector's biases while adding the nuanced patterns learned from synthetic data. A key aspect of our method is its adherence to the one-class classification framework since training set $D_{\text{train}}$ only have ground-truth normal data. We leverage LLM-generated code to synthesize anomalies tailored to the detector and the training dataset. This approach allows the detector to adapt to edge cases that it would otherwise fail to recognize, effectively broadening its coverage of the abnormal space while preserving its ability to model the normal distribution (*mitigating the challenge of fragile assumptions in TAD*). Overall, this straightforward yet principled strategy ensures that the benefits of LLM-guided anomaly synthesis are fully incorporated into the detector, enhancing both its discriminative power and generalization ability.

### 3.2.3 SUMMARY AND DISCUSSION

The LLM-DAS framework operates in a two-phase process that decouples data-agnostic reasoning from data-specific synthesis. In Phase 1, an LLM analyzes a high-level description of a detector and generates a universal, reusable Python code that encapsulates a strategy for synthesizing hard anomalies. Crucially, this phase is data-agnostic, preserving privacy. In Phase 2, this code is instantiated on a specific training set $D_{\text{train}}$ and its fitted detector $f_t$ to generate challenging anomalies. These anomalies are then used to train an enhancement classifier $\tilde{f}_t$, which is fused with $f_t$ to form the final detector $F_t$. Please the detailed Algorithm 1 in Appendix 6.3. The core rationale of this framework is to improve detector robustness. Traditional one-class detectors rely on fragile

Table 1: The statistical information of detection performance improvement achieved by LLM-DAS ($F_t$) over trained source detectors ($f_t$) on 36 datasets. "Baseline" means the average performance achieved by the corresponding source detector ($f_t$) on all datasets. "Improv. ($\Delta$)" indicates the average absolute performance improvement by the LLM-DAS ($F_t$) over $f_t$, "Improv. (%)" indicates the average relative performance improvement in percentage. "Win Count" represents the number of datasets that LLM-DAS ($F_t$) made improvements over $f_t$. $p$-value measures the improvement significance across the 36 datasets ($< 5\%$ is significant).

| Detector | AUC-PR | | | | | AUC-ROC | | | | |
|---|---|---|---|---|---|---|---|---|---|---|
| | Baseline | Improv. ($\Delta$) | Improv. (%) | Win Count | $p$-value | Baseline | Improv. ($\Delta$) | Improv. (%) | Win Count | $p$-value |
| PCA | .5975 | .0402 | 21.50 | 30 / 36 | .0271 | .7349 | .0325 | 7.64 | 27 / 36 | .0155 |
| IForest | .5724 | .0617 | 23.60 | 26 / 36 | .0010 | .7555 | .0385 | 10.90 | 28 / 36 | .0034 |
| OCSVM | .5295 | .0723 | 84.49 | 25 / 36 | .0239 | .7141 | .0437 | 10.02 | 25 / 36 | .0170 |
| ECOD | .5376 | .0512 | 23.05 | 28 / 36 | .0014 | .7056 | .0398 | 8.20 | 26 / 36 | .0183 |
| DRL | .7437 | .0412 | 15.81 | 25 / 36 | .0238 | .8509 | .0272 | 3.79 | 21 / 36 | .0035 |

assumptions that are often violated by real-world anomalies. By leveraging an LLM's unique capability for algorithmic reasoning, our method programmatically generates samples that specifically target these assumptions, exposing the detector to edge cases it would otherwise miss. This targeted augmentation leads to a more nuanced decision boundary and better generalization(a theoretical intuition is in Appendix 6.14). Overall, our method offers three key advantages: (i) **Data privacy**, as the LLM never observes the data; (ii) **Model-specificity**, leading to highly effective synthesis; and (iii) **Generalizability**, as the framework is a plug-and-play module for various detectors.

## 4 EXPERIMENTS

### 4.1 EXPERIMENTAL SETUP

**Datasets.** We adopt an extensive benchmark with 36 datasets selected from Outlier Detection DataSets (ODDS) (Rayana, 2016) and Anomaly Detection Benchmark (ADBench) (Han et al., 2022) following previous works (Yin et al., 2024; Ye et al., 2025). These datasets span diverse domains, including healthcare, science, and social sciences. The dataset properties are summarized in Appendix 6.2. Per the literature (Zong et al., 2018; Bergman & Hoshen, 2020; Yin et al., 2024; Thimonier et al., 2024; Ye et al., 2025), we construct the training set by subsampling 50% of the normal samples. The remaining 50% of the normal samples are then combined with the entire set of anomalies to form the test set. We employ Area Under the Precision-Recall Curve (AUC-PR) and Area Under the Receiver Operating Characteristic Curve (AUC-ROC) as our evaluation criteria.

**Implementation and Baselines Details.** For the code generation phase, we utilized the Gemini-2.5-Pro API. This represents a modest, two-time cost per detector type (Eq. 2 and Eq. 3), as the resulting synthesis $\text{Code}^t$ is reusable across any dataset. The subsequent anomaly instantiation and binary $\tilde{f}_t$ classifier training phases were performed locally and incurred no further monetary cost. The number of synthesized samples is set to 10% of the training set size by default. Following recent studies (Yin et al., 2024; Thimonier et al., 2024; Ye et al., 2025), we implement the hyperparameters of all detector models according to their original papers. In our experiments, we use PCA (Shyu et al., 2003), IForest (Liu et al., 2008), OCSVM (Schölkopf et al., 1999), ECOD (Li et al., 2022), and DRL (Ye et al., 2025) as five source detectors (based on different assumptions) to evaluate the effectiveness of LLM-DAS. In addition to these detector-based integrations, we also conduct direct comparisons against conventional and advanced TAD baselines to further evaluate benchmark performance. Details of these detectors are provided in Appendix 6.4.

### 4.2 MAIN RESULTS

**Improvement over Source Detectors.** We first evaluate the effectiveness of LLM-DAS by measuring its improvements over the original source detectors across 36 benchmark datasets. As summarized in Table 1, overall, LLM-DAS consistently enhances all source detectors across both AUC-PR and AUC-ROC metrics. For instance, when enhancing OCSVM, LLM-DAS improves the AUC-PR by an average of 0.0723, corresponding to an 84.49% relative improvement, and achieves gains on 25 out of 36 datasets. Similar trends can be observed for PCA, IForest, ECOD, and DRL, confirming that the ours is broadly effective across diverse detection algorithms. To assess whether the observed

improvements are statistically significant, we conduct a paired one-tailed t-test across the 36 datasets. The results show that most improvements achieve $p$-values below 0.05, indicating that the performance gains are statistically significant rather than random fluctuations. These results demonstrate that LLM-DAS can reliably enhance existing detectors by leveraging synthesized anomalies. Complete results can be found in Table 4 and Table 5 located in the Appendix 6.7.

**Comparison with Other Baseline Methods.** We further compare the performance of LLM-DAS with conventional and advanced TAD baselines. Since LLM-DAS and the recent finetune-based AnoLLM (Tsai et al., 2025) have some shared datasets,

we cite the results of AnoLLM due to its high computational cost, and perform other methods on the shared subsets. As shown in Fig. 3, LLM-DAS, when coupled with DRL, achieves the highest average performance across these datasets. Notably, although both AnoLLM and ours leverage LLMs, the ways they are used differ fundamentally: AnoLLM fine-tunes a small LLM with open-weights on each target dataset, learning a specific model for each dataset. In contrast, we view the LLM with the reasoning ability as an algorithmic analyst: it examines the logic of detection methods and generates corresponding code, making our approach dataset-agnostic and independent of further LLM fine-tuning. Even with a significantly lower computational cost than AnoLLM, our method still outperforms it. Full results are provided in Table 6 and Table 7 of Appendix 6.8.

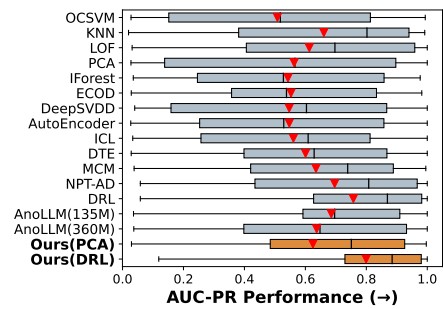

Figure 3: Comparison of all models' performance across different datasets (in AUC-PR). The red triangles represent the average value. AnoLLM has two versions (135M and 360M parameters).

## 4.3 FURTHER ANALYSIS

**Comparison with Other Anomaly Synthesis Methods.** We compare LLM-DAS with several representative anomaly synthesis strategies, including Gaussian noise injection, random outliers, and SMOTE-based (Chawla et al., 2002) sample generation. Fig. 4 (a) reports the average AUC-PR across datasets (detailed results and base synthesis strategy details in Appendix 6.13.1). We can find that, naive methods usually yield inconsistent results. While methods like random outlier generation may offer marginal gains for simple detectors like IForest, they are not universally effective and can be detrimental. For instance, random synthesis severely degrades the performance of sophisticated models like DRL, highlighting the risk of a generic policy conflicting with a detector's intrinsic logic. In contrast, LLM-DAS delivers consistent and substantial improvements across all models. Its detector-aware strategy generates hard anomalies tailored to each model's specific vulnerabilities, ensuring the augmented data is always informative and leads to a more robust decision boundary.

**Ablation Studies on LLM-DAS Design.** Our ablation studies (Fig. 4 (b)) validate the critical role of each core component in LLM-DAS by changing our designed prompt. First, to test our core detector-aware premise, a Generic variant was created by removing the detector's principles from the prompt. Second, to confirm the necessity of generating hard anomalies, a Simple variant was prompted without access to the model.predict_score() function, preventing the LLM from assessing anomaly difficulty. Finally, to evaluate our borderline heuristic, a Random variant was instructed to transform randomly selected normal samples rather than borderline normal samples into anomalies. The inferior performance of each variant confirms that detector-awareness, the capability to generate

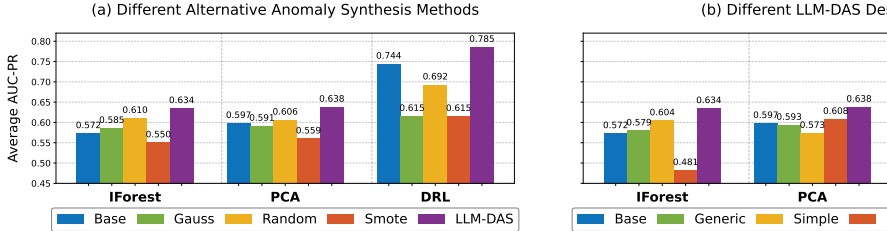

Figure 4: Performance comparison of anomaly synthesis methods (left) and LLM-DAS designs across different datasets (right) in terms of AUC-PR.

Table 2: Performance comparison in AUC-PR when enhancing OCSVM with the anomaly generation code designed for IForest, PCA, OCSVM, respectively. The average relative improvement is provided (↑ %).

| Baseline | Synthesis code | Vertebral | Glass | Imgseg | Shuttle | Speech | Cardio | Vowels | Average |
|---|---|---|---|---|---|---|---|---|---|
| OCSVM | IForest | +0.1025 | +0.0615 | +0.0546 | +0.0414 | -0.0015 | -0.1148 | -0.0905 | +0.0076(↑ 1.68%) |
| | PCA | -0.0037 | +0.1108 | +0.0378 | +0.0457 | -0.0025 | -0.1476 | -0.1899 | -0.0213(↓ 4.72%) |
| | OCSVM | +0.1216 | +0.111 | +0.0709 | +0.0418 | +0.0015 | -0.0531 | -0.0871 | +0.0295(↑ 6.53%) |

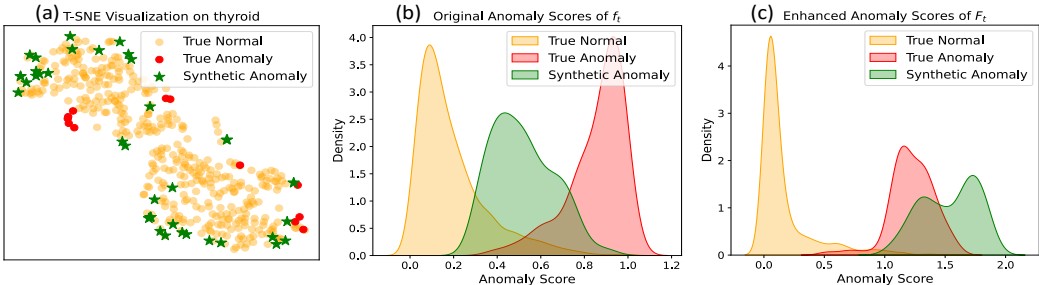

Figure 5: Visualization of synthetic hard anomalies and score distributions on Thyroid test dataset. (a) T-SNE plots of normal, real anomaly, and LLM-DAS–generated anomaly samples. (b) Kernel density estimation (KDE) of anomaly scores from the source detector $f_t$. (c) KDE of anomaly scores from the enhanced detector $F_t$.

challenging anomalies, and a focus on borderline samples are all critical to the success of LLM-DAS. Please find details in Appendix 6.13.2.

**Cross-Detector Synthesis Results.** To validate the critical role of our detector-aware design, we conducted a cross-detector experiment to test if a synthesis strategy for one detector could enhance another. We set the base detector $f_t$ as OCSVM and applied synthesis codes generated specifically for IForest, PCA, and OCSVM itself. The results in Table 2, which report the absolute AUC-PR change, are revealing. The "mismatched" scenarios (e.g., using PCA-specific code to enhance OCSVM) yielded inconsistent outcomes. While gains were observed on some datasets, this approach introduced a significant risk of performance degradation, evidenced by a substantial drop of 0.1899 on the Vowels dataset. In contrast, the "matched" synthesis (using OCSVM-specific code for the OCSVM detector) provided the most stable and significant average improvement. This experiment strongly demonstrates that the success of LLM-DAS stems not from a generic augmentation effect, but from the LLM's ability to programmatically exploit the unique mechanics of the target detector. It confirms that detector-awareness is the critical factor for achieving robust performance enhancement.

**Visualization and KDE Analysis of Synthetic Anomalies.** To further examine the quality of anomalies synthesized by LLM-DAS, we utilize T-SNE and Kernel Density Estimation (KDE) plots of anomaly scores on the Thyroid. Fig. 5 (a) shows that the Synthetic Anomalies (green stars) are strategically placed near the boundary of the True Normal manifold, confirming they are hard anomalies that intentionally avoid the sparse, obvious outlier regions. This "hardness" is quantitatively proven by the KDE plots of the original score distribution in Fig. 5 (b). The synthetic anomaly distribution severely overlaps with both true normal and anomaly distribution. This indicates the base detector struggles to distinguish the synthetic samples from normal data. Crucially, after training with LLM-DAS, the enhanced detector's KDE in Fig. 5 (c) shows a significant rightward shift in the synthetic anomaly scores. This boundary refinement not only elevates the synthetic anomaly scores but also pushes the true anomaly score distribution further away from the normal score distribution. This confirms that exposure to LLM-DAS generated hard samples results in a significantly tighter and more robust decision boundary, improving overall generalization.

**Additional Analysis.** We incorporate Appendix 6.15 to demonstrate how LLM-DAS inherently targets detector-specific weaknesses across diverse anomaly types. We introduce a controllable hardness-level prompting interface and a comprehensive failure-mode analysis in Appendix 6.16.

## 5 CONCLUSION

This paper introduces LLM-DAS, a novel framework that reimagines the role of LLMs in tabular anomaly detection. Instead of treating LLMs as direct data processors, we have successfully repositioned them as "algorithmists" that reason about the intrinsic logic of detectors. This paradigm shift can address two challenges: the LLM's inherent difficulty with raw tabular data and the fragile assump-

tions of conventional detectors. Our core contribution is a fully programmatic, privacy-preserving approach where the LLM generates detector-specific, data-agnostic, and reusable synthesis code to produce "hard" anomalies, enhancing model robustness. Extensive experiments on 36 benchmarks demonstrate that LLM-DAS significantly enhances a wide array of detectors. Looking forward, the "LLM as an algorithmist" paradigm opens up exciting new avenues, such as enhancing algorithms in other domains or even automating algorithmic discovery. Our work paves the way for more adaptive and robust machine learning systems by integrating the algorithmic reasoning of LLMs.

**Ethics Statement**    We confirm that this research complies with all applicable ethical guidelines and does not present any ethical issues.

**Reproducibility Statement**    To ensure reproducibility, we provide source code through the link in the abstract. Complete details regarding datasets, experimental settings, and implementation are documented in Appendix 6.2, Appendix 6.3 and Appendix 6.4.

## ACKNOWLEDGMENTS

The authors would like to thank the anonymous referees for their valuable comments. In this work, Hangting Ye, Jinmeng Li, Dandan Guo and Yi Chang are supported by the National Key R&D Program of China under Grant (No. 2023YFF0905400), the National Natural Science Foundation of China (No. 623B2043, No. U2341229, No. 62306125), the Fundamental and Interdisciplinary Disciplines Breakthrough Plan of the Ministry of Education of China (No. JYB2025XDXM903), and the New Cornerstone Science Foundation through the XPLORER PRIZE. Hongyuan Zha's research is supported in part by Shenzhen Stability Science Program 2023, and National Natural Science Foundation of China (No. 72495131).

Part of this work was carried out during Dandan Guo's visit to King Abdullah University of Science and Technology (KAUST).

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

## 6 APPENDIX

## TABLE OF CONTENTS

## 6.1 USE OF LLMs

Large Language Models were used as assistive tools in the preparation of this manuscript. We employed LLMs for grammar checking, LaTeX formatting, and improving the clarity of technical descriptions. The core scientific contributions and conclusions presented in this paper originate from the authors' work.

## 6.2 DATASETS DETAILS

We adopt an extensive benchmark with 36 datasets selected from Outlier Detection DataSets (ODDS) (Rayana, 2016) and Anomaly Detection Benchmark (ADBench) (Han et al., 2022) following previous works (Yin et al., 2024; Ye et al., 2025). These datasets span diverse domains, including healthcare, science, and social sciences. The dataset properties are summarized in Table 3.

## 6.3 LLM-DAS DETAILS

For the code generation phase, we utilized the Gemini-2.5-Pro API. This represents a modest, two-time cost per detector type (Eq. 2 and Eq. 3), as the resulting synthesis $\text{Code}^t$ is reusable across any dataset. The subsequent anomaly instantiation and binary $\tilde{f}_t$ classifier training phases were performed locally and incurred no further monetary cost. The number of synthesized samples is set to 10% of the training set size by default. The detailed algorithm is as follows.

---

**Algorithm 1** LLM-DAS training and inference workflow.

---

1: **Input:** Normal-only training set $D_{\text{train}}$, test set $D_{\text{test}}$ (labels not accessed), LLM API, detector type;

2: Obtaining $p_{\text{description}}^t$ that summarizes the algorithm $A_t$ of $t$ and provides its pseudo code describing the concrete steps (Eq. 2);

3: Formulate a prompt $p_{\text{code}}^t$ describing the detector mechanism, synthesis objectives and requirements (Eq. 1);

4: Query LLM with prompt $p_{\text{code}}^t$ to generate a detector-aware anomaly synthesis code $\text{Code}^t$ (Eq. 3);

5: Train the initial score function $f_t$ on $D_{\text{train}}$;

6: Extract interfaces $\mathcal{I}_t = D_{\text{train}} \cup f_t \cup N_{\text{syn}}$ (Eq. 4);

7: Instantiate the LLM-generated code $\text{Code}^t$ using the interfaces $\mathcal{I}_t$ to synthesize hard anomalies $D_{\text{syn}}^t$ (Eq. 5);

8: Augment the training set: $D_{\text{aug}}^t = D_{\text{train}} \cup D_{\text{syn}}^t$ (Eq. 6);

9: Train a binary classifier on $D_{\text{aug}}^t$ to obtain the enhanced score function $\tilde{f}_t$;

10: Integrate the source function $f_t$ with $\tilde{f}_t$ to form the final function $F_t$ (Eq. 7);

11: **Output:** Final anomaly scores $\{F(\mathbf{x}) \mid \mathbf{x} \in D_{\text{test}}\}$.

---

## 6.4 BASELINE DETAILS

### 6.4.1 BASE DETECTORS COUPLED WITH LLM-DAS

Following recent studies (Yin et al., 2024; Thimonier et al., 2024; Ye et al., 2025), we implement the hyperparameters of all detector models according to their original papers. In our experiments, we

Table 3: Dataset properties. We use 36 commonly used tabular anomaly detection datasets in this paper.

|  | Samples | Dims | Anomalies |
|---|---|---|---|
| Abalone | 4177 | 7 | 2081 |
| Amazon | 10000 | 768 | 500 |
| Annthyroid | 7200 | 6 | 534 |
| Arrhythmia | 452 | 274 | 66 |
| Breastw | 683 | 9 | 239 |
| Campaign | 41188 | 62 | 4640 |
| Cardio | 1831 | 21 | 176 |
| Cardiotocography | 2114 | 21 | 466 |
| Comm.and.crime | 1994 | 101 | 993 |
| Fault | 1941 | 27 | 673 |
| Glass | 214 | 9 | 9 |
| Hepatitis | 80 | 19 | 13 |
| Imgseg | 2310 | 18 | 990 |
| Ionosphere | 351 | 33 | 126 |
| Lympho | 148 | 18 | 6 |
| Mammography | 11183 | 6 | 260 |
| Mnist | 7603 | 100 | 700 |
| Musk | 3062 | 166 | 97 |
| Optdigits | 5216 | 64 | 150 |
| Parkinson | 195 | 22 | 147 |
| Pendigits | 6870 | 16 | 156 |
| Pima | 768 | 8 | 268 |
| Satellite | 6435 | 36 | 2036 |
| Satimage-2 | 5803 | 36 | 71 |
| Shuttle | 49097 | 9 | 3511 |
| SpamBase | 4207 | 57 | 1679 |
| Speech | 3686 | 400 | 61 |
| Thyroid | 3772 | 6 | 93 |
| Vertebral | 240 | 6 | 30 |
| Vowels | 1456 | 12 | 50 |
| WDBC | 367 | 30 | 10 |
| WPBC | 198 | 33 | 47 |
| Wbc | 378 | 30 | 21 |
| Wilt | 4819 | 5 | 257 |
| Wine | 129 | 13 | 10 |
| Yeast | 1484 | 8 | 507 |

use PCA (Shyu et al., 2003), IForest (Liu et al., 2008), OCSVM (Schölkopf et al., 1999), ECOD (Li et al., 2022), and DRL (Ye et al., 2025) as five source detectors (based on different assumptions) to evaluate the effectiveness of LLM-DAS. The details of these detectors and LLM's opinion towards their weakness (included in the generated code $\text{Code}^t$) are as follows.

**PCA.** Principal Component Analysis (PCA) is a commonly used method for dimensionality reduction and anomaly detection. It identifies directions in the data (called principal components) along which the variance is maximized. In anomaly detection, PCA can be used to reconstruct each sample from the top principal components and compute the reconstruction error as the anomaly score. Samples with higher reconstruction error are considered more anomalous.

**IForest.** Isolation Forest (IForest) is an ensemble-based anomaly detection method that isolates anomalies instead of profiling normal data points. The key intuition is that anomalies are few and different, so they are easier to isolate using random partitions. IForest builds multiple random binary trees (isolation trees) by recursively splitting data based on randomly chosen features and split values. Data points that require fewer splits to be isolated are more likely to be anomalies, while normal

points typically require more splits. The average path length across trees is used to compute an anomaly score, where shorter paths indicate higher anomaly likelihood.

**OCSVM.** One-Class Support Vector Machine (OCSVM) is a kernel-based anomaly detection method that learns a decision boundary around normal data. It maps the input data into a high-dimensional feature space using a kernel function, then finds the smallest region (a hyperplane or hypersphere) that encloses most of the normal samples. The assumption is that normal data lies in a dense region, while anomalies fall outside this boundary. OCSVM is particularly effective when the training set contains only normal samples, as it explicitly models the support of the normal data distribution.

**ECOD.** ECOD (Empirical Cumulative Distribution-based Outlier Detection) is a lightweight, univariate anomaly detection method that leverages the empirical cumulative distribution (ECDF) of each feature independently. It computes how extreme a value is relative to the distribution of that feature by evaluating its position in the ECDF. The core idea is that anomalous values often lie in the tails of the distribution. ECOD combines the tail probabilities across all features to produce an overall anomaly score, without assuming any parametric distribution. This makes it simple, interpretable, and effective for high-dimensional tabular data with only normal samples for training.

**DRL.** DRL (Decomposed Representation Learning) is an anomaly detection method designed for tabular data that learns only from normal samples. Instead of relying purely on reconstruction, DRL enforces a structured latent space where each normal representation is expressed as a linear combination of fixed, orthogonal basis vectors. This decomposition constraint encourages normal samples to occupy a well-defined subspace, while anomalies naturally fall outside of it. By amplifying discrepancies in this constrained representation space, DRL improves separation between normal and anomalous patterns. The anomaly score is derived from how poorly a test sample aligns with the decomposed latent space, optionally combined with reconstruction error.

**LLM's opinion on PCA's weakness.** PCA's Blind Spot: PCA-based anomaly detectors learn a low-dimensional subspace (the "principal subspace") that captures the maximum variance of the normal training data. The anomaly score is the reconstruction error, which is the distance of a sample to this subspace. Consequently, PCA is good at detecting samples that lie far from this subspace. Its primary weakness is detecting anomalies that lie *within* this learned subspace but are far from the cluster of normal data. Such points are "anomalous" because they are in an unobserved region, but have a low reconstruction error, fooling the detector.

**LLM's opinion on IForest's weakness.** The core weakness of the Isolation Forest (IForest) lies in its reliance on axis-aligned splits to isolate data points. A sample is deemed anomalous if it can be isolated with just a few splits (resulting in a short path length). Conversely, a sample is considered normal if it is "deep" within the data cluster and requires numerous splits for isolation (leading to a long path length). Therefore, it exhibits insufficient sensitivity to samples that reside in the principal component subspace and are far from the data center, to synthetic anomalies within dense regions, and to samples where all feature values are normal yet their combinations are novel. This is because these anomalous samples all result in longer split paths, thereby receiving lower anomaly scores in IForest.

**LLM's opinion on OCSVM's weakness.** The core mechanism of One-Class SVM (OCSVM) is to learn a decision boundary that encloses normal data, and determine whether a sample is anomalous based on the sample's relative position to this boundary (quantified by an anomaly score). The decision boundary of OCSVM is entirely defined by support vectors, while "borderline normal samples" themselves are the group "closest to anomalies" among normal data—they have the highest anomaly scores among all normal samples. When such samples are slightly pushed across the decision boundary and converted into anomalous samples, OCSVM becomes even less able to effectively distinguish between "normal samples inside the boundary" and "neighboring anomalous samples outside the boundary" : the two groups are extremely close in the feature space, and the difference in their anomaly scores is negligible. This ultimately leads to the formation of an ambiguous zone for anomaly determination near the decision boundary of OCSVM, resulting in obvious detection blind spots.

**LLM's opinion on ECOD's weakness.** ECOD assesses the anomaly of a sample by evaluating how extreme each feature value is in its univariate distribution,independently of other features. A hard anomaly for ECOD is therefore a sample whose feature values are individually common (i.e., not in the tails of their distributions) but whose *combination* of feature values is novel or unseen.

**LLM's opinion on DRL's weakness.** DRL first learns a structured latent space spanned by orthogonal basis vectors, and then uses the degree of deviation of samples from the "structured latent subspace defined by the decomposition process" as the basis for anomaly scoring. Under this mechanism, normal samples form a tight alignment with this latent subspace; once a sample's latent space representation deviates, the model will classify it as an anomaly. However,DRL often struggles to effectively detect samples that are inherently anomalous yet maintain a high degree of alignment with the learned latent subspace.

### 6.4.2 ADDITIONAL CONVENTIONAL AND ADVANCED TAD BASELINES.

In addition to these detector-based integrations, we also conduct direct comparisons against conventional and advanced TAD baselines to further evaluate benchmark performance. Specifically, KNN (Ramaswamy et al., 2000), LOF (Breunig et al., 2000) represent classic AD approaches that continue to maintain popularity. In addition, we compare our method to recent deep learning based methods, namely Deep SVDD (Ruff et al., 2018), AutoEncoder (Chen et al., 2018), ICL (Shenkar & Wolf, 2022), DTE (Livernoche et al., 2024), MCM (Yin et al., 2024) and NPT-AD (Thimonier et al., 2024). A recent attempt AnoLLLM (Tsai et al., 2025) has been made to apply LLMs to tabular anomaly detection by fine-tuning them on normal samples via next-token prediction, enabling the model to act as a generative learner of the normal distribution. However, this approach still requires direct access to raw feature values and incurs the additional cost of fine-tuning, limiting its practicality.

### 6.4.3 IMPLEMENTATION DETAILS.

We use the popular PyOD python package (Zhao et al., 2019) to implement OCSVM, LOF, PCA, IForest, ECOD, Deep SVDD and AutoEncoder. We use the DeepOD python library (Xu et al., 2023) to implement ICL. The implementation of the other methods is based on their official open-source code releases. Following latest works (Yin et al., 2024; Ye et al., 2025), We implement all baseline models' hyperparameters following their original papers. All the methods are implemented with identical dataset partitioning and preprocessing procedures, following previous works (Ye et al., 2025).

## 6.5 Example Prompt Used in LLM-DAS

Here, we take the IForest as an example, i.e., $t =$ "IForest".

---

**Example Prompt $p_{\text{initial}}^t$ (the input of Eq. 2) for Generating Detector Description.**

You are an expert in anomaly detection systems. The training set contains only normal samples. I will provide you the <IForest>. Please return a description of this detector, and the pseudo code of the algorithm (step1, step2,... ). The answer format for another detector is:
**Here you should provide one paragraph for description**
The main steps of <detector type> for anomaly detection are:
**Here you should provide the main steps for this algorithm**
* xxx.
* xxx.
* xxx.
...

---

**Example Prompt $p_{\text{description}}^t$ (the output of Eq. 2), Serving as One Part of $p_{\text{code}}^t$.**

Isolation Forest (IForest) is an ensemble-based anomaly detection method that isolates anomalies instead of profiling normal data points. The key intuition is that anomalies are few and different, so they are easier to isolate using random partitions. IForest builds multiple random binary trees (isolation trees) by recursively splitting data based on randomly chosen features and split values. Data points that require fewer splits to be isolated are more likely to be anomalies, while normal points typically require more splits. The average path length across trees is used to compute an anomaly score, where shorter paths indicate higher anomaly likelihood.
The main steps of Isolation Forest (IForest) for anomaly detection are:
* Randomly select a subsample of the training data to build each tree.
* For each tree, recursively split the data by choosing a random feature and a random split value within its range, until the data is isolated or the maximum tree depth is reached.
* For each sample, compute its path length (the number of splits required for isolation) in every tree.
* Average the path lengths across all trees and normalize them to compute the anomaly score.
* Classify samples as anomalies if their score exceeds a chosen threshold (higher score $\rightarrow$ more anomalous).

---

**Example Prompt** $p_{code}^t = p_{description}^t + p_{objective}^t + p_{requirements}^t$ **(the input of Eq. 3) for Generating Synthesis Code**

You are an expert in anomaly detection systems. The training set contains only normal samples. We use a IForest detector, where the anomaly score is computed using model.predict_score(). The higher the score, the more anomalous the sample.

**The description of IForest.**

Isolation Forest (IForest) is an ensemble-based anomaly detection method that isolates anomalies instead of profiling normal data points. The key intuition is that anomalies are few and different, so they are easier to isolate using random partitions. IForest builds multiple random binary trees (isolation trees) by recursively splitting data based on randomly chosen features and split values. Data points that require fewer splits to be isolated are more likely to be anomalies, while normal points typically require more splits. The average path length across trees is used to compute an anomaly score, where shorter paths indicate higher anomaly likelihood.

The main steps of Isolation Forest (IForest) for anomaly detection are:

* Randomly select a subsample of the training data to build each tree.
* For each tree, recursively split the data by choosing a random feature and a random split value within its range, until the data is isolated or the maximum tree depth is reached.
* For each sample, compute its path length (the number of splits required for isolation) in every tree.
* Average the path lengths across all trees and normalize them to compute the anomaly score.
* Classify samples as anomalies if their score exceeds a chosen threshold (higher score → more anomalous).

**Your task** is to write a Python function generate_hard_anomalies(...) that generates anomalies which are the most difficult for the IForest detector to detect. This means that you the generated anomalies should have relatively low anomaly score, thus they are hard to be detected. But these anomalies are helpful to build a more robust detector. After the Python function is completed, users can provide the function with:

* A trained IForest model (model) that exposes predict_score(),
* training samples (X_train).

**Requirements:** Your should strictly follow below requirements:

1. You must use your expertise to give anomalies generation policies that are specific designed for IForest, not a model-agnostic policy.

2. Generated samples should have as low a score as possible from model.predict_score(). To achieve it, you can first find the set of 'borderline' normal training samples based on your unique and professional understanding to IForest, not only based on the anomaly score. Then transform them to anomalies that is tailor-designed for IForest. Please note that the transformation function should be specific for IForest, which means that it is not a general transformation for other detectors.

3. For the model, you can only use the function model.predict_score.

4. Use NumPy to generate the samples, and output an array of shape (n_samples, d). And it should generate anomalies as much as I want.

5. The function should allow setting:

* the number of samples (n_samples),
* the trained IForest model (model),
* training samples (X_train).

Return only the complete Python function generate_hard_anomalies(...), with policy you used for genenrating anomalies and clear comments explaining key steps.

## 6.6 EXAMPLE GENERATED CODE BY LLM

Code 1: Generated Code (the output of Eq. 3) for IForest-Specific Hard Anomaly Generation

```
import numpy as np
def generate_hard_anomalies(n_samples: int, model, X_train: np.
    ndarray):
```

```
 3        """
 4        Generates anomalies that are difficult for a trained Isolation
              Forest (IForest) model to detect.
 5
 6        These "hard" anomalies are designed to have relatively low
              anomaly scores, making them resemble normal data from the
              perspective of the IForest, thus challenging the detector.
 7
 8        Policy for Generating IForest-Specific Hard Anomalies:
 9        ------------------------------------------------------
10        The core weakness of IForest is that it relies on axis-aligned
              splits to isolate points.
11        A sample is considered anomalous if it can be isolated in a
              few splits (short path length).
12        Conversely, a sample is normal if it is "deep" within the data
              cloud, requiring many splits to be isolated (long path
              length).
13
14        Our strategy exploits this by creating new points that are
              conceptually anomalous but are geometrically located in a
              way that maximizes their path length, thereby fooling the
              IForest.
15
16        The policy consists of two main steps:
17
18        1.  Identify 'Borderline' Normal Samples:
19            These are the normal training samples that lie on the
                  fringe or edge of the dense data cloud.
20            The IForest model is already less certain about these
                  points, assigning them the highest anomaly scores among
                   all normal samples.
21            They serve as perfect "seeds" for our hard anomalies
                  because they are already close to the decision boundary
                  .
22            We identify these by finding the training samples in the
                  top percentile of anomaly scores.
23
24        2.  Transform Seeds into Hard Anomalies via 'Controlled
              Extrapolation':
25            We transform these borderline seeds into anomalies.
26            A naive transformation (e.g., adding large random noise)
                  would create an obvious outlier with a short path
                  length, which is an*easy* anomaly.
27            To create a *hard* anomaly, we must move the seed point in
                   a way that doesn't significantly shorten its path
                  length.
28
29            The chosen transformation is a controlled extrapolation
                  along the vector pointing from the center of the
                  training data to the borderline seed point.
30            -  `new_anomaly = seed + magnitude * (seed - data_center)`
31            This pushes the borderline point slightly further away
                  from the data's core, making it a true anomaly.
32            However, because it's moved along an existing axis of the
                  data's distribution, it remains "camouflaged".
33            The new point is not in a completely isolated region of
                  the feature space but rather just outside the densest
                  part of the manifold.
34            An IForest will still require a relatively high number of
                  splits to isolate it compared to a random, distant
                  outlier, resulting in a deceptively low anomaly score.
35
36        This tailored approach ensures the generated anomalies
              specifically target the operational logic of the Isolation
              Forest, making them truly "hard" to detect.
```

```
37
38        Args:
39            n_samples (int): The number of hard anomaly samples to
                  generate.
40            model: A trained Isolation Forest model with a '
                  predict_score()' method.The method should return higher
                  scores for more anomalous samples.
41            X_train (np.ndarray): The training data (normal samples)
                  used to train the model,with shape (n_train_samples,
                  n_features).
42
43        Returns:
44            np.ndarray: An array of generated hard anomalies with
                  shape (n_samples, n_features).
45        """
46        # 1. All package imports must be done inside the function.
47        import numpy as np
48
49        if not hasattr(model, 'predict_score'):
50            raise AttributeError("The provided model must have a '
                  predict_score' method.")
51        if not isinstance(X_train, np.ndarray) or X_train.ndim != 2:
52            raise ValueError("X_train must be a 2D NumPy array.")
53        if n_samples <= 0:
54            raise ValueError("n_samples must be a positive integer.")
55
56        # 2. Identify 'borderline' normal training samples.
57        # These are the normal samples that the model already finds
              most anomalous.
58        # We define them as the top 20% of training samples with the
              highest anomaly scores.
59        # This percentile is a tunable parameter but 80 (for top 20%)
              is a robust choice.
60        print("Step 1: Identifying borderline normal samples from the
              training set...")
61        train_scores = model.predict_score(X_train)
62
63        # Find the score threshold for the 80th percentile.
64        # Samples with scores above this are our "borderline" seeds.
65        score_threshold = np.percentile(train_scores, 80)
66
67        borderline_indices = np.where(train_scores >= score_threshold)
                  [0]
68
69        # If no samples are above the threshold (e.g., all scores are
                  identical),
70        # fall back to using the entire training set as potential
                  seeds.
71        if len(borderline_indices) == 0:
72            borderline_indices = np.arange(len(X_train))
73
74        borderline_samples = X_train[borderline_indices]
75        print(f"Found {len(borderline_samples)} borderline samples to
              use as seeds.")
76
77        # 3. Transform borderline samples into hard anomalies.
78        # We will generate each hard anomaly from a randomly chosen
              borderline seed.
79        print(f"Step 2: Generating {n_samples} hard anomalies via
              controlled extrapolation...")
80
81        # Calculate the geometric center (mean) of the normal data
              distribution.
82        data_center = np.mean(X_train, axis=0)
83
```

```
84      hard_anomalies = []
85      for _ in range(n_samples):
86          # Randomly select a borderline sample to serve as the base
                 for our new anomaly.
87          # We sample with replacement, allowing us to generate more
                 anomalies than seeds.
88          seed_idx = np.random.randint(0, len(borderline_samples))
89          seed_sample = borderline_samples[seed_idx]
90
91          # Calculate the direction vector pointing from the data
                 center to the seed sample.
92          # This vector represents a direction of natural variation
                 in the data.
93          direction_vector = seed_sample - data_center
94
95          # Define a small, random magnitude for the extrapolation.
96          # A small magnitude ensures the new point is not an
                 obvious outlier but just
97          # beyond the boundary of the normal data cloud.
98          # Values between 5% and 25% of the vector length work well
                 .
99          magnitude = np.random.uniform(0.05, 0.25)
100
101          # Create the new hard anomaly by pushing the seed sample
                 along the direction vector.
102          perturbation = direction_vector * magnitude
103          new_anomaly = seed_sample + perturbation
104
105          hard_anomalies.append(new_anomaly)
106
107      print("Generation complete.")
108      # 4. Return the generated samples as a NumPy array.
109      return np.array(hard_anomalies)
```

## 6.7 FULL RESULTS OF IMPROVEMENT OVER SOURCE DETECTORS

We evaluate the effectiveness of LLM-DAS by measuring its improvements over the original source detectors across 36 benchmark datasets. The full results are provided in Table 4 and Table 5.

## 6.8 FULL RESULTS OF COMPARISON WITH OTHER BASELINE METHODS

We further compare the performance of LLM-DAS with conventional and advanced TAD baselines. The full results are detailed in Table 6 and Table 7. Since LLM-DAS and the recent finetune-based AnoLLM (Tsai et al., 2025) have some shared datasets, we cite the results of AnoLLM due to its high computational cost, and perform other methods on the shared subsets.

Table 4: The performance improvement achieved by LLM-DAS ($F_t$) over trained source detectors ($f_t$) on 36 datasets in terms of AUC-PR ($\uparrow$).

| | PCA | +LLM-DAS | IForest | +LLM-DAS | OCSVM | +LLM-DAS | ECOD | +LLM-DAS | DRL | +LLM-DAS |
|---|---|---|---|---|---|---|---|---|---|---|
| Abalone | 0.839±0 | **0.841±0** | 0.848±0.003 | **0.855±0.005** | 0.846±0 | **0.849±0.005** | 0.655±0 | 0.645±0.002 | 0.885±0.006 | **0.902±0.001** |
| Amazon | 0.107±0 | **0.108±0.001** | 0.109±0.002 | 0.109±0.002 | 0.105±0 | **0.106±0** | 0.104±0 | 0.104±0.002 | 0.121±0.007 | 0.12±0.008 |
| Annthyroid | 0.566±0 | **0.66±0.009** | 0.615±0.053 | **0.62±0.055** | 0.183±0 | **0.248±0.01** | 0.4±0 | 0.304±0.008 | 0.676±0.018 | **0.762±0.021** |
| Arrhythmia | 0.534±0 | **0.551±0.013** | 0.51±0.034 | **0.602±0.037** | 0.534±0 | **0.609±0.012** | 0.446±0.005 | **0.466±0.003** | 0.627±0.036 | **0.73±0.014** |
| Breastw | 0.993±0 | **0.996±0.001** | 0.945±0 | **0.995±0.002** | 0.993±0 | **0.995±0** | 0.952±0 | **0.977±0.038** | 0.997±0.004 | 0.997±0.001 |
| Campaign | 0.488±0 | 0.488±0 | 0.461±0.007 | 0.452±0.007 | 0.475±0 | 0.435±0 | 0.471±0 | **0.503±0** | 0.501±0.006 | 0.495±0.004 |
| Cardio | 0.863±0 | 0.824±0.014 | 0.702±0.039 | 0.622±0.032 | 0.861±0 | 0.808±0.015 | 0.364±0 | **0.716±0.026** | 0.833±0.024 | **0.886±0.038** |
| Cardiotocography | 0.697±0 | **0.734±0.006** | 0.604±0.032 | **0.638±0.041** | 0.662±0 | **0.703±0.057** | 0.697±0 | 0.664±0.009 | 0.754±0 | **0.805±0.026** |
| Comm.and.crime | 0.889±0 | 0.881±0.002 | 0.894±0.009 | **0.896±0.006** | 0.837±0 | **0.843±0.022** | 0.685±0 | **0.702±0.021** | 0.916±0.004 | **0.947±0.004** |
| Fault | 0.604±0 | 0.603±0.009 | 0.595±0.014 | 0.577±0.02 | 0.606±0 | 0.542±0.01 | 0.517±0 | **0.573±0.007** | 0.665±0 | **0.705±0.043** |
| Glass | 0.09±0 | **0.128±0.016** | 0.095±0.006 | **0.12±0.02** | 0.09±0 | **0.201±0.033** | 0.111±0 | **0.125±0.045** | 0.167±0 | **0.493±0.116** |
| Hepatitis | 0.583±0 | **0.626±0.038** | 0.418±0.022 | **0.566±0.069** | 0.282±0 | **0.3±0.007** | 0.405±0 | **0.477±0.024** | 0.663±0.109 | **0.696±0.107** |
| Imgseg | 0.772±0 | **0.824±0.014** | 0.756±0.012 | **0.859±0.023** | 0.788±0 | **0.859±0.011** | 0.737±0 | **0.844±0.02** | 0.924±0.007 | **0.944±0.008** |
| Ionosphere | 0.897±0 | **0.926±0.026** | 0.977±0.008 | 0.869±0.035 | 0.897±0 | 0.838±0.012 | 0.971±0 | 0.896±0.007 | 0.99±0.013 | 0.985±0.004 |
| Lympho | 1±0 | 0.989±0.022 | 0.959±0.023 | **0.979±0.02** | 0.811±0 | **0.934±0.03** | 0.897±0 | **0.97±0.028** | 1±0.062 | 0.871±0.07 |
| Mammography | 0.417±0 | **0.579±0.008** | 0.333±0.032 | **0.571±0.019** | 0.418±0 | **0.536±0.022** | 0.538±0 | **0.543±0.007** | 0.841±0.02 | 0.529±0.025 |
| Mnist | 0.65±0 | 0.65±0 | 0.535±0.013 | **0.601±0.01** | 0.169±0 | **0.181±0.012** | 0.302±0.007 | **0.322±0.004** | 0.887±0.017 | **0.911±0.018** |
| Musk | 1±0 | 0.997±0.006 | 0.528±0.202 | **1±0** | 0.061±0 | **0.868±0.075** | 0.982±0 | **1±0** | 1±0.009 | 1±0 |
| Optdigits | 0.06±0 | 0.06±0 | 0.157±0.015 | **0.179±0.043** | 0.069±0 | **0.928±0** | 0.067±0.006 | **0.087±0.003** | 0.936±0 | **0.957±0.017** |
| Parkinson | 0.93±0 | **0.944±0.004** | 0.96±0.004 | 0.951±0.011 | 0.889±0 | **0.92±0.021** | 0.891±0 | 0.863±0.025 | 0.921±0.009 | **0.966±0.007** |
| Pendigits | 0.386±0 | **0.485±0.047** | 0.513±0.02 | **0.602±0.049** | 0.518±0 | 0.491±0.044 | 0.415±0 | **0.498±0.028** | 0.936±0.135 | 0.885±0.025 |
| Pima | 0.701±0 | **0.751±0.005** | 0.666±0.007 | **0.74±0.017** | 0.701±0 | 0.651±0.036 | 0.588±0 | 0.569±0.014 | 0.745±0.012 | **0.791±0.007** |
| Satellite | 0.778±0 | **0.792±0.001** | 0.858±0.006 | 0.808±0.007 | 0.778±0 | **0.832±0.005** | 0.833±0 | 0.803±0.006 | 0.869±0.008 | **0.894±0.009** |
| Satimage-2 | 0.919±0 | **0.931±0.01** | 0.885±0.007 | 0.871±0.085 | 0.919±0 | 0.9±0.014 | 0.778±0 | **0.934±0.008** | 0.97±0.044 | **0.98±0.006** |
| Shuttle | 0.963±0 | **0.991±0.002** | 0.917±0.005 | **0.995±0.001** | 0.949±0 | **0.991±0.001** | 0.982±0 | **0.992±0** | 0.982±0.006 | **0.999±0** |
| SpamBase | 0.818±0 | **0.823±0.002** | 0.89±0.006 | 0.889±0.007 | 0.814±0 | 0.814±0 | 0.713±0 | **0.737±0.007** | 0.841±0.008 | **0.933±0.013** |
| Speech | 0.028±0 | **0.029±0.001** | 0.035±0.009 | 0.032±0.007 | 0.028±0 | **0.029±0.002** | 0.029±0 | **0.031±0.003** | 0.058±0.011 | **0.119±0.01** |
| Thyroid | 0.813±0 | **0.815±0.022** | 0.606±0.025 | **0.886±0.053** | 0.813±0 | 0.633±0.014 | 0.681±0 | **0.765±0.01** | 0.863±0.021 | **0.877±0.017** |
| Vertebral | 0.138±0 | **0.143±0.004** | 0.134±0.001 | **0.269±0.084** | 0.152±0 | **0.273±0.038** | 0.192±0 | **0.403±0.068** | 0.285±0.043 | **0.566±0.093** |
| Vowels | 0.105±0 | **0.109±0.01** | 0.098±0.013 | **0.178±0.011** | 0.297±0 | 0.21±0.02 | 0.177±0 | **0.274±0.015** | 0.451±0.025 | **0.892±0.016** |
| Wbc | 0.839±0 | **0.84±0.013** | 0.857±0.015 | 0.823±0.033 | 0.839±0 | 0.793±0.034 | 0.722±0 | **0.881±0.016** | 0.974±0.032 | 0.934±0.018 |
| WDBC | 0.983±0 | 0.98±0.01 | 0.975±0.029 | **0.994±0.009** | 0.435±0 | **0.874±0.059** | 0.773±0 | **0.93±0.046** | 1±0.016 | 1±0 |
| Wilt | 0.064±0 | **0.097±0.011** | 0.085±0.004 | **0.278±0.049** | 0.225±0 | 0.219±0.06 | 0.077±0 | **0.375±0.043** | 0.454±0.058 | **0.593±0.05** |
| Wine | 0.133±0 | **0.846±0.044** | 0.246±0.026 | **0.456±0.076** | 0.142±0 | **0.334±0.172** | 0.358±0 | 0.289±0.098 | 1±0.03 | 1±0 |
| WPBC | 0.394±0 | **0.408±0.015** | 0.376±0.012 | **0.399±0.017** | 0.397±0 | **0.407±0.015** | 0.353±0 | **0.393±0.013** | 0.502±0.057 | **0.567±0.044** |
| Yeast | 0.468±0 | **0.505±0.005** | 0.465±0.004 | **0.546±0.019** | 0.48±0 | **0.515±0.007** | 0.494±0 | **0.543±0.017** | 0.542±0.008 | 0.528±0.009 |
| Average AUC-PR | 0.597±0 | **0.638±0.011** | 0.572±0.02 | **0.634±0.027** | 0.53±0 | **0.602±0.024** | 0.538±0.001 | **0.589±0.019** | 0.744±0.024 | **0.785±0.024** |

Table 5: The performance improvement achieved by LLM-DAS ($F_t$) over trained source detectors ($f_t$) on 36 datasets in terms of AUC-ROC ($\uparrow$).

| | PCA | +LLM-DAS | IForest | +LLM-DAS | OCSVM | +LLM-DAS | ECOD | +LLM-DAS | DRL | +LLM-DAS |
|---|---|---|---|---|---|---|---|---|---|---|
| Abalone | $0.704_{\pm0}$ | $\mathbf{0.705}_{\pm0.001}$ | $0.735_{\pm0.006}$ | $\mathbf{0.746}_{\pm0.007}$ | $0.724_{\pm0}$ | $\mathbf{0.728}_{\pm0.007}$ | $0.487_{\pm0}$ | $0.475_{\pm0.002}$ | $0.809_{\pm0.008}$ | $\mathbf{0.837}_{\pm0.003}$ |
| Amazon | $0.549_{\pm0}$ | $0.549_{\pm0}$ | $0.559_{\pm0.008}$ | $0.559_{\pm0.008}$ | $0.542_{\pm0}$ | $0.542_{\pm0}$ | $0.538_{\pm0}$ | $\mathbf{0.539}_{\pm0.004}$ | $0.57_{\pm0.016}$ | $0.553_{\pm0.025}$ |
| Annthyroid | $0.852_{\pm0}$ | $\mathbf{0.906}_{\pm0.006}$ | $0.911_{\pm0.022}$ | $\mathbf{0.925}_{\pm0.018}$ | $0.555_{\pm0}$ | $\mathbf{0.667}_{\pm0.013}$ | $0.785_{\pm0}$ | $0.745_{\pm0.013}$ | $0.924_{\pm0.007}$ | $\mathbf{0.949}_{\pm0.013}$ |
| Arrhythmia | $0.768_{\pm0}$ | $\mathbf{0.776}_{\pm0.015}$ | $0.773_{\pm0.01}$ | $\mathbf{0.795}_{\pm0.022}$ | $0.769_{\pm0}$ | $\mathbf{0.774}_{\pm0.011}$ | $0.72_{\pm0.007}$ | $\mathbf{0.74}_{\pm0.005}$ | $0.774_{\pm0.013}$ | $\mathbf{0.832}_{\pm0.007}$ |
| Breastw | $0.994_{\pm0}$ | $\mathbf{0.996}_{\pm0.001}$ | $0.972_{\pm0}$ | $\mathbf{0.995}_{\pm0.002}$ | $0.994_{\pm0}$ | $\mathbf{0.996}_{\pm0}$ | $0.965_{\pm0}$ | $\mathbf{0.985}_{\pm0.021}$ | $0.997_{\pm0.003}$ | $0.997_{\pm0}$ |
| Campaign | $0.771_{\pm0}$ | $0.771_{\pm0}$ | $0.727_{\pm0.005}$ | $\mathbf{0.729}_{\pm0.005}$ | $0.763_{\pm0}$ | $0.689_{\pm0}$ | $0.756_{\pm0}$ | $\mathbf{0.775}_{\pm0.001}$ | $0.771_{\pm0.004}$ | $0.753_{\pm0.004}$ |
| Cardio | $0.966_{\pm0}$ | $0.958_{\pm0.006}$ | $0.922_{\pm0.017}$ | $0.919_{\pm0.015}$ | $0.965_{\pm0}$ | $0.954_{\pm0.002}$ | $0.637_{\pm0}$ | $\mathbf{0.931}_{\pm0.005}$ | $0.956_{\pm0.009}$ | $\mathbf{0.968}_{\pm0.015}$ |
| Cardiotocography | $0.789_{\pm0}$ | $\mathbf{0.804}_{\pm0.007}$ | $0.725_{\pm0.028}$ | $\mathbf{0.768}_{\pm0.036}$ | $0.752_{\pm0}$ | $\mathbf{0.761}_{\pm0.034}$ | $0.789_{\pm0}$ | $0.778_{\pm0.008}$ | $0.84_{\pm0}$ | $\mathbf{0.853}_{\pm0.029}$ |
| Comm.and.crime | $0.787_{\pm0}$ | $0.779_{\pm0.002}$ | $0.804_{\pm0.014}$ | $\mathbf{0.807}_{\pm0.01}$ | $0.705_{\pm0}$ | $\mathbf{0.713}_{\pm0.035}$ | $0.517_{\pm0}$ | $\mathbf{0.525}_{\pm0.031}$ | $0.846_{\pm0.008}$ | $\mathbf{0.894}_{\pm0.007}$ |
| Fault | $0.559_{\pm0}$ | $\mathbf{0.592}_{\pm0.005}$ | $0.561_{\pm0.018}$ | $0.558_{\pm0.018}$ | $0.568_{\pm0}$ | $0.561_{\pm0.006}$ | $0.504_{\pm0}$ | $\mathbf{0.533}_{\pm0.005}$ | $0.628_{\pm0}$ | $\mathbf{0.698}_{\pm0.05}$ |
| Glass | $0.548_{\pm0}$ | $\mathbf{0.62}_{\pm0.03}$ | $0.577_{\pm0.014}$ | $\mathbf{0.624}_{\pm0.027}$ | $0.548_{\pm0}$ | $\mathbf{0.644}_{\pm0.015}$ | $0.624_{\pm0}$ | $0.425_{\pm0.073}$ | $0.697_{\pm0}$ | $\mathbf{0.906}_{\pm0.044}$ |
| Hepatitis | $0.812_{\pm0}$ | $\mathbf{0.835}_{\pm0.022}$ | $0.726_{\pm0.02}$ | $\mathbf{0.778}_{\pm0.021}$ | $0.496_{\pm0}$ | $\mathbf{0.513}_{\pm0.007}$ | $0.699_{\pm0}$ | $\mathbf{0.707}_{\pm0.052}$ | $0.812_{\pm0.038}$ | $\mathbf{0.869}_{\pm0.035}$ |
| Imgseg | $0.674_{\pm0}$ | $\mathbf{0.708}_{\pm0.031}$ | $0.686_{\pm0.021}$ | $\mathbf{0.833}_{\pm0.049}$ | $0.741_{\pm0}$ | $\mathbf{0.778}_{\pm0.016}$ | $0.624_{\pm0}$ | $\mathbf{0.813}_{\pm0.024}$ | $0.909_{\pm0.009}$ | $\mathbf{0.915}_{\pm0.011}$ |
| Ionosphere | $0.877_{\pm0}$ | $\mathbf{0.911}_{\pm0.026}$ | $0.968_{\pm0.006}$ | $0.844_{\pm0.031}$ | $0.877_{\pm0}$ | $0.839_{\pm0.005}$ | $0.957_{\pm0}$ | $0.873_{\pm0.008}$ | $0.984_{\pm0.015}$ | $0.978_{\pm0.007}$ |
| Lympho | $1_{\pm0}$ | $0.999_{\pm0.003}$ | $0.995_{\pm0.003}$ | $\mathbf{0.998}_{\pm0.003}$ | $0.981_{\pm0}$ | $\mathbf{0.992}_{\pm0.004}$ | $0.991_{\pm0}$ | $\mathbf{0.997}_{\pm0.003}$ | $1_{\pm0.002}$ | $0.991_{\pm0.003}$ |
| Mammography | $0.899_{\pm0}$ | $\mathbf{0.907}_{\pm0.002}$ | $0.822_{\pm0.005}$ | $\mathbf{0.903}_{\pm0.003}$ | $0.9_{\pm0}$ | $0.897_{\pm0.005}$ | $0.825_{\pm0}$ | $\mathbf{0.907}_{\pm0.001}$ | $0.946_{\pm0.012}$ | $0.883_{\pm0.013}$ |
| Mnist | $0.902_{\pm0}$ | $0.902_{\pm0}$ | $0.862_{\pm0.006}$ | $\mathbf{0.876}_{\pm0.005}$ | $0.5_{\pm0}$ | $\mathbf{0.508}_{\pm0.008}$ | $0.749_{\pm0.011}$ | $0.769_{\pm0.002}$ | $0.971_{\pm0.004}$ | $\mathbf{0.974}_{\pm0.004}$ |
| Musk | $1_{\pm0}$ | $1_{\pm0}$ | $0.952_{\pm0.024}$ | $\mathbf{1}_{\pm0}$ | $0.5_{\pm0}$ | $\mathbf{0.986}_{\pm0.009}$ | $0.999_{\pm0}$ | $\mathbf{1}_{\pm0}$ | $1_{\pm0}$ | $1_{\pm0}$ |
| Optdigits | $0.582_{\pm0}$ | $0.582_{\pm0}$ | $0.824_{\pm0.026}$ | $\mathbf{0.837}_{\pm0.032}$ | $0.634_{\pm0}$ | $\mathbf{0.997}_{\pm0}$ | $0.615_{\pm0.009}$ | $\mathbf{0.635}_{\pm0.001}$ | $0.997_{\pm0}$ | $0.99_{\pm0.005}$ |
| Parkinson | $0.693_{\pm0}$ | $\mathbf{0.729}_{\pm0.013}$ | $0.768_{\pm0.016}$ | $0.74_{\pm0.035}$ | $0.604_{\pm0}$ | $\mathbf{0.67}_{\pm0.058}$ | $0.515_{\pm0}$ | $0.472_{\pm0.071}$ | $0.673_{\pm0.035}$ | $\mathbf{0.832}_{\pm0.026}$ |
| Pendigits | $0.944_{\pm0}$ | $\mathbf{0.949}_{\pm0.005}$ | $0.967_{\pm0.003}$ | $\mathbf{0.975}_{\pm0.005}$ | $0.964_{\pm0}$ | $0.951_{\pm0.001}$ | $0.93_{\pm0}$ | $\mathbf{0.945}_{\pm0.006}$ | $0.998_{\pm0.019}$ | $0.994_{\pm0.002}$ |
| Pima | $0.713_{\pm0}$ | $\mathbf{0.739}_{\pm0.004}$ | $0.674_{\pm0.006}$ | $\mathbf{0.74}_{\pm0.017}$ | $0.713_{\pm0}$ | $0.627_{\pm0.032}$ | $0.583_{\pm0}$ | $0.552_{\pm0.014}$ | $0.765_{\pm0.015}$ | $\mathbf{0.782}_{\pm0.009}$ |
| Satellite | $0.666_{\pm0}$ | $\mathbf{0.678}_{\pm0.001}$ | $0.803_{\pm0.007}$ | $0.754_{\pm0.014}$ | $0.666_{\pm0}$ | $\mathbf{0.799}_{\pm0.007}$ | $0.788_{\pm0}$ | $0.788_{\pm0.006}$ | $0.832_{\pm0.008}$ | $\mathbf{0.873}_{\pm0.013}$ |
| Satimage-2 | $0.982_{\pm0}$ | $\mathbf{0.984}_{\pm0.001}$ | $0.994_{\pm0.002}$ | $0.989_{\pm0.002}$ | $0.982_{\pm0}$ | $\mathbf{0.984}_{\pm0.004}$ | $0.965_{\pm0}$ | $\mathbf{0.993}_{\pm0.001}$ | $0.997_{\pm0.004}$ | $\mathbf{0.998}_{\pm0.001}$ |
| Shuttle | $0.994_{\pm0}$ | $\mathbf{0.998}_{\pm0.001}$ | $0.996_{\pm0.001}$ | $\mathbf{0.999}_{\pm0.001}$ | $0.997_{\pm0}$ | $\mathbf{0.999}_{\pm0}$ | $0.998_{\pm0}$ | $0.998_{\pm0}$ | $0.999_{\pm0}$ | $\mathbf{1}_{\pm0}$ |
| SpamBase | $0.814_{\pm0}$ | $\mathbf{0.815}_{\pm0.002}$ | $0.858_{\pm0.009}$ | $0.857_{\pm0.007}$ | $0.768_{\pm0}$ | $0.768_{\pm0}$ | $0.688_{\pm0}$ | $\mathbf{0.703}_{\pm0.008}$ | $0.829_{\pm0.008}$ | $\mathbf{0.912}_{\pm0.019}$ |
| Speech | $0.364_{\pm0}$ | $\mathbf{0.402}_{\pm0.014}$ | $0.381_{\pm0.014}$ | $\mathbf{0.4}_{\pm0.008}$ | $0.367_{\pm0}$ | $\mathbf{0.385}_{\pm0.008}$ | $0.36_{\pm0}$ | $\mathbf{0.394}_{\pm0.037}$ | $0.582_{\pm0.031}$ | $0.577_{\pm0.035}$ |
| Thyroid | $0.986_{\pm0}$ | $0.985_{\pm0.002}$ | $0.927_{\pm0.001}$ | $\mathbf{0.995}_{\pm0.002}$ | $0.986_{\pm0}$ | $0.956_{\pm0.008}$ | $0.883_{\pm0}$ | $\mathbf{0.972}_{\pm0.009}$ | $0.991_{\pm0.003}$ | $\mathbf{0.987}_{\pm0.003}$ |
| Vertebral | $0.175_{\pm0}$ | $\mathbf{0.192}_{\pm0.015}$ | $0.144_{\pm0.011}$ | $\mathbf{0.423}_{\pm0.101}$ | $0.265_{\pm0}$ | $\mathbf{0.574}_{\pm0.062}$ | $0.412_{\pm0}$ | $\mathbf{0.65}_{\pm0.038}$ | $0.622_{\pm0.057}$ | $\mathbf{0.763}_{\pm0.075}$ |
| Vowels | $0.523_{\pm0}$ | $\mathbf{0.559}_{\pm0.011}$ | $0.59_{\pm0.031}$ | $\mathbf{0.727}_{\pm0.034}$ | $0.756_{\pm0}$ | $0.716_{\pm0.017}$ | $0.615_{\pm0}$ | $\mathbf{0.766}_{\pm0.022}$ | $0.851_{\pm0.004}$ | $\mathbf{0.987}_{\pm0.003}$ |
| Wbc | $0.967_{\pm0}$ | $0.966_{\pm0.001}$ | $0.972_{\pm0.003}$ | $0.953_{\pm0.018}$ | $0.967_{\pm0}$ | $0.952_{\pm0.009}$ | $0.875_{\pm0}$ | $\mathbf{0.968}_{\pm0.008}$ | $0.996_{\pm0.008}$ | $0.988_{\pm0.003}$ |
| WDBC | $0.999_{\pm0}$ | $0.999_{\pm0.001}$ | $0.998_{\pm0.002}$ | $\mathbf{1}_{\pm0.001}$ | $0.964_{\pm0}$ | $\mathbf{0.994}_{\pm0.002}$ | $0.979_{\pm0}$ | $\mathbf{0.993}_{\pm0.006}$ | $1_{\pm0.001}$ | $1_{\pm0}$ |
| Wilt | $0.261_{\pm0}$ | $\mathbf{0.474}_{\pm0.06}$ | $0.46_{\pm0.027}$ | $\mathbf{0.765}_{\pm0.03}$ | $0.792_{\pm0}$ | $0.762_{\pm0.022}$ | $0.375_{\pm0}$ | $\mathbf{0.801}_{\pm0.026}$ | $0.936_{\pm0.007}$ | $0.919_{\pm0.011}$ |
| Wine | $0.447_{\pm0}$ | $\mathbf{0.933}_{\pm0.015}$ | $0.657_{\pm0.095}$ | $\mathbf{0.748}_{\pm0.065}$ | $0.485_{\pm0}$ | $\mathbf{0.669}_{\pm0.221}$ | $0.743_{\pm0}$ | $0.648_{\pm0.11}$ | $1_{\pm0.007}$ | $1_{\pm0}$ |
| WPBC | $0.469_{\pm0}$ | $\mathbf{0.471}_{\pm0.012}$ | $0.498_{\pm0.011}$ | $\mathbf{0.502}_{\pm0.006}$ | $0.47_{\pm0}$ | $\mathbf{0.472}_{\pm0.024}$ | $0.471_{\pm0}$ | $\mathbf{0.517}_{\pm0.024}$ | $0.62_{\pm0.045}$ | $\mathbf{0.67}_{\pm0.031}$ |
| Yeast | $0.432_{\pm0}$ | $\mathbf{0.456}_{\pm0.005}$ | $0.41_{\pm0.007}$ | $\mathbf{0.526}_{\pm0.024}$ | $0.448_{\pm0}$ | $\mathbf{0.464}_{\pm0.002}$ | $0.446_{\pm0}$ | $\mathbf{0.526}_{\pm0.031}$ | $0.512_{\pm0.015}$ | $0.49_{\pm0.017}$ |
| Average AUC-ROC | $0.735_{\pm0}$ | $\mathbf{0.767}_{\pm0.009}$ | $0.755_{\pm0.014}$ | $\mathbf{0.794}_{\pm0.019}$ | $0.714_{\pm0}$ | $\mathbf{0.758}_{\pm0.018}$ | $0.706_{\pm0.001}$ | $\mathbf{0.745}_{\pm0.019}$ | $0.851_{\pm0.012}$ | $\mathbf{0.878}_{\pm0.015}$ |

Table 6: Comparison of AUC-PR (↑) results between other baseline methods and LLM-DAS.

| | OCSVM | KNN | LOF | PCA | IForest | ECOD | DeepSVDD | AutoEncoder | ICL | DTE | MCM | NPT-AD | DRL | AnoLLM(135M) | AnoLLM(360M) | Ours(PCA) | Ours(DRL) |
|---|---|---|---|---|---|---|---|---|---|---|---|---|---|---|---|---|---|
| Annthyroid | 0.1831 | 0.3525 | 0.4513 | 0.5657 | 0.6149 | 0.4002 | 0.3235 | 0.5291 | 0.4114 | 0.6288 | 0.3215 | 0.62 | 0.6761 | 0.631 | 0.648 | 0.6602 | 0.7618 |
| Arrhythmia | 0.5339 | 0.6008 | 0.5277 | 0.5336 | 0.5097 | 0.4461 | 0.6036 | 0.3029 | 0.6155 | 0.4912 | 0.6107 | 0.4345 | 0.627 | 0.636 | 0.642 | 0.5508 | 0.7298 |
| Breastw | 0.9934 | 0.9712 | 0.9923 | 0.9934 | 0.9449 | 0.9522 | 0.9924 | 0.9896 | 0.9656 | 0.8825 | 0.9952 | 0.9813 | 0.9966 | 0.991 | 0.992 | 0.9956 | 0.9972 |
| Glass | 0.0896 | 0.1099 | 0.0923 | 0.0896 | 0.0952 | 0.1113 | 0.0912 | 0.1079 | 0.2573 | 0.2151 | 0.1905 | 0.2204 | 0.167 | 0.247 | 0.234 | 0.1278 | 0.4934 |
| Ionosphere | 0.8969 | 0.9297 | 0.9591 | 0.8969 | 0.9768 | 0.9713 | 0.867 | 0.7328 | 0.9777 | 0.9683 | 0.9802 | 0.9812 | 0.9895 | 0.933 | 0.932 | 0.926 | 0.9846 |
| Lympho | 0.8107 | 0.9401 | 0.9762 | 1 | 0.9593 | 0.8972 | 0.9749 | 0.2709 | 0.6091 | 0.8677 | 0.4204 | 0.9929 | 1 | 0.856 | 0.938 | 0.9889 | 0.8706 |
| Mammography | 0.4178 | 0.381 | 0.4063 | 0.4165 | 0.3334 | 0.538 | 0.419 | 0.253 | 0.1894 | 0.3985 | 0.4755 | 0.364 | 0.8406 | 0.592 | 0.364 | 0.5793 | 0.5288 |
| Musk | 0.0614 | 0.9917 | 1 | 1 | 0.5279 | 0.982 | 1 | 1 | 1 | 1 | 0.639 | 1 | 1 | 1 | 1 | 0.9969 | 1 |
| Optdigits | 0.0692 | 0.8589 | 0.4363 | 0.0602 | 0.157 | 0.0669 | 0.1159 | 0.1418 | 0.1696 | 0.1534 | 0.8885 | 0.4203 | 0.9356 | 0.75 | 0.398 | 0.0602 | 0.9566 |
| Pendigits | 0.5178 | 0.9692 | 0.7855 | 0.3863 | 0.5133 | 0.4145 | 0.0616 | 0.8904 | 0.4039 | 0.4844 | 0.8258 | 0.9671 | 0.936 | 0.623 | 0.554 | 0.4847 | 0.8848 |
| Pima | 0.7008 | 0.7098 | 0.697 | 0.7008 | 0.6662 | 0.5877 | 0.7165 | 0.7174 | 0.6965 | 0.6798 | 0.7389 | 0.7527 | 0.7449 | 0.677 | 0.674 | 0.7509 | 0.7912 |
| Satellite | 0.7778 | 0.8515 | 0.8088 | 0.7778 | 0.8583 | 0.8334 | 0.8217 | 0.8218 | 0.8799 | 0.8479 | 0.8532 | 0.8576 | 0.8692 | 0.91 | 0.891 | 0.7921 | 0.8942 |
| Satimage-2 | 0.9192 | 0.9555 | 0.9692 | 0.9192 | 0.8846 | 0.7775 | 0.9427 | 0.9688 | 0.8124 | 0.6821 | 0.985 | 0.9862 | 0.9703 | 0.988 | 0.974 | 0.9311 | 0.9804 |
| Shuttle | 0.9488 | 0.957 | 0.9601 | 0.9627 | 0.9172 | 0.9815 | 0.9818 | 0.9316 | 0.9811 | 0.9403 | 0.9479 | 0.9151 | 0.9819 | 0.997 | 0.996 | 0.9906 | 0.9991 |
| Speech | 0.0279 | 0.0197 | 0.0315 | 0.0277 | 0.0353 | 0.0287 | 0.04 | 0.027 | 0.0335 | 0.0285 | 0.038 | 0.0584 | 0.0584 | 0.036 | 0.037 | 0.0293 | 0.1189 |
| Thyroid | 0.8134 | 0.5903 | 0.7892 | 0.8134 | 0.6055 | 0.6807 | 0.7282 | 0.8096 | 0.6575 | 0.8167 | 0.8417 | 0.8179 | 0.8626 | 0.696 | 0.74 | 0.8146 | 0.877 |
| Vertebral | 0.1517 | 0.1239 | 0.2063 | 0.1381 | 0.1342 | 0.1917 | 0.159 | 0.1476 | 0.1598 | 0.2514 | 0.1949 | 0.2279 | 0.2854 | 0.289 | 0.181 | 0.1433 | 0.5657 |
| Vowels | 0.2969 | 0.3146 | 0.3277 | 0.1051 | 0.0984 | 0.1772 | 0.1717 | 0.3475 | 0.1574 | 0.381 | 0.0977 | 0.9498 | 0.4506 | 0.839 | 0.599 | 0.109 | 0.8918 |
| Wbc | 0.8391 | 0.8022 | 0.8412 | 0.8391 | 0.8573 | 0.7217 | 0.834 | 0.8578 | 0.7218 | 0.3997 | 0.8887 | 0.8079 | 0.9742 | 0.873 | 0.753 | 0.84 | 0.9341 |
| Wine | 0.1424 | 0.9917 | 0.1253 | 0.1325 | 0.2458 | 0.3578 | 0.1476 | 0.1605 | 0.5659 | 0.9985 | 0.9335 | 0.7746 | 1 | 0.522 | 0.529 | 0.8459 | 1 |
| Yeast | 0.4803 | 0.4737 | 0.4866 | 0.4678 | 0.4654 | 0.4943 | 0.4953 | 0.4833 | 0.5097 | 0.4974 | 0.4631 | 0.4888 | 0.5416 | 0.301 | 0.302 | 0.5053 | 0.5281 |
| Average AUC-PR | 0.5082 | 0.6607 | 0.6129 | 0.5632 | 0.5429 | 0.5529 | 0.547 | 0.5472 | 0.5607 | 0.6006 | 0.6348 | 0.6961 | 0.7575 | 0.6851 | 0.637 | 0.6249 | **0.7994** |
| Average ranking | 12.1429 | 10.7143 | 9.3810 | 11.6429 | 12.1429 | 11.7619 | 10.2381 | 11.2381 | 10.0952 | 10 | 7.9048 | 6.7619 | 3.1667 | 6.619 | 8.119 | 8.5714 | **2.5** |
| Win | 0 | 1 | 1 | 2 | 0 | 0 | 1 | 1 | 1 | 1 | 0 | 2 | 7 | 3 | 1 | 0 | **12** |

Table 7: Comparison of AUC-ROC (↑) results between other baseline methods and LLM-DAS.

| | OCSVM | KNN | LOF | PCA | IForest | ECOD | DeepSVDD | AutoEncoder | ICL | DTE | MCM | NPT-AD | DRL | AnoLLM(135M) | AnoLLM(360M) | Ours(PCA) | Ours(DRL) |
|---|---|---|---|---|---|---|---|---|---|---|---|---|---|---|---|---|---|
| Annthyroid | 0.5551 | 0.6903 | 0.7216 | 0.8519 | 0.9112 | 0.7845 | 0.5678 | 0.7295 | 0.6997 | 0.909 | 0.6894 | 0.8682 | 0.9239 | 0.927 | 0.931 | 0.9065 | 0.9492 |
| Arrhythmia | 0.7689 | 0.7933 | 0.7688 | 0.7684 | 0.7734 | 0.7199 | 0.7941 | 0.5682 | 0.8145 | 0.5912 | 0.8114 | 0.7185 | 0.7742 | 0.825 | 0.822 | 0.7757 | 0.8316 |
| Breastw | 0.9938 | 0.9714 | 0.9937 | 0.9938 | 0.9719 | 0.9649 | 0.9925 | 0.9909 | 0.9725 | 0.9278 | 0.9955 | 0.9848 | 0.9968 | 0.992 | 0.993 | 0.6196 | 0.9972 |
| Glass | 0.548 | 0.6141 | 0.562 | 0.548 | 0.5771 | 0.6235 | 0.5566 | 0.6088 | 0.835 | 0.6964 | 0.7225 | 0.7875 | 0.6969 | 0.819 | 0.797 | 0.9109 | 0.9061 |
| Ionosphere | 0.8765 | 0.9167 | 0.9454 | 0.8765 | 0.9683 | 0.9569 | 0.8552 | 0.6231 | 0.971 | 0.9542 | 0.9726 | 0.9735 | 0.984 | 0.909 | 0.924 | 0.9986 | 0.9779 |
| Lympho | 0.9812 | 0.987 | 0.9977 | 1 | 0.9945 | 0.9906 | 0.9977 | 0.7856 | 0.9546 | 0.9899 | 0.9257 | 0.9993 | 1 | 0.968 | 0.993 | 0.9074 | 0.9911 |
| Mammography | 0.9003 | 0.864 | 0.8922 | 0.8993 | 0.822 | 0.8251 | 0.8879 | 0.8472 | 0.6548 | 0.8862 | 0.9053 | 0.8873 | 0.9455 | 0.915 | 0.876 | 0.9998 | 0.8834 |
| Musk | 0.5 | 0.9917 | 1 | 1 | 0.9521 | 0.9987 | 1 | 1 | 1 | 1 | 0.9752 | 1 | 1 | 1 | 1 | 0.5817 | 1 |
| Optdigits | 0.6338 | 0.9862 | 0.9665 | 0.5817 | 0.8239 | 0.6145 | 0.7603 | 0.6694 | 0.787 | 0.8238 | 0.9947 | 0.9317 | 0.9972 | 0.983 | 0.939 | 0.9487 | 0.9902 |
| Pendigits | 0.9636 | 0.9906 | 0.9905 | 0.9437 | 0.9666 | 0.9295 | 0.4563 | 0.9937 | 0.9142 | 0.9761 | 0.9919 | 0.9987 | 0.9979 | 0.971 | 0.964 | 0.7393 | 0.9941 |
| Pima | 0.7133 | 0.6723 | 0.6913 | 0.7133 | 0.6737 | 0.5834 | 0.7348 | 0.7163 | 0.6727 | 0.6788 | 0.7639 | 0.7553 | 0.7651 | 0.663 | 0.654 | 0.6783 | 0.7818 |
| Satellite | 0.6663 | 0.8139 | 0.7391 | 0.6663 | 0.8026 | 0.7884 | 0.7659 | 0.7233 | 0.8549 | 0.7661 | 0.7962 | 0.806 | 0.832 | 0.902 | 0.877 | 0.9837 | 0.8726 |
| Satimage-2 | 0.9817 | 0.9891 | 0.9961 | 0.9817 | 0.9938 | 0.965 | 0.9881 | 0.9979 | 0.9792 | 0.9967 | 0.9992 | 0.9995 | 0.9968 | 1 | 0.999 | 0.9977 | 0.9984 |
| Shuttle | 0.9969 | 0.9893 | 0.9983 | 0.9936 | 0.9961 | 0.9978 | 0.9952 | 0.9944 | 0.9935 | 0.9993 | 0.9975 | 0.9931 | 0.9992 | 1 | 1 | 0.4020 | 0.9999 |
| Speech | 0.3673 | 0.3561 | 0.3759 | 0.3638 | 0.3812 | 0.3596 | 0.5071 | 0.3633 | 0.4883 | 0.3817 | 0.4409 | 0.5925 | 0.5821 | 0.47 | 0.47 | 0.9849 | 0.5774 |
| Thyroid | 0.9855 | 0.9525 | 0.9856 | 0.9855 | 0.9271 | 0.8827 | 0.9887 | 0.978 | 0.9518 | 0.9863 | 0.9804 | 0.9787 | 0.9911 | 0.975 | 0.983 | 0.1917 | 0.9873 |
| Vertebral | 0.2654 | 0.1171 | 0.4317 | 0.1746 | 0.1444 | 0.4124 | 0.2706 | 0.2375 | 0.2821 | 0.543 | 0.3767 | 0.5347 | 0.6216 | 0.565 | 0.408 | 0.5591 | 0.7629 |
| Vowels | 0.7557 | 0.8174 | 0.8564 | 0.5229 | 0.5902 | 0.6147 | 0.5734 | 0.7905 | 0.702 | 0.8142 | 0.6529 | 0.9935 | 0.8511 | 0.982 | 0.938 | 0.9656 | 0.9869 |
| Wbc | 0.9667 | 0.9536 | 0.967 | 0.9667 | 0.9715 | 0.8747 | 0.9633 | 0.9737 | 0.908 | 0.8054 | 0.9814 | 0.9619 | 0.9963 | 0.964 | 0.952 | 0.9333 | 0.9883 |
| Wine | 0.485 | 0.9917 | 0.4083 | 0.4467 | 0.6571 | 0.7433 | 0.5067 | 0.5356 | 0.915 | 0.9944 | 0.9538 | 0.9622 | 1 | 0.909 | 0.851 | 0.4557 | 1 |
| Yeast | 0.4483 | 0.4366 | 0.4571 | 0.4324 | 0.4095 | 0.4464 | 0.479 | 0.4503 | 0.5076 | 0.4458 | 0.4259 | 0.4691 | 0.512 | 0.744 | 0.73 | 0.7674 | 0.4905 |
| Average | 0.7311 | 0.8045 | 0.7974 | 0.7481 | 0.7766 | 0.7655 | 0.7448 | 0.7418 | 0.8028 | 0.8174 | 0.8263 | 0.8665 | 0.8792 | 0.8801 | 0.8620 | 0.7766 | **0.9032** |
| Rank | 11.8095 | 11.3810 | 8.8333 | 11.5952 | 11.2857 | 12.7143 | 10.3095 | 11.2857 | 10.4762 | 9.3810 | 7.8571 | 6.8571 | 3.7143 | 6.2381 | 7.1905 | 8.6667 | **3.4048** |
| Win | 0 | 0 | 1 | 2 | 0 | 0 | 1 | 1 | 1 | 1 | 0 | 3 | 6 | 3 | 2 | 6 | **7** |

## 6.9 Varying LLM backbones

To investigate the impact of using different LLMs in our framework, given their distinct prior knowledge and reasoning abilities from being trained on various text corpora, we measured performance using not only Gemini-2.5-Pro but also the GPT-4o and Qwen3 as a backbone for comparison. Though the performance of LLM-DAS on Qwen3 slightly drops compared to Gemini-2.5-pro and GPT-4o, it still surpasses the base detector by a large margin. The results clearly show that LLM-DAS maintains its strong performance across all three LLMs.

Table 8: Effects of various LLM backbones. The AUC-PR results are averaged across all datasets. Here, we use OCSVM as the base detector.

| | OCSVM | +LLM-DAS (Qwen3) | +LLM-DAS (GPT-4o) | +LLM-DAS (Gemini-2.5-Pro) |
|---|---|---|---|---|
| Abalone | 0.8459 | 0.8460 | 0.8465 | 0.8487 |
| Amazon | 0.1050 | 0.1048 | 0.1058 | 0.1061 |
| Annthyroid | 0.1831 | 0.4490 | 0.4532 | 0.2483 |
| Arrhythmia | 0.5339 | 0.5898 | 0.5965 | 0.6086 |
| Breastw | 0.9934 | 0.9966 | 0.9969 | 0.9954 |
| Campaign | 0.4749 | 0.4417 | 0.4409 | 0.4349 |
| Cardio | 0.8614 | 0.8625 | 0.8551 | 0.8083 |
| Cardiotocography | 0.6619 | 0.6244 | 0.6773 | 0.7029 |
| Comm.and.crime | 0.8371 | 0.8360 | 0.8394 | 0.8434 |
| Fault | 0.6062 | 0.5597 | 0.5637 | 0.5417 |
| Glass | 0.0896 | 0.2215 | 0.2204 | 0.2006 |
| Hepatitis | 0.2815 | 0.4196 | 0.4481 | 0.3002 |
| Imgseg | 0.7883 | 0.7970 | 0.8056 | 0.8592 |
| Ionosphere | 0.8969 | 0.8212 | 0.8211 | 0.8384 |
| Lympho | 0.8107 | 0.6816 | 0.7346 | 0.9337 |
| Mammography | 0.4178 | 0.4181 | 0.4440 | 0.5359 |
| Mnist | 0.1686 | 0.4366 | 0.4573 | 0.1806 |
| Musk | 0.0614 | 0.9605 | 0.9561 | 0.8676 |
| Optdigits | 0.0692 | 0.8242 | 0.8456 | 0.9281 |
| Parkinson | 0.8892 | 0.9381 | 0.9420 | 0.9200 |
| Pendigits | 0.5178 | 0.5584 | 0.5563 | 0.4912 |
| Pima | 0.7008 | 0.6090 | 0.6377 | 0.6514 |
| Satellite | 0.7778 | 0.8225 | 0.8326 | 0.8319 |
| Satimage-2 | 0.9192 | 0.7065 | 0.8207 | 0.8996 |
| Shuttle | 0.9488 | 0.9955 | 0.9958 | 0.9906 |
| SpamBase | 0.8136 | 0.8331 | 0.8305 | 0.8138 |
| Speech | 0.0279 | 0.0275 | 0.0278 | 0.0294 |
| Thyroid | 0.8134 | 0.3506 | 0.3505 | 0.6326 |
| Vertebral | 0.1517 | 0.2292 | 0.2336 | 0.2733 |
| Vowels | 0.2969 | 0.1860 | 0.1843 | 0.2098 |
| Wbc | 0.8391 | 0.6796 | 0.6811 | 0.7928 |
| WDBC | 0.4348 | 0.8900 | 0.8733 | 0.8740 |
| Wilt | 0.2254 | 0.1914 | 0.2619 | 0.2193 |
| Wine | 0.1424 | 0.4390 | 0.4679 | 0.3335 |
| WPBC | 0.3974 | 0.4010 | 0.3978 | 0.4065 |
| Yeast | 0.4803 | 0.4880 | 0.4872 | 0.5148 |
| Average | 0.5295 | 0.5899 | 0.6025 | 0.6019 |

## 6.10 Computational Efficiency.

As shown in Table 9, LLM-DAS remains computationally efficient in both training and inference stages. This efficiency comes from two key properties: (i) LLM-DAS does not require repeatedly querying or fine-tuning LLMs; instead, the detector-specific synthesis code is generated once per anomaly type and reused across datasets, and (ii) inference only involves an additional lightweight binary classifier on top of the source detector, introducing negligible overhead.

## 6.11 Visualization and KDE Analysis of Synthetic Anomalies

To further examine the quality of anomalies synthesized by LLM-DAS, we utilize T-SNE and Kernel Density Estimation (KDE) plots of anomaly scores on the Thyroid and Vertebral datasets.

Table 9: Runtime in seconds of baseline $f_t$ and LLM-DAS $F_t$ for the training and inference, averaged over all datasets. The training time of $f_t$ is not included in the LLM-DAS training time. LLM-DAS training time includes anomalies synthesis and binary classifier $\tilde{f}_t$ training, while excluding the two-time LLM querying cost for code generation, as the resulting synthesis code is reusable across any dataset.

| Stage | PCA | IForest | OCSVM | ECOD | DRL |
|---|---|---|---|---|---|
| Baseline Train | 0.2891 | 0.8091 | 4.4825 | 0.1313 | 23.6555 |
| LLM-DAS Train | 0.5186 | 1.1474 | 6.4229 | 0.1890 | 1.5253 |
| Baseline Test | 0.2402 | 0.2391 | 1.8455 | 0.2727 | 0.0777 |
| LLM-DAS Test | 0.3389 | 0.3089 | 2.5990 | 0.3207 | 0.2390 |

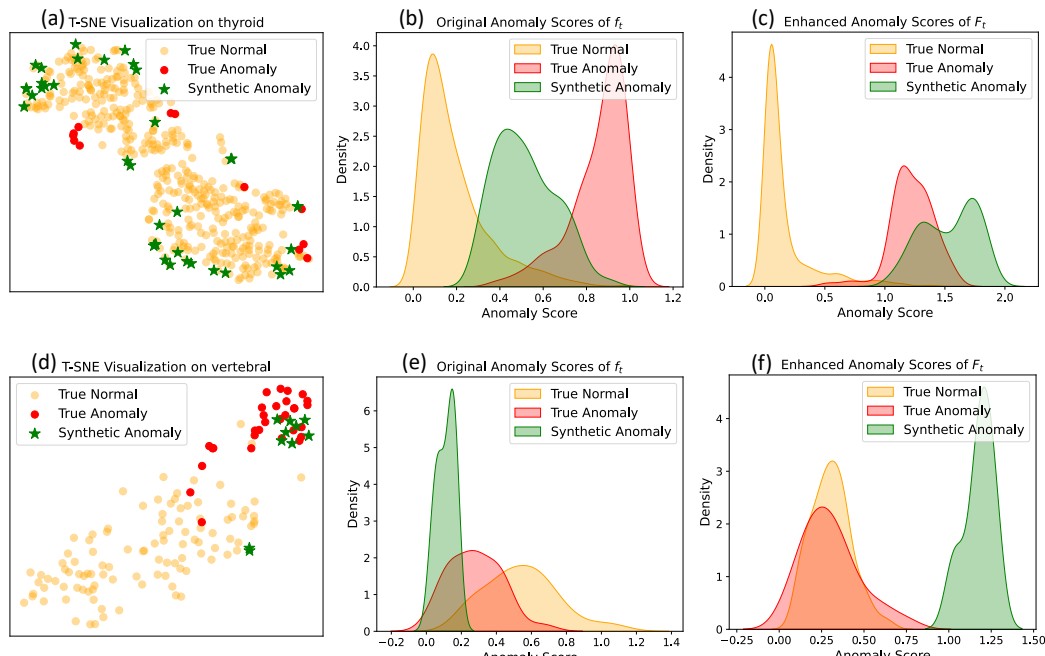

Figure 6: Visualization of synthetic hard anomalies and score distributions on Thyroid and Vertebral test dataset. (a, d) T-SNE plots of normal, real anomaly, and LLM-DAS–generated anomaly samples. (b, e) Kernel density estimation (KDE) of anomaly scores from the source detector $f_t$. (c, f) KDE of anomaly scores from the enhanced detector $F_t$.

## 6.12 SENSITIVITY ANALYSIS

We incorporate the sensitivity analysis for the number of synthetic samples (set to the ratio of training set size), performance robustness across different queries and the type of binary classifiers. We also verify the effectiveness of summing $f_t$ and $\tilde{f}_t$.

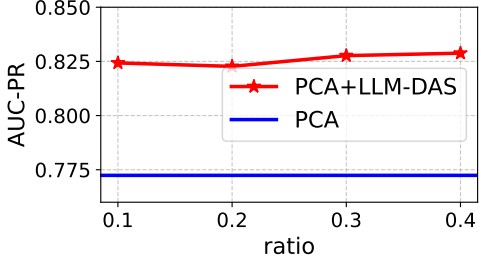 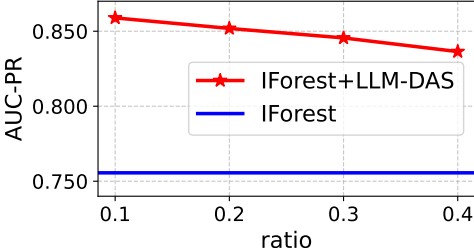

Figure 7: AUC-PR-based performance comparison of different anomalous samples injection ratios on the Imgseg dataset, with results for PCA (left) and IForest (right).

Table 10: AUC-PR performance of PCA and PCA+LLM-DAS under different anomaly synthesis ratios.

| | PCA | LLM-DAS with various anomaly synthesis ratios | | | | | | | | | |
| --- | --- | --- | --- | --- | --- | --- | --- | --- | --- | --- | --- |
| | | 0.1 | 0.2 | 0.3 | 0.4 | 0.5 | 0.6 | 0.7 | 0.8 | 0.9 | 1 |
| Abalone | 0.839 | 0.841 | 0.841 | 0.841 | 0.841 | 0.842 | 0.841 | 0.842 | 0.841 | 0.841 | 0.841 |
| Amazon | 0.107 | 0.108 | 0.108 | 0.108 | 0.107 | 0.108 | 0.108 | 0.107 | 0.108 | 0.108 | 0.108 |
| Annthyroid | 0.566 | 0.660 | 0.701 | 0.690 | 0.698 | 0.698 | 0.695 | 0.712 | 0.692 | 0.691 | 0.688 |
| Arrhythmia | 0.534 | 0.551 | 0.545 | 0.551 | 0.553 | 0.555 | 0.551 | 0.549 | 0.553 | 0.554 | 0.557 |
| Breastw | 0.993 | 0.996 | 0.996 | 0.995 | 0.995 | 0.995 | 0.995 | 0.995 | 0.996 | 0.995 | 0.996 |
| Campaign | 0.488 | 0.488 | 0.488 | 0.488 | 0.488 | 0.488 | 0.488 | 0.488 | 0.488 | 0.488 | 0.489 |
| Cardio | 0.863 | 0.824 | 0.828 | 0.850 | 0.826 | 0.812 | 0.826 | 0.826 | 0.828 | 0.830 | 0.846 |
| Cardiotocography | 0.697 | 0.734 | 0.738 | 0.741 | 0.726 | 0.734 | 0.735 | 0.741 | 0.737 | 0.738 | 0.750 |
| Comm.and.crime | 0.889 | 0.881 | 0.875 | 0.872 | 0.876 | 0.873 | 0.872 | 0.876 | 0.878 | 0.875 | 0.877 |
| Fault | 0.604 | 0.603 | 0.607 | 0.589 | 0.591 | 0.583 | 0.591 | 0.597 | 0.583 | 0.600 | 0.576 |
| Glass | 0.090 | 0.128 | 0.111 | 0.114 | 0.111 | 0.105 | 0.106 | 0.106 | 0.103 | 0.103 | 0.101 |
| Hepatitis | 0.583 | 0.626 | 0.611 | 0.555 | 0.584 | 0.559 | 0.563 | 0.615 | 0.571 | 0.564 | 0.564 |
| Imgseg | 0.772 | 0.824 | 0.823 | 0.828 | 0.829 | 0.825 | 0.840 | 0.832 | 0.838 | 0.831 | 0.831 |
| Ionosphere | 0.897 | 0.926 | 0.919 | 0.944 | 0.939 | 0.902 | 0.912 | 0.926 | 0.922 | 0.910 | 0.914 |
| Lympho | 1.000 | 0.989 | 0.969 | 0.968 | 0.986 | 0.963 | 0.962 | 0.968 | 0.969 | 0.990 | 0.995 |
| Mammography | 0.417 | 0.579 | 0.580 | 0.579 | 0.574 | 0.578 | 0.573 | 0.577 | 0.571 | 0.578 | 0.577 |
| Mnist | 0.650 | 0.650 | 0.650 | 0.650 | 0.650 | 0.662 | 0.651 | 0.650 | 0.650 | 0.650 | 0.650 |
| Musk | 1.000 | 0.997 | 1.000 | 1.000 | 1.000 | 1.000 | 1.000 | 1.000 | 1.000 | 1.000 | 1.000 |
| Optdigits | 0.060 | 0.060 | 0.060 | 0.060 | 0.060 | 0.060 | 0.060 | 0.060 | 0.060 | 0.060 | 0.060 |
| Parkinson | 0.930 | 0.944 | 0.946 | 0.944 | 0.943 | 0.944 | 0.943 | 0.941 | 0.942 | 0.941 | 0.941 |
| Pendigits | 0.386 | 0.485 | 0.467 | 0.485 | 0.478 | 0.550 | 0.544 | 0.548 | 0.585 | 0.558 | 0.546 |
| Pima | 0.701 | 0.751 | 0.751 | 0.751 | 0.751 | 0.750 | 0.748 | 0.748 | 0.744 | 0.742 | 0.739 |
| Satellite | 0.778 | 0.792 | 0.793 | 0.790 | 0.789 | 0.790 | 0.790 | 0.789 | 0.788 | 0.788 | 0.787 |
| Satimage-2 | 0.919 | 0.931 | 0.929 | 0.928 | 0.935 | 0.931 | 0.929 | 0.931 | 0.931 | 0.931 | 0.933 |
| Shuttle | 0.963 | 0.991 | 0.991 | 0.991 | 0.991 | 0.991 | 0.990 | 0.991 | 0.991 | 0.991 | 0.991 |
| SpamBase | 0.818 | 0.823 | 0.825 | 0.824 | 0.824 | 0.823 | 0.823 | 0.825 | 0.826 | 0.826 | 0.825 |
| Speech | 0.028 | 0.029 | 0.031 | 0.033 | 0.031 | 0.032 | 0.035 | 0.029 | 0.029 | 0.030 | 0.030 |
| Thyroid | 0.813 | 0.815 | 0.832 | 0.852 | 0.863 | 0.869 | 0.879 | 0.875 | 0.882 | 0.881 | 0.877 |
| Vertebral | 0.138 | 0.143 | 0.139 | 0.142 | 0.142 | 0.141 | 0.142 | 0.143 | 0.140 | 0.140 | 0.141 |
| Vowels | 0.105 | 0.109 | 0.123 | 0.132 | 0.131 | 0.133 | 0.137 | 0.156 | 0.128 | 0.137 | 0.151 |
| Wbc | 0.839 | 0.840 | 0.834 | 0.840 | 0.827 | 0.818 | 0.831 | 0.846 | 0.825 | 0.826 | 0.826 |
| WDBC | 0.983 | 0.980 | 0.992 | 0.989 | 0.974 | 0.964 | 0.977 | 0.974 | 0.972 | 0.972 | 0.970 |
| Wilt | 0.064 | 0.097 | 0.108 | 0.106 | 0.120 | 0.141 | 0.132 | 0.144 | 0.123 | 0.135 | 0.137 |
| Wine | 0.133 | 0.846 | 0.801 | 0.801 | 0.782 | 0.741 | 0.757 | 0.730 | 0.714 | 0.726 | 0.732 |
| WPBC | 0.394 | 0.408 | 0.406 | 0.415 | 0.398 | 0.413 | 0.415 | 0.416 | 0.409 | 0.400 | 0.404 |
| Yeast | 0.468 | 0.505 | 0.508 | 0.508 | 0.511 | 0.513 | 0.512 | 0.514 | 0.513 | 0.511 | 0.514 |
| Average | 0.597 | 0.638 | 0.637 | 0.638 | 0.637 | 0.636 | 0.638 | 0.641 | 0.637 | 0.637 | 0.638 |

Table 11: AUC-PR performance of IForest and IForest+LLM-DAS under different anomaly synthesis ratios.

| | IForest | LLM-DAS with various anomaly synthesis ratios | | | | | | | | | |
| --- | --- | --- | --- | --- | --- | --- | --- | --- | --- | --- | --- |
| | | 0.1 | 0.2 | 0.3 | 0.4 | 0.5 | 0.6 | 0.7 | 0.8 | 0.9 | 1 |
| Abalone | 0.848 | 0.855 | 0.856 | 0.857 | 0.859 | 0.858 | 0.858 | 0.858 | 0.858 | 0.859 | 0.858 |
| Amazon | 0.109 | 0.109 | 0.109 | 0.109 | 0.109 | 0.109 | 0.109 | 0.109 | 0.109 | 0.109 | 0.109 |
| Annthyroid | 0.615 | 0.620 | 0.628 | 0.628 | 0.630 | 0.647 | 0.634 | 0.643 | 0.636 | 0.633 | 0.647 |
| Arrhythmia | 0.510 | 0.602 | 0.590 | 0.586 | 0.581 | 0.571 | 0.566 | 0.556 | 0.556 | 0.562 | 0.571 |
| Breastw | 0.945 | 0.995 | 0.994 | 0.995 | 0.995 | 0.996 | 0.996 | 0.996 | 0.996 | 0.997 | 0.996 |
| Campaign | 0.461 | 0.452 | 0.452 | 0.452 | 0.452 | 0.452 | 0.452 | 0.452 | 0.452 | 0.452 | 0.452 |
| Cardio | 0.702 | 0.622 | 0.674 | 0.634 | 0.640 | 0.647 | 0.675 | 0.671 | 0.670 | 0.676 | 0.676 |
| Cardiotocography | 0.604 | 0.638 | 0.666 | 0.669 | 0.667 | 0.684 | 0.669 | 0.682 | 0.684 | 0.681 | 0.681 |
| Comm.and.crime | 0.894 | 0.896 | 0.895 | 0.894 | 0.894 | 0.894 | 0.894 | 0.893 | 0.893 | 0.894 | 0.891 |
| Fault | 0.595 | 0.577 | 0.574 | 0.568 | 0.562 | 0.564 | 0.563 | 0.557 | 0.555 | 0.557 | 0.555 |
| Glass | 0.095 | 0.120 | 0.119 | 0.124 | 0.121 | 0.119 | 0.106 | 0.119 | 0.098 | 0.094 | 0.096 |
| Hepatitis | 0.418 | 0.566 | 0.587 | 0.556 | 0.596 | 0.673 | 0.652 | 0.655 | 0.619 | 0.598 | 0.642 |
| Imgseg | 0.756 | 0.859 | 0.852 | 0.846 | 0.836 | 0.824 | 0.829 | 0.836 | 0.829 | 0.829 | 0.829 |
| Ionosphere | 0.977 | 0.869 | 0.818 | 0.763 | 0.746 | 0.699 | 0.675 | 0.670 | 0.654 | 0.641 | 0.631 |
| Lympho | 0.959 | 0.979 | 0.973 | 0.965 | 0.943 | 0.924 | 0.956 | 0.955 | 0.932 | 0.925 | 0.963 |
| Mammography | 0.333 | 0.571 | 0.584 | 0.575 | 0.576 | 0.572 | 0.576 | 0.572 | 0.567 | 0.565 | 0.567 |
| Mnist | 0.535 | 0.601 | 0.610 | 0.610 | 0.610 | 0.610 | 0.610 | 0.610 | 0.610 | 0.610 | 0.610 |
| Musk | 0.528 | 1.000 | 1.000 | 0.999 | 0.999 | 0.989 | 0.996 | 0.990 | 0.962 | 0.959 | 0.933 |
| Optdigits | 0.157 | 0.179 | 0.161 | 0.161 | 0.161 | 0.161 | 0.161 | 0.161 | 0.161 | 0.161 | 0.161 |
| Parkinson | 0.960 | 0.951 | 0.955 | 0.954 | 0.958 | 0.962 | 0.963 | 0.963 | 0.963 | 0.962 | 0.964 |
| Pendigits | 0.513 | 0.602 | 0.604 | 0.608 | 0.596 | 0.626 | 0.630 | 0.639 | 0.652 | 0.642 | 0.643 |
| Pima | 0.666 | 0.740 | 0.738 | 0.735 | 0.740 | 0.736 | 0.739 | 0.743 | 0.741 | 0.742 | 0.744 |
| Satellite | 0.858 | 0.808 | 0.803 | 0.804 | 0.801 | 0.799 | 0.796 | 0.795 | 0.792 | 0.794 | 0.794 |
| Satimage-2 | 0.885 | 0.871 | 0.844 | 0.830 | 0.823 | 0.771 | 0.737 | 0.783 | 0.760 | 0.800 | 0.787 |
| Shuttle | 0.917 | 0.995 | 0.995 | 0.995 | 0.995 | 0.994 | 0.994 | 0.994 | 0.994 | 0.994 | 0.995 |
| SpamBase | 0.890 | 0.889 | 0.890 | 0.890 | 0.890 | 0.889 | 0.890 | 0.889 | 0.889 | 0.889 | 0.888 |
| Speech | 0.035 | 0.032 | 0.037 | 0.037 | 0.034 | 0.033 | 0.036 | 0.035 | 0.032 | 0.039 | 0.037 |
| Thyroid | 0.606 | 0.886 | 0.895 | 0.891 | 0.945 | 0.912 | 0.914 | 0.914 | 0.936 | 0.933 | 0.935 |
| Vertebral | 0.134 | 0.269 | 0.304 | 0.315 | 0.320 | 0.335 | 0.385 | 0.384 | 0.373 | 0.389 | 0.382 |
| Vowels | 0.098 | 0.178 | 0.150 | 0.161 | 0.151 | 0.172 | 0.186 | 0.160 | 0.182 | 0.165 | 0.190 |
| Wbc | 0.857 | 0.823 | 0.844 | 0.842 | 0.846 | 0.840 | 0.841 | 0.830 | 0.832 | 0.827 | 0.831 |
| WDBC | 0.975 | 0.994 | 0.987 | 0.989 | 0.976 | 0.986 | 0.984 | 0.972 | 0.973 | 0.944 | 0.962 |
| Wilt | 0.085 | 0.278 | 0.263 | 0.238 | 0.207 | 0.187 | 0.181 | 0.179 | 0.165 | 0.157 | 0.154 |
| Wine | 0.246 | 0.456 | 0.307 | 0.361 | 0.352 | 0.414 | 0.449 | 0.472 | 0.483 | 0.518 | 0.514 |
| WPBC | 0.376 | 0.399 | 0.396 | 0.395 | 0.390 | 0.405 | 0.406 | 0.400 | 0.405 | 0.408 | 0.412 |
| Yeast | 0.465 | 0.546 | 0.574 | 0.600 | 0.597 | 0.603 | 0.603 | 0.603 | 0.610 | 0.604 | 0.607 |
| Average | 0.572 | 0.634 | 0.631 | 0.629 | 0.628 | 0.629 | 0.631 | 0.632 | 0.628 | 0.628 | 0.631 |

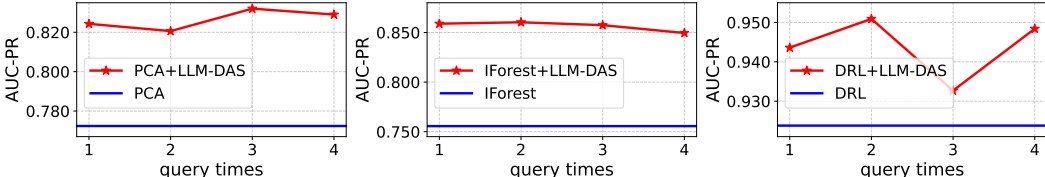

Figure 8: AUC-PR-based performance comparison of different query times on the Imgseg dataset, with results for PCA (left), IForest (middle) and DRL(right).

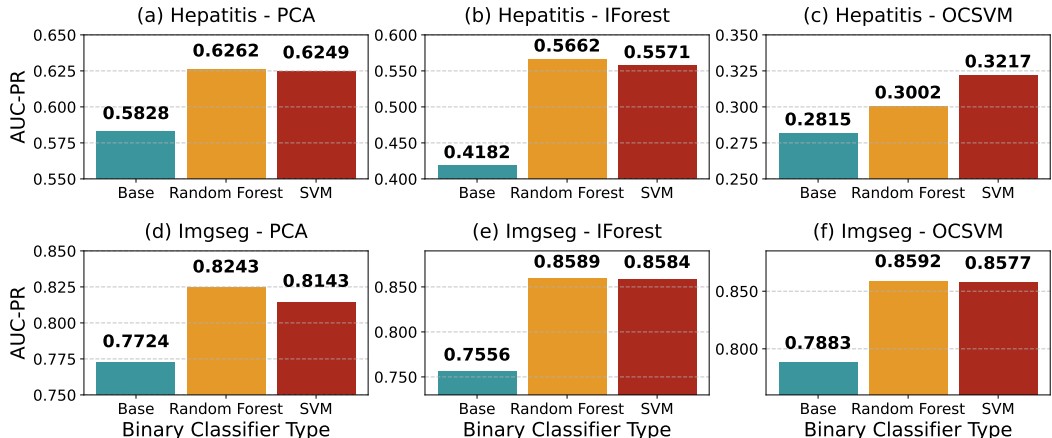

Figure 9: AUC-PR-based performance comparison of different binary classifier choices for LLM-DAS on the Hepatitis and Imgseg datasets, where PCA (left), IForest (middle), and OCSVM (right) serve as the base detectors. "Base" corresponds to the base detector performance. "Random Forest" and "SVM" correspond to LLM-DAS's performance with different binary classifier types.

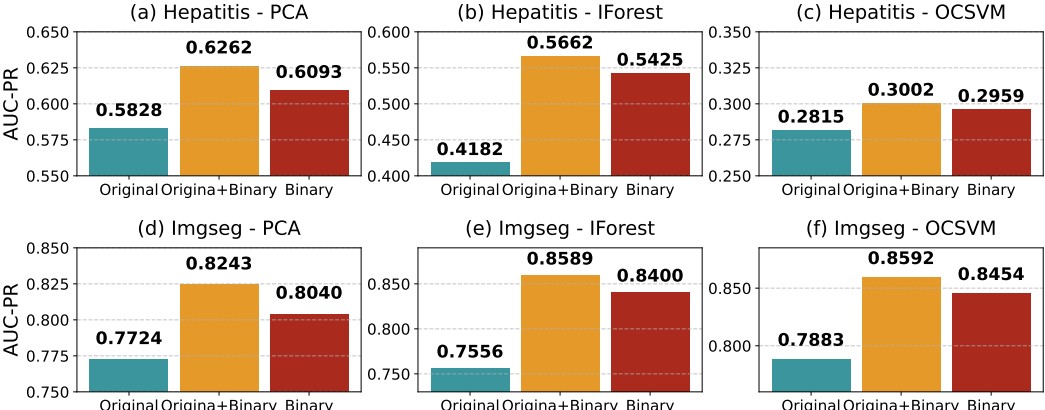

Figure 10: AUC-PR-based performance comparison of ensemble strategies on the Hepatitis and Imgseg datasets, where PCA (left), IForest (middle), and OCSVM (right) serve as the base detectors. "Original" denotes the base detector performance. "Origina+Binary"denotes the performance of LLM-DAS (our proposed method, combining binary classifiers with original detectors), while "Binary" denotes the scheme using binary classifiers directly without integrating original detectors.

### 6.13 ADDITIONAL RESULTS

### 6.13.1 COMPARISON WITH ALTERNATIVE ANOMALY SYNTHESIS METHODS

To further verify the effectiveness of LLM-DAS, we compared several representative anomaly synthesis strategies, including *Gaussian noise injection*, *random outliers*, and *SMOTE-based sample generation*, which are denoted as **Gauss**, **Random**, and **Smote** respectively. The results are in Table 12 to Table 14.

Three Common Anomalies synthesis strategies:

- *Gaussian noise injection* is a method for generating anomalous samples by introducing Gaussian-distributed noise to normal training samples.The process works as follows: first, randomly select a specified number of base samples from the normal training data. Then, generate noise that follows a Gaussian distribution with a predefined mean and standard deviation. This noise is then added to the selected base samples, resulting in new samples that deviate from the original normal data.

- *Random outliers* refer to anomalous samples generated by randomly sampling points within an extended feature space derived from the range of normal training samples. This method first expands the boundaries of each feature dimension beyond the minimum and maximum values of normal data, then randomly distributes the generated anomalies within this extended range, thereby ensuring the anomalies deviate from the distribution of normal samples.

- *SMOTE-based sample generation* draws on the SMOTE algorithm, which is originally used to address data imbalance issues. It can generate boundary anomalous samples near normal training samples.The method first randomly selects base samples from the normal training data and finds k nearest neighbors for each base sample. Then, it calculates the difference between each base sample and a randomly chosen neighbor, and extrapolates along the line connecting them to generate new samples.To create boundary anomalous samples that are closer to or slightly beyond the boundary of the normal data distribution, the interpolation factor is set within the range of 0.5 to 1.5. This ensures that the generated samples either lie near the boundary of the normal data distribution or just outside it, thus effectively serving as boundary anomalous samples.

Table 12: AUC-PR (↑) performance comparison of PCA under different alternative anomaly synthesis methods across 36 datasets.

| | PCA | Gauss | Random | Smote | LLM-DAS |
|---|---|---|---|---|---|
| Abalone | 0.8393 | 0.8307 | 0.8418 | 0.7683 | 0.8411 |
| Amazon | 0.1072 | 0.1079 | 0.1077 | 0.1116 | 0.1077 |
| Annthyroid | 0.5657 | 0.6143 | 0.5667 | 0.2574 | 0.6602 |
| Arrhythmia | 0.5336 | 0.4718 | 0.5406 | 0.5216 | 0.5508 |
| Breastw | 0.9934 | 0.9881 | 0.9939 | 0.9872 | 0.9956 |
| Campaign | 0.4884 | 0.4860 | 0.4845 | 0.3820 | 0.4884 |
| Cardio | 0.8628 | 0.7322 | 0.9025 | 0.7755 | 0.8243 |
| Cardiotocography | 0.6969 | 0.6646 | 0.7145 | 0.5911 | 0.7337 |
| Comm.and.crime | 0.8892 | 0.8628 | 0.8829 | 0.7624 | 0.8806 |
| Fault | 0.6035 | 0.5131 | 0.5451 | 0.5475 | 0.6034 |
| Glass | 0.0896 | 0.0975 | 0.1055 | 0.0937 | 0.1278 |
| Hepatitis | 0.5828 | 0.4790 | 0.4874 | 0.5052 | 0.6262 |
| Imgseg | 0.7724 | 0.7722 | 0.7793 | 0.8002 | 0.8243 |
| Ionosphere | 0.8969 | 0.9480 | 0.9372 | 0.9295 | 0.9260 |
| Lympho | 1.0000 | 0.8056 | 0.9754 | 0.9440 | 0.9889 |
| Mammography | 0.4165 | 0.4184 | 0.4990 | 0.4003 | 0.5793 |
| Mnist | 0.6499 | 0.6584 | 0.6445 | 0.5175 | 0.6499 |
| Musk | 1.0000 | 0.9981 | 0.9990 | 1.0000 | 0.9969 |
| Optdigits | 0.0602 | 0.0601 | 0.0604 | 0.0724 | 0.0602 |
| Parkinson | 0.9297 | 0.9372 | 0.9442 | 0.9300 | 0.9436 |
| Pendigits | 0.3863 | 0.6063 | 0.4464 | 0.3828 | 0.4847 |
| Pima | 0.7008 | 0.6723 | 0.7054 | 0.6732 | 0.7509 |
| Satellite | 0.7778 | 0.8137 | 0.7897 | 0.8035 | 0.7921 |
| Satimage-2 | 0.9192 | 0.9163 | 0.9517 | 0.8954 | 0.9311 |
| Shuttle | 0.9627 | 0.9845 | 0.9891 | 0.9726 | 0.9906 |
| SpamBase | 0.8184 | 0.8230 | 0.8384 | 0.7829 | 0.8228 |
| Speech | 0.0277 | 0.0295 | 0.0250 | 0.0275 | 0.0293 |
| Thyroid | 0.8134 | 0.8486 | 0.8230 | 0.6885 | 0.8146 |
| Vertebral | 0.1381 | 0.1933 | 0.1424 | 0.1771 | 0.1433 |
| Vowels | 0.1051 | 0.1570 | 0.1680 | 0.1958 | 0.1090 |
| Wbc | 0.8391 | 0.7876 | 0.8159 | 0.7477 | 0.8400 |
| WDBC | 0.9833 | 0.8165 | 0.9912 | 0.6980 | 0.9802 |
| Wilt | 0.0641 | 0.1281 | 0.0642 | 0.1334 | 0.0972 |
| Wine | 0.1325 | 0.1859 | 0.1772 | 0.1401 | 0.8459 |
| WPBC | 0.3940 | 0.3522 | 0.3871 | 0.4112 | 0.4085 |
| Yeast | 0.4678 | 0.5049 | 0.4825 | 0.5126 | 0.5053 |
| Average | 0.5975 | 0.5907 | 0.6058 | 0.5594 | **0.6376** |

Table 13: AUC-PR (↑) performance comparison of IForest under different alternative anomaly synthesis methods across 36 datasets.

|  | IForest | Gauss | Random | Smote | LLM-DAS |
|---|---|---|---|---|---|
| Abalone | 0.8481 | 0.8177 | 0.8523 | 0.8056 | 0.8555 |
| Amazon | 0.1091 | 0.1094 | 0.1093 | 0.1097 | 0.1093 |
| Annthyroid | 0.6149 | 0.6642 | 0.6365 | 0.1515 | 0.6200 |
| Arrhythmia | 0.5097 | 0.5044 | 0.5848 | 0.5562 | 0.6021 |
| Breastw | 0.9449 | 0.9817 | 0.9949 | 0.9839 | 0.9947 |
| Campaign | 0.4608 | 0.4396 | 0.4421 | 0.3518 | 0.4517 |
| Cardio | 0.7018 | 0.6447 | 0.8473 | 0.6018 | 0.6220 |
| Cardiotocography | 0.6036 | 0.6355 | 0.6835 | 0.5607 | 0.6382 |
| Comm.and.crime | 0.8940 | 0.8577 | 0.8503 | 0.7641 | 0.8958 |
| Fault | 0.5948 | 0.4997 | 0.5393 | 0.5677 | 0.5774 |
| Glass | 0.0952 | 0.1162 | 0.1495 | 0.1218 | 0.1205 |
| Hepatitis | 0.4182 | 0.5088 | 0.4368 | 0.4816 | 0.5662 |
| Imgseg | 0.7556 | 0.7894 | 0.7987 | 0.8346 | 0.8589 |
| Ionosphere | 0.9768 | 0.9374 | 0.9237 | 0.9006 | 0.8685 |
| Lympho | 0.9593 | 0.6675 | 0.9642 | 0.9503 | 0.9794 |
| Mammography | 0.3334 | 0.3690 | 0.4970 | 0.3024 | 0.5712 |
| Mnist | 0.5349 | 0.6283 | 0.5880 | 0.4381 | 0.6011 |
| Musk | 0.5279 | 0.9168 | 1.0000 | 1.0000 | 1.0000 |
| Optdigits | 0.1570 | 0.1458 | 0.1417 | 0.1555 | 0.1790 |
| Parkinson | 0.9595 | 0.9639 | 0.9583 | 0.9175 | 0.9514 |
| Pendigits | 0.5133 | 0.6070 | 0.5365 | 0.4698 | 0.6020 |
| Pima | 0.6662 | 0.6985 | 0.7074 | 0.6792 | 0.7397 |
| Satellite | 0.8583 | 0.8206 | 0.8281 | 0.8118 | 0.8079 |
| Satimage-2 | 0.8846 | 0.9063 | 0.9495 | 0.9119 | 0.8706 |
| Shuttle | 0.9172 | 0.9870 | 0.9937 | 0.9813 | 0.9954 |
| SpamBase | 0.8902 | 0.8707 | 0.8809 | 0.8227 | 0.8892 |
| Speech | 0.0353 | 0.0331 | 0.0279 | 0.0301 | 0.0318 |
| Thyroid | 0.6055 | 0.7906 | 0.8381 | 0.5110 | 0.8856 |
| Vertebral | 0.1342 | 0.1557 | 0.1354 | 0.2320 | 0.2686 |
| Vowels | 0.0984 | 0.1729 | 0.1436 | 0.1676 | 0.1779 |
| Wbc | 0.8573 | 0.8015 | 0.8271 | 0.7239 | 0.8233 |
| WDBC | 0.9749 | 0.7128 | 0.9625 | 0.6298 | 0.9936 |
| Wilt | 0.0848 | 0.1246 | 0.0871 | 0.1531 | 0.2784 |
| Wine | 0.2458 | 0.3130 | 0.1687 | 0.2132 | 0.4559 |
| WPBC | 0.3760 | 0.3829 | 0.3998 | 0.4076 | 0.3993 |
| Yeast | 0.4654 | 0.4964 | 0.4717 | 0.4933 | 0.5462 |
| Average | 0.5724 | 0.5853 | 0.6099 | 0.5498 | **0.6341** |

Table 14: AUC-PR (↑) performance comparison of DRL under different alternative anomaly synthesis methods across 36 datasets.

| | DRL | Gauss | Random | Smote | LLM-DAS |
|---|---|---|---|---|---|
| Abalone | 0.8850 | 0.8394 | 0.8747 | 0.8202 | 0.9021 |
| Amazon | 0.1206 | 0.0972 | 0.1150 | 0.1021 | 0.1197 |
| Annthyroid | 0.6761 | 0.1731 | 0.6310 | 0.1727 | 0.7618 |
| Arrhythmia | 0.6270 | 0.6273 | 0.6518 | 0.6992 | 0.7298 |
| Breastw | 0.9966 | 0.9857 | 0.9942 | 0.9639 | 0.9972 |
| Campaign | 0.5013 | 0.4854 | 0.4898 | 0.4625 | 0.4945 |
| Cardio | 0.8325 | 0.8480 | 0.9142 | 0.8427 | 0.8861 |
| Cardiotocography | 0.7540 | 0.7323 | 0.7148 | 0.7252 | 0.8046 |
| Comm.and.crime | 0.9164 | 0.9421 | 0.9455 | 0.9386 | 0.9470 |
| Fault | 0.6649 | 0.5765 | 0.6417 | 0.5880 | 0.7053 |
| Glass | 0.1670 | 0.3107 | 0.2948 | 0.2273 | 0.4934 |
| Hepatitis | 0.6627 | 0.6880 | 0.6856 | 0.6870 | 0.6957 |
| Imgseg | 0.9238 | 0.9239 | 0.9300 | 0.9363 | 0.9436 |
| Ionosphere | 0.9895 | 0.9530 | 0.9636 | 0.9539 | 0.9846 |
| Lympho | 1.0000 | 0.6193 | 0.7607 | 0.7348 | 0.8706 |
| Mammography | 0.8406 | 0.3719 | 0.4496 | 0.2873 | 0.5288 |
| Mnist | 0.8870 | 0.9067 | 0.9049 | 0.8693 | 0.9109 |
| Musk | 1.0000 | 0.8481 | 1.0000 | 0.9998 | 1.0000 |
| Optdigits | 0.9356 | 0.2927 | 0.2713 | 0.3133 | 0.9566 |
| Parkinson | 0.9210 | 0.9649 | 0.9629 | 0.9636 | 0.9659 |
| Pendigits | 0.9360 | 0.0688 | 0.2553 | 0.2345 | 0.8848 |
| Pima | 0.7449 | 0.6972 | 0.7100 | 0.7083 | 0.7912 |
| Satellite | 0.8692 | 0.8408 | 0.8857 | 0.7997 | 0.8942 |
| Satimage-2 | 0.9703 | 0.9160 | 0.9666 | 0.9248 | 0.9804 |
| Shuttle | 0.9819 | 0.0884 | 0.9924 | 0.0875 | 0.9991 |
| SpamBase | 0.8413 | 0.6783 | 0.8975 | 0.8278 | 0.9335 |
| Speech | 0.0584 | 0.0516 | 0.0511 | 0.0476 | 0.1189 |
| Thyroid | 0.8626 | 0.2007 | 0.7836 | 0.1660 | 0.8770 |
| Vertebral | 0.2854 | 0.2369 | 0.2592 | 0.2277 | 0.5657 |
| Vowels | 0.4506 | 0.8461 | 0.8216 | 0.8381 | 0.8918 |
| Wbc | 0.9742 | 0.8381 | 0.7973 | 0.7393 | 0.9341 |
| WDBC | 1.0000 | 0.9164 | 0.9363 | 0.8364 | 1.0000 |
| Wilt | 0.4543 | 0.6363 | 0.4872 | 0.3760 | 0.5926 |
| Wine | 1.0000 | 0.9564 | 0.9292 | 0.9423 | 1.0000 |
| WPBC | 0.5017 | 0.4507 | 0.4409 | 0.4720 | 0.5671 |
| Yeast | 0.5416 | 0.5263 | 0.5143 | 0.6136 | 0.5281 |
| Average | 0.7437 | 0.6149 | 0.6923 | 0.6147 | **0.7849** |

### 6.13.2 ABLATION STUDIES ON LLM-DAS DESIGN

To further validate the effectiveness of the key components in LLM-DAS, we conduct an ablation study by systematically removing distinct components of our framework. The results are in Table 16 to Table 18. Specifically, we compare the full LLM-DAS model with its variants that:

- *Remove detector-awareness and use generic prompts* (**Generic**): modify the prompt by removing descriptions of the specific detector's principles, and instruct the LLM to generate a "general-purpose" synthesis code. By comparing its performance with that of LLM-DAS, we can demonstrate that "awareness" the detector's underlying principles and designing targeted anomaly sample generation strategies are crucial.

- *Replace "hard" anomalies with "simple" ones* (**Simple**): modify the prompt such that the LLM generates only "simple" anomalies that are far from the normal cluster. We remove the availability of model.predict_score() for evaluating the anomaly degree of generated samples, thereby eliminating the capability to generate more difficult anomalies based on model.predict_score() values. By comparing the performance of this setup against that of LLM-DAS, we can verify that the "hard" requirement (for generating hard-to-detect anomalies) is critical.

- *Replace borderline synthesis with random selection* (**Random**): generates anomalies by first transforming "randomly selected" normal samples into anomalies through prompt modification, unlike LLM-DAS, which emphasizes selecting or transforming "borderline normal samples" from the training set into anomalies. By comparing the performance of this "randomly selected normal sample-based transformation" method against that of LLM-DAS, we can verify that the "borderline" heuristic is critical.

Table 15: Ablation Studies on the Different Components of LLM-DAS Design

| Variants | Awareness | Hard | Borderline |
|----------|-----------|------|------------|
| Generic | - | ✓ | ✓ |
| Simple | ✓ | - | - |
| Random | ✓ | ✓ | - |
| LLM-DAS | ✓ | ✓ | ✓ |

Table 16: AUC-PR (↑) performance comparison of PCA under different LLM prompt designs across 36 datasets.

| | PCA | Generic | Simple | Random | LLM-DAS |
|---|---|---|---|---|---|
| Abalone | 0.8393 | 0.8290 | 0.8252 | 0.8415 | 0.8411 |
| Amazon | 0.1072 | 0.1079 | 0.1055 | 0.1078 | 0.1077 |
| Annthyroid | 0.5657 | 0.5614 | 0.5329 | 0.5418 | 0.6602 |
| Arrhythmia | 0.5336 | 0.5497 | 0.4569 | 0.5442 | 0.5508 |
| Breastw | 0.9934 | 0.9944 | 0.9916 | 0.9937 | 0.9956 |
| Campaign | 0.4884 | 0.4784 | 0.4894 | 0.4881 | 0.4884 |
| Cardio | 0.8628 | 0.7465 | 0.7576 | 0.7830 | 0.8243 |
| Cardiotocography | 0.6969 | 0.6627 | 0.7612 | 0.6716 | 0.7337 |
| Comm.and.crime | 0.8892 | 0.8909 | 0.8854 | 0.8312 | 0.8806 |
| Fault | 0.6035 | 0.5320 | 0.5045 | 0.5421 | 0.6034 |
| Glass | 0.0896 | 0.0968 | 0.0829 | 0.0999 | 0.1278 |
| Hepatitis | 0.5828 | 0.5799 | 0.5948 | 0.4836 | 0.6262 |
| Imgseg | 0.7724 | 0.8150 | 0.8054 | 0.8124 | 0.8243 |
| Ionosphere | 0.8969 | 0.8291 | 0.9571 | 0.9246 | 0.9260 |
| Lympho | 1.0000 | 0.9549 | 0.8556 | 0.9718 | 0.9889 |
| Mammography | 0.4165 | 0.4808 | 0.3893 | 0.4725 | 0.5793 |
| Mnist | 0.6499 | 0.6423 | 0.6549 | 0.6543 | 0.6499 |
| Musk | 1.0000 | 0.9855 | 0.9981 | 0.9585 | 0.9969 |
| Optdigits | 0.0602 | 0.0677 | 0.0602 | 0.0602 | 0.0602 |
| Parkinson | 0.9297 | 0.9341 | 0.9323 | 0.9343 | 0.9436 |
| Pendigits | 0.3863 | 0.5422 | 0.6302 | 0.4028 | 0.4847 |
| Pima | 0.7008 | 0.7064 | 0.6244 | 0.7034 | 0.7509 |
| Satellite | 0.7778 | 0.7923 | 0.7697 | 0.7886 | 0.7921 |
| Satimage-2 | 0.9192 | 0.8532 | 0.2350 | 0.8704 | 0.9311 |
| Shuttle | 0.9627 | 0.9862 | 0.9791 | 0.9745 | 0.9906 |
| SpamBase | 0.8184 | 0.8237 | 0.8289 | 0.8312 | 0.8228 |
| Speech | 0.0277 | 0.0279 | 0.0311 | 0.0280 | 0.0293 |
| Thyroid | 0.8134 | 0.8244 | 0.7363 | 0.8364 | 0.8146 |
| Vertebral | 0.1381 | 0.1426 | 0.1391 | 0.1426 | 0.1433 |
| Vowels | 0.1051 | 0.0930 | 0.2136 | 0.1203 | 0.1090 |
| Wbc | 0.8391 | 0.7705 | 0.7304 | 0.8249 | 0.8400 |
| WDBC | 0.9833 | 0.8506 | 0.9124 | 0.9226 | 0.9802 |
| Wilt | 0.0641 | 0.0658 | 0.0642 | 0.0641 | 0.0972 |
| Wine | 0.1325 | 0.2685 | 0.1809 | 0.7635 | 0.8459 |
| WPBC | 0.3940 | 0.3872 | 0.4341 | 0.3855 | 0.4085 |
| Yeast | 0.4678 | 0.4723 | 0.4789 | 0.4964 | 0.5053 |
| Average | 0.5975 | 0.5929 | 0.5730 | 0.6076 | **0.6376** |

Table 17: AUC-PR (↑) performance comparison of IForest under different LLM prompt designs across 36 datasets.

|  | IForest | Generic | Simple | Random | LLM-DAS |
|---|---|---|---|---|---|
| Abalone | 0.8481 | 0.8390 | 0.8492 | 0.7452 | 0.8555 |
| Amazon | 0.1091 | 0.1107 | 0.1099 | 0.1102 | 0.1093 |
| Annthyroid | 0.6149 | 0.5911 | 0.6243 | 0.2804 | 0.6200 |
| Arrhythmia | 0.5097 | 0.5794 | 0.5951 | 0.4967 | 0.6021 |
| Breastw | 0.9449 | 0.9966 | 0.9921 | 0.9266 | 0.9947 |
| Campaign | 0.4608 | 0.4332 | 0.4517 | 0.4330 | 0.4517 |
| Cardio | 0.7018 | 0.6611 | 0.7771 | 0.5234 | 0.6220 |
| Cardiotocography | 0.6036 | 0.6570 | 0.6513 | 0.5278 | 0.6382 |
| Comm.and.crime | 0.8940 | 0.8794 | 0.7943 | 0.8661 | 0.8958 |
| Fault | 0.5948 | 0.5347 | 0.5894 | 0.5602 | 0.5774 |
| Glass | 0.0952 | 0.1004 | 0.1063 | 0.0890 | 0.1205 |
| Hepatitis | 0.4182 | 0.5292 | 0.4908 | 0.3799 | 0.5662 |
| Imgseg | 0.7556 | 0.8238 | 0.8268 | 0.6780 | 0.8589 |
| Ionosphere | 0.9768 | 0.7474 | 0.9119 | 0.7488 | 0.8685 |
| Lympho | 0.9593 | 0.9577 | 0.9213 | 0.8498 | 0.9794 |
| Mammography | 0.3334 | 0.4366 | 0.4321 | 0.1642 | 0.5712 |
| Mnist | 0.5349 | 0.5728 | 0.6154 | 0.6294 | 0.6011 |
| Musk | 0.5279 | 0.8382 | 1.0000 | 0.8611 | 1.0000 |
| Optdigits | 0.1570 | 0.1339 | 0.1579 | 0.1423 | 0.1790 |
| Parkinson | 0.9595 | 0.9615 | 0.9557 | 0.9156 | 0.9514 |
| Pendigits | 0.5133 | 0.6743 | 0.5427 | 0.5022 | 0.6020 |
| Pima | 0.6662 | 0.7244 | 0.7110 | 0.6084 | 0.7397 |
| Satellite | 0.8583 | 0.8203 | 0.8217 | 0.8158 | 0.8079 |
| Satimage-2 | 0.8846 | 0.5511 | 0.9515 | 0.1444 | 0.8706 |
| Shuttle | 0.9172 | 0.9956 | 0.9912 | 0.9978 | 0.9954 |
| SpamBase | 0.8902 | 0.8832 | 0.8175 | 0.8648 | 0.8892 |
| Speech | 0.0353 | 0.0267 | 0.0296 | 0.0351 | 0.0318 |
| Thyroid | 0.6055 | 0.7622 | 0.8280 | 0.1924 | 0.8856 |
| Vertebral | 0.1342 | 0.1385 | 0.1339 | 0.1814 | 0.2686 |
| Vowels | 0.0984 | 0.1127 | 0.1054 | 0.0940 | 0.1779 |
| Wbc | 0.8573 | 0.7837 | 0.8531 | 0.2100 | 0.8233 |
| WDBC | 0.9749 | 0.8893 | 0.9687 | 0.1164 | 0.9936 |
| Wilt | 0.0848 | 0.0955 | 0.0871 | 0.1913 | 0.2784 |
| Wine | 0.2458 | 0.1551 | 0.1886 | 0.5410 | 0.4559 |
| WPBC | 0.3760 | 0.3752 | 0.3990 | 0.4400 | 0.3993 |
| Yeast | 0.4654 | 0.4798 | 0.4668 | 0.4708 | 0.5462 |
| Average | 0.5724 | 0.5792 | 0.6041 | 0.4815 | **0.6341** |

Table 18: AUC-PR (↑) performance comparison of DRL under different LLM prompt designs across 36 datasets.

|  | DRL | Generic | Simple | Random | LLM-DAS |
|---|---|---|---|---|---|
| Abalone | 0.8850 | 0.8813 | 0.8716 | 0.8625 | 0.9021 |
| Amazon | 0.1206 | 0.1023 | 0.1011 | 0.0958 | 0.1197 |
| Annthyroid | 0.6761 | 0.3477 | 0.5409 | 0.3578 | 0.7618 |
| Arrhythmia | 0.6270 | 0.6657 | 0.6138 | 0.6380 | 0.7298 |
| Breastw | 0.9966 | 0.9954 | 0.9934 | 0.9879 | 0.9972 |
| Campaign | 0.5013 | 0.4957 | 0.4857 | 0.4852 | 0.4945 |
| Cardio | 0.8325 | 0.8978 | 0.9116 | 0.8597 | 0.8861 |
| Cardiotocography | 0.7540 | 0.6607 | 0.7109 | 0.7084 | 0.8046 |
| Comm.and.crime | 0.9164 | 0.9460 | 0.9442 | 0.9438 | 0.9470 |
| Fault | 0.6649 | 0.5967 | 0.6044 | 0.6537 | 0.7053 |
| Glass | 0.1670 | 0.3493 | 0.3166 | 0.3036 | 0.4934 |
| Hepatitis | 0.6627 | 0.6876 | 0.6880 | 0.6808 | 0.6957 |
| Imgseg | 0.9238 | 0.9301 | 0.9308 | 0.9283 | 0.9436 |
| Ionosphere | 0.9895 | 0.9486 | 0.9770 | 0.9514 | 0.9846 |
| Lympho | 1.0000 | 0.7374 | 0.7339 | 0.7490 | 0.8706 |
| Mammography | 0.8406 | 0.4620 | 0.4819 | 0.3158 | 0.5288 |
| Mnist | 0.8870 | 0.9165 | 0.9083 | 0.9106 | 0.9109 |
| Musk | 1.0000 | 0.9925 | 1.0000 | 1.0000 | 1.0000 |
| Optdigits | 0.9356 | 0.3192 | 0.2817 | 0.2909 | 0.9566 |
| Parkinson | 0.9210 | 0.9643 | 0.9628 | 0.9659 | 0.9659 |
| Pendigits | 0.9360 | 0.0538 | 0.3199 | 0.0467 | 0.8848 |
| Pima | 0.7449 | 0.7499 | 0.6457 | 0.6541 | 0.7912 |
| Satellite | 0.8692 | 0.8659 | 0.8843 | 0.8770 | 0.8942 |
| Satimage-2 | 0.9703 | 0.9384 | 0.9641 | 0.9353 | 0.9804 |
| Shuttle | 0.9819 | 0.1962 | 0.9934 | 0.8961 | 0.9991 |
| SpamBase | 0.8413 | 0.8272 | 0.7837 | 0.7212 | 0.9335 |
| Speech | 0.0584 | 0.0478 | 0.0502 | 0.0494 | 0.1189 |
| Thyroid | 0.8626 | 0.2966 | 0.6301 | 0.3089 | 0.8770 |
| Vertebral | 0.2854 | 0.2329 | 0.2564 | 0.2389 | 0.5657 |
| Vowels | 0.4506 | 0.8592 | 0.8147 | 0.8985 | 0.8918 |
| Wbc | 0.9742 | 0.7369 | 0.8001 | 0.7317 | 0.9341 |
| WDBC | 1.0000 | 0.8423 | 0.9250 | 0.7377 | 1.0000 |
| Wilt | 0.4543 | 0.6526 | 0.4637 | 0.6250 | 0.5926 |
| Wine | 1.0000 | 0.9591 | 0.9577 | 0.9635 | 1.0000 |
| WPBC | 0.5017 | 0.4675 | 0.4550 | 0.4361 | 0.5671 |
| Yeast | 0.5416 | 0.5457 | 0.5195 | 0.5373 | 0.5281 |
| Average | 0.7437 | 0.6436 | 0.6812 | 0.6485 | **0.7849** |

### 6.14 A Simple Theoretical Illustration of Boundary Refinement

This appendix provides a stylized theoretical perspective on why detector-aware synthetic hard anomalies can improve the detection of true hard anomalies. While the main paper focuses on empirical validation across five detector families, a lightweight theoretical model helps clarify the geometric mechanism underlying the improvements observed in Fig. 5 and Tables 1, 4–5.

#### 6.14.1 Motivation and Theoretical Abstraction

**Motivation.** Empirically, we observe that most detection errors arise from *hard anomalies*— true anomalies that lie close to the decision boundary of the base detector $f_t$. In contrast, *easy anomalies* that lie far from this boundary are already well-detected by $f_t$ without augmentation. LLM-DAS is explicitly designed to target this near-boundary region: it identifies borderline normal samples and transforms them into synthetic hard anomalies that reflect the failure modes of $f_t$. These synthetic anomalies provide precisely the type of near-boundary supervision that one-class detectors inherently lack. The enhancement classifier $\tilde{f}_t$ is trained on these hard synthetic samples, and the final fused detector

$$F_t(x) = \mathrm{Norm}(f_t(x)) + \mathrm{Norm}(\tilde{f}_t(x))$$

combines the strengths of both models.

**Theoretical abstraction.** To formalize this mechanism, we adopt the following stylized model:

- The set of true anomalies is conceptually decomposed into *easy anomalies* (far from the decision boundary of $f_t$) and *hard anomalies* (near the boundary, where most errors occur).

- LLM-DAS generates *detector-aware synthetic hard anomalies* by transforming normal samples that lie near the decision boundary of $f_t$.

- If these synthetic anomalies geometrically cover the true hard anomalies within a small neighborhood, then any margin learned by the enhancement classifier $\tilde{f}$ on the synthetic anomalies should transfer to the nearby true hard anomalies.

This abstraction leads to the proposition below, which formalizes this margin-transfer intuition under a mild smoothness assumption on $\tilde{f}$.

#### 6.14.2 Margin Transfer from Synthetic Hard Anomalies

**Proposition 1 (Margin transfer via detector-aware synthetic anomalies)** *Let $\mathcal{X} \subset \mathbb{R}^d$ be the input space and $\tau \in \mathbb{R}$ a decision threshold. Let $\tilde{f} : \mathcal{X} \to \mathbb{R}$ be a B-Lipschitz scoring function:*

$$|\tilde{f}(x) - \tilde{f}(x')| \leq B\|x - x'\|_2 \quad \forall x, x'.$$

*Suppose a set of detector-aware synthetic anomalies $S_{\mathrm{syn}} = \{x_{\mathrm{syn}}\}$ satisfies*

$$\tilde{f}(x_{\mathrm{syn}}) \geq \tau + \gamma, \qquad \gamma > 0.$$

*Assume the set of true hard anomalies $\mathcal{D}_A$ is $\varepsilon$-covered by $S_{\mathrm{syn}}$, i.e., for each $x_a \in \mathcal{D}_A$ there exists $x_{\mathrm{syn}} \in S_{\mathrm{syn}}$ with $\|x_a - x_{\mathrm{syn}}\|_2 \leq \varepsilon$. Then each true hard anomaly satisfies*

$$\tilde{f}(x_a) \geq \tau + \gamma - B\varepsilon.$$

*In particular, if $B\varepsilon < \gamma$, then $\tilde{f}(x_a) > \tau$, and $x_a$ is correctly detected under threshold $\tau$.*

*Proof.* Fix any $x_a \in \mathcal{D}_A$. By $\varepsilon$-covering, choose $x_{\mathrm{syn}} \in S_{\mathrm{syn}}$ with $\|x_a - x_{\mathrm{syn}}\|_2 \leq \varepsilon$. By Lipschitzness,

$$|\tilde{f}(x_a) - \tilde{f}(x_{\mathrm{syn}})| \leq B\varepsilon.$$

Thus,

$$\tilde{f}(x_a) \geq \tilde{f}(x_{\mathrm{syn}}) - B\varepsilon \geq (\tau + \gamma) - B\varepsilon.$$

If $B\varepsilon < \gamma$, then $\tilde{f}(x_a) > \tau$, completing the proof.

**Interpretation.** The proposition states that if synthetic hard anomalies are close (in geometry) to the true hard anomalies ($\downarrow \varepsilon$), and if the enhancement classifier can enforce a margin ($\uparrow \gamma$) on the synthetic samples, then the true hard anomalies inherit a positive margin up to a Lipschitz slack ($B\varepsilon < \gamma$). This formalizes how near-boundary synthetic anomalies help the enhanced classifier learn a tighter decision boundary exactly where the base detector is weak.

### 6.14.3 REMARKS ON APPLICABILITY AND MODEL CLASSES

**Smooth vs. non-smooth enhancement classifiers.** The margin-transfer proposition relies on a Lipschitz regularity condition on $\tilde{f}$, which holds for smooth discriminative models such as linear or kernel SVMs. As shown in Fig. 9 of Appendix 6.12, LLM-DAS yields clear performance improvements when $\tilde{f}$ is instantiated as an SVM, consistent with the proposition.

Although tree ensembles are not Lipschitz in the Euclidean sense, they follow similar qualitative geometry: synthetic anomalies near the boundary tend to fall into similar leaves or partitions as true hard anomalies, providing comparable refinement. Extending the formal analysis to non-smooth models is an interesting direction for future work.

### 6.14.4 CONVERGENCE AND STABILITY OF THE FUSED DETECTOR

The final fused detector is

$$F_t(x) = \text{Norm}(f_t(x)) + \text{Norm}(\tilde{f}_t(x)).$$

Both components are bounded and monotone, so the enhancement step preserves the stability and asymptotic behavior of the base detector $f_t$, while adding a bounded corrective term concentrated around the decision boundary. Thus the enhancement procedure behaves as a stable additive refinement: it reduces bias near hard anomalies while introducing only controlled variance determined by the number and quality of the synthetic samples and the complexity of $\tilde{f}_t$.

### 6.14.5 JUSTIFICATION FOR LOCAL BOUNDARY REFINEMENT MECHANISM IN LLM-DAS

A complex, global decision boundary may not be uniformly "tightened". The effectiveness of LLM-DAS does not rely on such global coverage but on a targeted, local refinement mechanism. Our conceptual justification is twofold and is strongly supported by our anomaly-type analysis.

**(1) Local Patching of Algorithmic Weak Spots:** The LLM in LLM-DAS acts as an automatic analyst that identifies a detector's structural weaknesses from its algorithm description. It then generates a program to create anomalies that specifically target these weaknesses. The goal is not to resample the entire boundary but to patch critical, high-error regions where the base detector $f_t$ is known to fail. The performance improvement stems from precisely this targeted effort.

**(2) Empirical Evidence via Anomaly-Type Alignment (Connecting to Appendix 6.15):** Our experiments in Appendix 6.15 (Table 19, Fig. 11-13) provide direct, empirical validation of this local-patching principle. They demonstrate that LLM-DAS automatically synthesizes anomalies whose distribution is closest to the specific anomaly types (e.g., dependency and local anomalies) that each base detector (e.g., IForest, OCSVM) struggles with most, as per established benchmarks like ADBench.

- This is crucial: Different anomaly types challenge different segments of the decision boundary. By generating anomalies that align with a detector's known algorithmic blind spots, LLM-DAS proves it can automatically locate and target the most impactful "weak regions" of the boundary, even if they constitute only a portion of the whole.
- The significant performance gains we observe on these specific anomaly types confirm that patching these localized regions is both necessary and sufficient for overall improvement.

**(3) Preservation of Existing Strengths via Ensembling:** Furthermore, our framework is designed to be robust and non-destructive. As noted in our analysis, for anomaly types where the base detector already performs well (e.g., IForest on Cluster anomalies in Table 19), LLM-DAS maintains stable or slightly improved results. This indicates that our method enhances robustness without overfitting to the particular anomaly categories it uses for augmentation. This is a direct benefit of our ensemble

design $F(x) = \text{Norm}(f_t(x)) + \text{Norm}(\tilde{f}_t(x))$. The original detector $f_t$ continues to dominate in regions it already models accurately, preserving its discriminative ability for "easy" anomalies, while the enhancement classifier $\tilde{f}_t$ extends the coverage toward harder cases targeted by the LLM.

Therefore, while a single set of synthetic anomalies may not cover the entire boundary, its coverage is highly strategic. It focuses on the regions that matter most for reducing the base detector's error, which is why the method delivers consistent and significant gains beyond purely empirical observations.

### 6.14.6 SUMMARY

Together with Fig. 5, this stylized analysis provides a formal illustration of the empirical finding that LLM-DAS tightens the decision boundary around hard anomalies. The proposition explains why detector-aware synthetic anomalies bring improved separation precisely in the near-boundary region, while the ensemble structure ensures that easy anomalies remain correctly handled by the original detector.

### 6.15 DETECTOR-AWARE ANOMALY SYNTHESIS ACROSS DIFFERENT ANOMALY TYPES

#### 6.15.1 MOTIVATION AND EXPERIMENTAL PHILOSOPHY.

A core challenge in tabular anomaly detection (TAD) is the diversity of anomaly types. The ground-truth anomaly type is not standardized across datasets, and different datasets may correspond to different or mixed anomaly mechanisms, making real-world anomalies inherently complex and heterogeneous. Different detectors inherently excel at detecting different kinds of anomalies due to their underlying algorithms' assumptions. According to ADBench (Han et al., 2022), many tabular detectors—such as IForest, OCSVM—struggle to detect "Dependency" and "Local" anomalies, while performing well on "Clustered" types.

The key innovation of LLM-DAS is that it does not require a priori specification of anomaly types. Instead, it positions the LLM as an "algorithmist" that reasons about a detector's intrinsic logic to generate code $\text{Code}^t$ for synthesizing anomalies that are specifically "hard" for that detector. A critical question thus arises: What types of anomalies does this detector-aware code actually produce?

To answer this scientifically, we need a controlled experimental setup where ground-truth anomaly types are known. We, therefore, adopt the established synthesis protocol from ADBench (Han et al., 2022) to create a benchmark where different, pure anomaly types are injected into the same normal data backbone. This allows us to rigorously analyze the relationship between a detector's weaknesses and the characteristics of the anomalies synthesized by its corresponding $\text{Code}^t$.

#### 6.15.2 EXPERIMENTAL SETUP: CONSTRUCTING TYPE-SPECIFIC TESTBEDS

To analyze how LLM-DAS adapts to different anomaly mechanisms in a controlled and detector-aware manner, we follow the official ADBench (Han et al., 2022) protocol to construct four benchmark variants of the same real-world dataset ("Hepatitis"). The core idea is to build a generative model (e.g., Gaussian mixture model GMM) using the normal samples from the real-world dataset ("Hepatitis") and discard its original anomalies as we do not know their types. Then, We could generate normal samples and different types of anomalies based on their definitions by tweaking the generative model. The generation of normal samples is the same in all settings if not noted, and we provide the generation process of four types of anomalies below.

**Unified anomaly-generation protocol.** Each anomaly type is created using the following standardized procedures from ADBench (Han et al., 2022):

- **Local anomalies:** Generated by scaling the covariance matrix of Gaussian mixture components ($\hat{\Sigma} \leftarrow \alpha\hat{\Sigma}$ with $\alpha = 5$), producing locally deviant samples within dense regions of the normal manifold.

- **Global anomalies:** They are more different from the normal data, drawn from a uniform distribution $\text{Unif}(\alpha \min(X_k), \alpha \max(X_k))$ with $\alpha = 1.1$, where the boundaries are defined as the min and max of an input feature, e.g., $k$-th feature $X_k$.

- **Dependency anomalies:** Produced by removing feature dependencies modeled by Vine Copula, where the joint density is replaced by independent marginals, breaking the correlation structure among features.

- **Clustered anomalies:** Formed by scaling the mean vector of the normal data ($\hat{\mu} \leftarrow \alpha\hat{\mu}$ with $\alpha = 5$), creating compact abnormal clusters separated from the normal distribution.

For each anomaly type $\tau \in \{\text{Local}, \text{Global}, \text{Dependency}, \text{Clustered}\}$, we obtain a dataset consisting of: $\{D_{\text{train}}^{(\tau)}$ (normal only), $D_{\text{test}}^{(\tau)}$ (normal + anomalies of type $\tau$)$\}$. Importantly, the synthesized anomalies produced by LLM-DAS and normal samples are kept the same across all variants, ensuring that the only varying factor is the anomaly type in the test set.

#### 6.15.3 ANALYSIS PROCEDURE FOR LLM-DAS

Given a detector $t$ (e.g., IForest), we examine how the LLM-generated synthesis code $\text{Code}^t$ adapts to different anomaly types:

1. **Train the base detector.** Train $f_t$ on the normal-only dataset $D_{\text{train}}^{(\tau)}$.

2. **Generate detector-specific synthetic anomalies.** We instantiate the LLM-generated, detector-specific synthesis code $\text{Code}^t$ using $D_{\text{train}}^{(\tau)}$ and $f_t$. This produces a set of synthetic anomalies $A(D_{\text{train}}^{(\tau)}, t)$:

3. **Measure distributional similarity. (Table 19)** Compute the Wasserstein distance between $A(D_{\text{train}}^{(\tau)}, t)$ and each of the four ground-truth anomaly-type distributions in $D_{\text{test}}^{(\tau)}$.

4. **Evaluate performance improvement. (Table 19)** We use $A(D_{\text{train}}^{(\tau)}, t)$ to augment $D_{\text{train}}^{(\tau)}$ and train a binary classifier $\tilde{f}_t$. We then fuse the $\tilde{f}_t$ and $f_t$ to obtain the final detector $F_t$, and evaluate performance on $D_{\text{test}}^{(\tau)}$ to obtain the improvement $\Delta$.

5. **Kernel Density Estimation (KDE). (Fig. 11–13)** We visualize the kernel density estimation (KDE) of anomaly scores before and after applying LLM-DAS, illustrating how the synthesized anomalies reshape the decision boundary.

Together, these analyses enable a detailed examination of *(i)* what types of anomalies LLM-DAS tends to generate, and *(ii)* how such generated samples influence the detector's behavior and decision-space separation under different anomaly mechanisms.

### 6.15.4 ORIGINAL DETECTORS' PRIORI ON DIFFERENT TYPES OF ANOMALIES.

According to ADBench (Han et al., 2022), many tabular detectors—such as IForest, OCSVM—struggle to detect "Dependency" and "Local" anomalies, while performing well on "Clustered" types. To further broaden our observations, we also evaluate LLM-DAS paired with LOF, since LOF shows the opposite pattern: it performs better on "Dependency" and "Local" anomalies but poorly on "Clustered" anomalies. Although LOF was not part of our main benchmark suite, it serves as a valuable complementary case for examining whether LLM-DAS adaptively targets detector's weakest anomaly type.

### 6.15.5 KEY FINDINGS: LLM-DAS AUTOMATICALLY TARGETS THE ANOMALY TYPES THAT EACH DETECTOR PERFORMS WORST ON

Table 19 shows that the base detectors exhibit distinct weaknesses across different anomaly types: IForest and OCSVM perform poorly on *Dependency*- and *Local*-type anomalies, while LOF struggles on *Cluster*-type anomalies. These findings are consistent with the conclusions of ADBench (Han et al., 2022), confirming that each detector has its own characteristic sensitivity to specific anomaly structures.

As shown in Table 19, the anomalies synthesized by LLM-DAS are distributionally closer to the anomaly types on which the corresponding detectors underperform— *Dependency* and *Local* for IForest and OCSVM, and *Cluster* for LOF. This observation aligns with our core claim that LLM-DAS is not generating arbitrary or predefined anomaly types, but instead dynamically adapts its synthesis behavior based on the weaknesses of the base detector. In other words, LLM-DAS learns to identify and reproduce the challenging regions within each detector's decision space rather than following a fixed anomaly taxonomy.

Furthermore, combining the quantitative and visual evidence from Table 19 and Fig. 11–13, we observe that the anomalies synthesized by LLM-DAS significantly improve performance and enhance score separation between normal and abnormal samples on those hard anomaly types where the base detectors originally perform poorly. This confirms that LLM-DAS effectively addresses the intrinsic blind spots of each detector. Meanwhile, for anomaly types that the base detectors already handle well, LLM-DAS maintains stable or slightly improved performance, demonstrating its robustness and balanced ensemble design.

Overall, these results validate that LLM-DAS adaptively captures the detector-specific difficulty patterns across heterogeneous anomaly types, thereby inherently accounting for anomaly diversity rather than ignoring it.

**Discussion and Future Work.** These findings highlight that LLM-DAS does not generate arbitrary anomalies, but adaptively learns to emphasize those challenging for the base detector. Meanwhile,

Table 19: Detector-aware anomaly synthesis: Wasserstein distance versus performance improvement. For each detector and each canonical anomaly type (*Local*, *Global*, *Dependency*, *Clustered*), we report: (i) the Wasserstein distance between the LLM-DAS–generated anomalies and the corresponding ground-truth anomaly-type distribution, (ii) the base detector performance ($f_t$), (iii) the enhanced performance after applying LLM-DAS ($F_t$), and (iv) the improvement $\Delta$ of $F_t$ over $f_t$. Across detectors, the anomaly types with the *smallest* Wasserstein distances (highlighted in blue) consistently correspond to the *largest* performance gains, demonstrating that LLM-DAS automatically synthesizes anomalies that resemble each detector's weakest anomaly mechanism.

| Detector type | Anomaly type | Wasserstein distance ↓ | Base detector ($f_t$) ↑ | LLM-DAS ($F_t$) ↑ | Improv. ($\Delta$) ↑ |
|---|---|---|---|---|---|
| IForest | Dependency | **0.942** | 0.419 | 0.701 | **0.282** |
| | Local | **0.958** | 0.575 | 0.717 | **0.142** |
| | Cluster | 1.694 | 0.781 | 0.797 | 0.016 |
| | Global | 1.824 | 0.701 | 0.722 | 0.021 |
| OCSVM | Dependency | **0.651** | 0.246 | 0.727 | **0.481** |
| | Local | **0.681** | 0.560 | 0.696 | **0.136** |
| | Cluster | 1.630 | 0.789 | 0.803 | 0.014 |
| | Global | 1.779 | 0.719 | 0.722 | 0.002 |
| LOF | Dependency | 1.117 | 0.777 | 0.791 | 0.014 |
| | Local | 1.877 | 0.737 | 0.761 | 0.024 |
| | Cluster | **0.897** | 0.141 | 0.757 | **0.616** |
| | Global | 1.723 | 0.647 | 0.676 | 0.029 |

our ensemble strategy ensures that the original detector $f_t$ continues to dominate detection of easy anomaly types it already handles well, preserving generalization (see Appendix 6.14). In future work, we plan to design a controllable synthesis module that allows users to explicitly specify target anomaly types (e.g., *dependency* or *clustered*) to further enhance interpretability and diagnostic use.

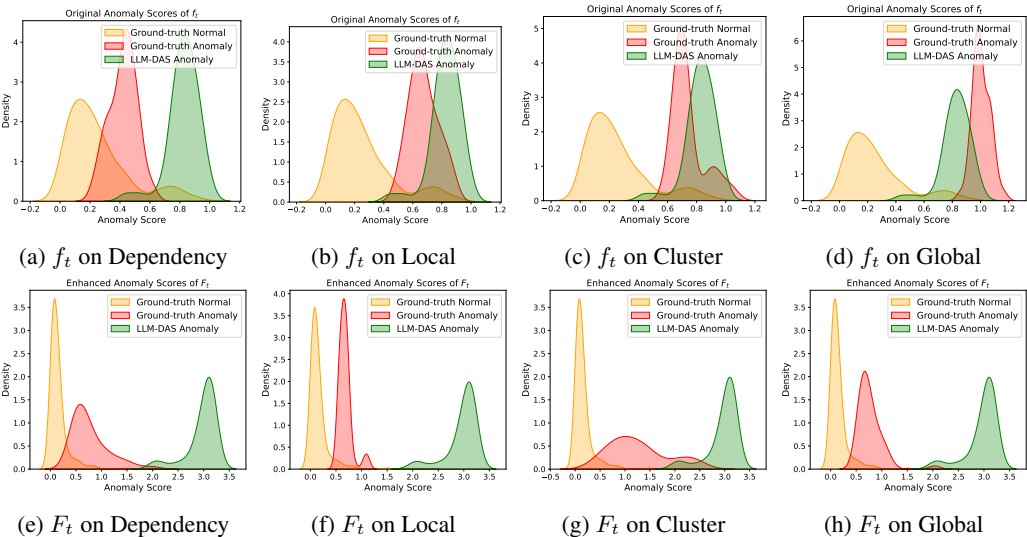

(a) $f_t$ on Dependency     (b) $f_t$ on Local     (c) $f_t$ on Cluster     (d) $f_t$ on Global

(e) $F_t$ on Dependency     (f) $F_t$ on Local     (g) $F_t$ on Cluster     (h) $F_t$ on Global

Figure 11: Kernel density estimation (KDE) of anomaly scores from the source IForest detector $f_t$ and the LLM-DAS–enhanced detector $F_t$. The first row (subplots a–d) shows the original detector $f_t$, and the second row (e–h) shows the corresponding results after applying LLM-DAS. Each column corresponds to one anomaly type (*Dependency*, *Local*, *Cluster*, *Global*). The enhanced detector $F_t$ exhibits improved boundary separation, especially for the ***Dependency*** and ***Local*** types, whose characteristics are most similar to the synthesized anomalies.

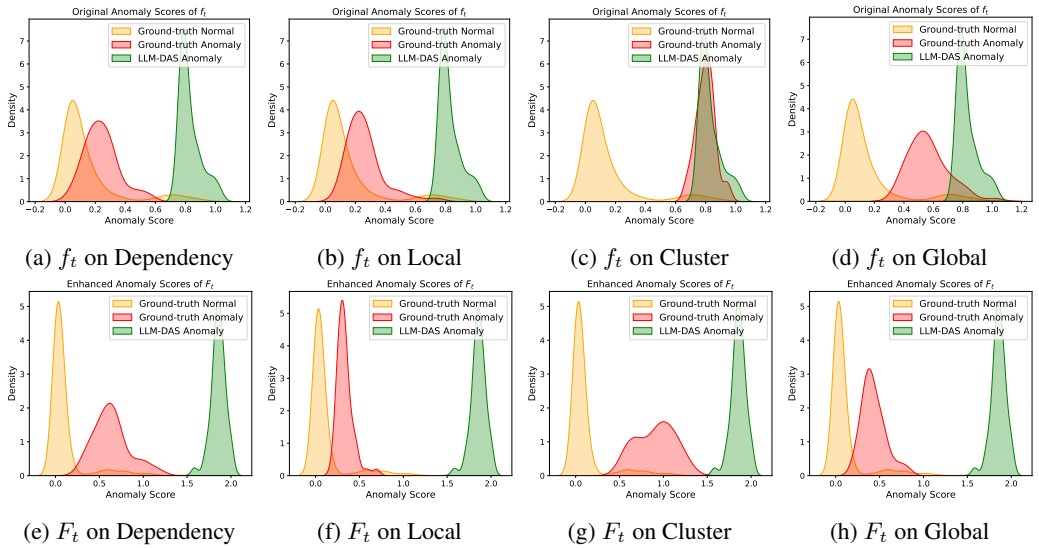

Figure 12: Kernel density estimation (KDE) of anomaly scores from the source OCSVM detector $f_t$ and the LLM-DAS–enhanced detector $F_t$. The first row (subplots a–d) shows the original detector $f_t$, and the second row (e–h) shows the corresponding results after applying LLM-DAS. Each column corresponds to one anomaly type (*Dependency*, *Local*, *Cluster*, *Global*). The enhanced detector $F_t$ exhibits improved boundary separation, especially for the ***Dependency*** **and** ***Local*** types, whose characteristics are most similar to the synthesized anomalies.

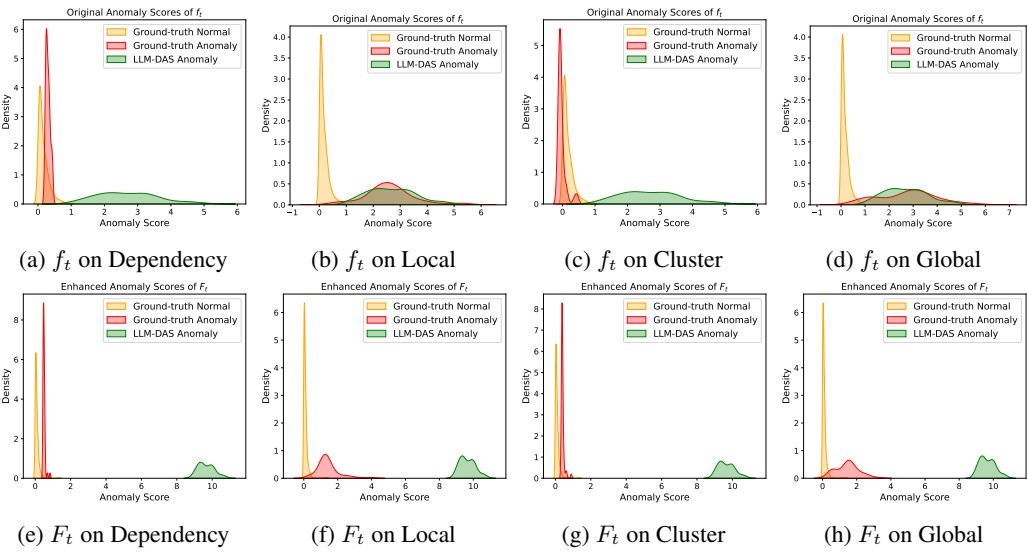

Figure 13: Kernel density estimation (KDE) of anomaly scores from the source LOF detector $f_t$ and the LLM-DAS–enhanced detector $F_t$. The first row (subplots a–d) shows the original detector $f_t$, and the second row (e–h) shows the corresponding results after applying LLM-DAS. Each column corresponds to one anomaly type (*Dependency*, *Local*, *Cluster*, *Global*). The enhanced detector $F_t$ exhibits improved boundary separation, especially for the ***Cluster*** types, whose characteristics are most similar to the synthesized anomalies.

## 6.16 Hardness-Controlled Synthetic Anomalies and Side-Effects Analysis

**Motivation.**  An important question is (i) whether LLM-DAS can generate anomalies with *different levels of hardness*, and (ii) whether overly hard synthetic anomalies may negatively impact generalization in some datasets. To systematically study these aspects, we extend the synthesis prompt with an explicit *hardness-level interface* (prompt is detailed in Prompt 6.16) and introduce a quantitative metric for anomaly hardness based on Wasserstein distances. This appendix provides the experimental protocol and detailed results that support our claim.

**Hardness-level interface and experimental setup.**  We modify the original LLM-DAS prompt so that the LLM is required to produce synthesis code that accepts a discrete hardness level `"High"`, `"Medium"`, or `"Low"`. Intuitively:

- **High** aims to generate anomalies that are close to the normal-data manifold and are therefore *hard to detect*;
- **Medium** generates anomalies moderately separated from normal data;
- **Low** generates anomalies further away, which are *easier* to detect.

The original LLM-DAS implementation in the main paper corresponds closely to the **High-hardness** behavior, i.e., its synthesized anomalies typically lie near the decision boundary and are thus challenging for the base detector.

We instantiate this interface on two representative base detectors: PCA and IForest. For each dataset, we compare:

- the base detector alone (PCA / IForest),
- the original LLM-DAS (denoted "LLM-DAS (Ours)"),
- LLM-DAS with hardness levels High / Medium / Low (denoted "LLM-DAS (Hardness Level)").

**Performance effect of hardness control.**  Table 20 reports the performance of PCA and IForest-based LLM-DAS under different hardness settings. Rows highlighted in blue correspond to datasets where the original LLM-DAS already improves over the base detector. Rows highlighted in red correspond to *fail cases* where the original LLM-DAS underperforms the base detector.

Several consistent patterns emerge:

- **Original $\approx$ High hardness.** For most datasets, the original LLM-DAS and the High-hardness variant achieve very similar performance. This confirms that the default policy learned by LLM-DAS indeed focuses on generating *hard* anomalies.
- **Robust gains across hardness levels.** On the majority of datasets (blue rows), all hardness levels (High/Medium/Low) outperform the base detector, and the original (High) version is often the best or close to the best. This shows that LLM-DAS is not fragile with respect to the exact hardness level, as long as synthetic anomalies inhabit a reasonably informative region near the boundary.
- **Failure cases and recovery by lowering hardness.** In a few datasets (red rows), the original LLM-DAS underperforms the base detector. Crucially, these failures are *reversible*: by lowering the hardness level (Medium or Low), performance recovers to the baseline or even exceeds it. For example, for PCA on lympho, the original LLM-DAS yields $0.989$, while Medium/Low hardness both achieve $1.000$; for IForest on cardio, the original LLM-DAS yields $0.622$, while Low hardness reaches $0.718$.

These results empirically validate that (i) LLM-DAS can generate anomalies at different difficulty levels, and (ii) the hardness control interface provides a simple and effective knob to mitigate rare side effects.

**Quantifying hardness via Wasserstein distance.**  To make hardness evaluation model-agnostic, we compute the multivariate Wasserstein distance between the synthetic anomalies and the normal

Table 20: AUC-PR performance of LLM-DAS with different hardness levels of anomaly synthesis. We extend the synthesis prompt (detailed in Prompt 6.16) so that the LLM-generated code includes a discrete hardness parameter (`High`, `Medium`, `Low`). Rows in blue indicate datasets where the original LLM-DAS improves over the base detector. Rows in red indicate cases where the original LLM-DAS underperforms the base detector, but performance can be recovered or improved by lowering the hardness.

| Datasets | Base detector PCA | +LLM-DAS (Ours) | +LLM-DAS (Hardness Level) | | |
| --- | --- | --- | --- | --- | --- |
| | | | High | Medium | Low |
| imgseg | 0.772 | **0.824** | 0.820 | 0.796 | 0.787 |
| lympho | **1.000** | 0.989 | 0.987 | **1.000** | **1.000** |
| satellite | 0.778 | 0.792 | **0.799** | 0.796 | 0.790 |
| satimage-2 | 0.919 | 0.931 | **0.933** | **0.933** | 0.927 |
| vertebral | 0.138 | 0.143 | **0.144** | 0.143 | 0.141 |
| WPBC | 0.394 | 0.408 | **0.411** | 0.402 | 0.401 |
| musk | 1.000 | 0.997 | 0.996 | **1.000** | **1.000** |
| cadio | 0.863 | 0.824 | 0.826 | 0.847 | **0.864** |
| WDBC | 0.983 | 0.980 | 0.981 | 0.983 | **0.989** |
| mammography | 0.417 | **0.579** | 0.575 | 0.563 | 0.561 |
| pima | 0.701 | **0.751** | 0.749 | 0.740 | 0.734 |

| Datasets | Base detector IForest | +LLM-DAS | +LLM-DAS (Hardness Level) | | |
| --- | --- | --- | --- | --- | --- |
| | | | High | Medium | Low |
| imgseg | 0.756 | **0.859** | 0.854 | 0.843 | 0.847 |
| lympho | 0.959 | **0.979** | 0.976 | 0.976 | 0.968 |
| satellite | 0.858 | 0.808 | 0.811 | 0.849 | **0.862** |
| satimage-2 | 0.885 | 0.871 | 0.870 | 0.892 | **0.906** |
| vertebral | 0.134 | **0.269** | 0.249 | 0.237 | 0.231 |
| WPBC | 0.376 | **0.399** | 0.396 | 0.386 | 0.384 |
| musk | 0.528 | **1.000** | **1.000** | **1.000** | **1.000** |
| cadio | 0.702 | 0.622 | 0.624 | 0.687 | **0.718** |
| WDBC | 0.975 | 0.994 | **0.997** | 0.991 | 0.983 |
| mammography | 0.333 | **0.571** | **0.571** | 0.565 | 0.554 |
| pima | 0.666 | 0.740 | **0.747** | 0.728 | 0.721 |

Table 21: Wasserstein distance metrics for quantifying anomaly hardness. $d(\mathcal{A}, \mathcal{N})$ is the multivariate Wasserstein distance between anomalies $\mathcal{A}$ and normal samples $\mathcal{N}$ via optimal-transport in the feature space. Lower distance indicates higher hardness (more overlap with normal). The ratio $d(\mathcal{A}_{\text{Syn}}, \mathcal{N})/d(\mathcal{A}_{\text{True}}, \mathcal{N})$ is the *Relative Hardness Ratio*.

| Base detector: PCA | | | | | | |
| --- | --- | --- | --- | --- | --- | --- |
| Datasets | $d(\mathcal{A}_{\text{Syn}}, \mathcal{N}) / d(\mathcal{A}_{\text{True}}, \mathcal{N})$ | $d(\mathcal{A}_{\text{True}}, \mathcal{N})$ | $d(\mathcal{A}_{\text{Syn}}, \mathcal{N})$ | $d(\mathcal{A}_{\text{Syn}_{\text{High}}}, \mathcal{N})$ | $d(\mathcal{A}_{\text{Syn}_{\text{Medium}}}, \mathcal{N})$ | $d(\mathcal{A}_{\text{Syn}_{\text{Low}}}, \mathcal{N})$ |
| imgseg | 1.870 | 4.963 | 9.279 | 9.148 | 10.534 | 11.953 |
| lympho | 0.540 | 17.529 | 9.474 | 9.504 | 18.017 | 58.429 |
| satellite | 1.431 | 7.828 | 11.198 | 11.078 | 11.386 | 13.125 |
| satimage-2 | 0.457 | 14.716 | 6.728 | 6.734 | 7.194 | 8.066 |
| vertebral | 8.141 | 1.691 | 13.765 | 13.895 | 14.911 | 16.732 |
| WPBC | 0.915 | 5.407 | 4.947 | 4.962 | 5.237 | 5.875 |
| musk | 0.261 | 31.207 | 8.157 | 8.179 | 9.249 | 12.112 |
| cardio | 0.724 | 12.054 | 8.726 | 8.736 | 9.088 | 9.769 |
| WDBC | 0.322 | 16.559 | 5.340 | 5.724 | 8.706 | 9.159 |
| mammography | 1.424 | 5.439 | 7.743 | 7.746 | 7.903 | 8.813 |
| pima | 5.944 | 2.376 | 14.124 | 14.173 | 17.924 | 20.202 |

| Base detector: IForest | | | | | | |
| --- | --- | --- | --- | --- | --- | --- |
| Datasets | $d(\mathcal{A}_{\text{Syn}}, \mathcal{N}) / d(\mathcal{A}_{\text{True}}, \mathcal{N})$ | $d(\mathcal{A}_{\text{True}}, \mathcal{N})$ | $d(\mathcal{A}_{\text{Syn}}, \mathcal{N})$ | $d(\mathcal{A}_{\text{Syn}_{\text{High}}}, \mathcal{N})$ | $d(\mathcal{A}_{\text{Syn}_{\text{Medium}}}, \mathcal{N})$ | $d(\mathcal{A}_{\text{Syn}_{\text{Low}}}, \mathcal{N})$ |
| imgseg | 1.050 | 4.963 | 5.213 | 5.214 | 5.678 | 6.168 |
| lympho | 0.224 | 17.529 | 3.920 | 3.919 | 4.050 | 4.263 |
| satellite | 0.772 | 7.828 | 6.042 | 6.056 | 6.773 | 7.922 |
| satimage-2 | 0.222 | 14.716 | 3.262 | 3.258 | 4.910 | 7.228 |
| vertebral | 6.440 | 1.691 | 10.889 | 10.768 | 10.910 | 11.729 |
| WPBC | 0.685 | 5.407 | 3.705 | 3.718 | 4.057 | 4.763 |
| musk | 0.125 | 31.207 | 3.906 | 3.917 | 4.917 | 5.196 |
| cardio | 0.379 | 12.054 | 4.573 | 4.588 | 7.112 | 8.878 |
| WDBC | 0.174 | 16.559 | 2.888 | 2.919 | 3.071 | 4.471 |
| mammography | 1.518 | 5.439 | 8.257 | 8.276 | 8.377 | 8.658 |
| pima | 3.557 | 2.376 | 8.453 | 8.438 | 8.950 | 9.052 |

set using an optimal-transport formulation EMD (Earth Mover's Distance), where the two point

clouds are treated as empirical distributions in the feature space. Unlike the classical 1D Wasserstein distance computed over scalar score distributions, our implementation uses the full multivariate point sets (normal vs. synthetic anomalies) and measures their geometric discrepancy through an optimal transport cost.

We denote $d(\mathscr{A}, \mathscr{n})$ as the multivariate Wasserstein distance between anomalies $\mathscr{A}$ and normal samples $\mathscr{n}$ via optimal-transport in the feature space. A lower value of $d(\mathscr{A}, \mathscr{n})$ indicates that anomalies lie closer to the normal manifold and are therefore *harder* to detect.

Table 21 reports:

- $d(\mathscr{A}_{\text{True}}, \mathscr{n})$: the inherent difficulty of the dataset (how far true anomalies are from normal samples),

- $d(\mathscr{A}_{\text{Syn}}, \mathscr{n})$: distance for anomalies synthesized by the original LLM-DAS,

- $d(\mathscr{A}_{\text{Syn}_{\text{High}}}, \mathscr{n})$, $d(\mathscr{A}_{\text{Syn}_{\text{Medium}}}, \mathscr{n})$, $d(\mathscr{A}_{\text{Syn}_{\text{Low}}}, \mathscr{n})$: distances under each hardness level,

- the ratio $\frac{d(\mathscr{A}_{\text{Syn}}, \mathscr{n})}{d(\mathscr{A}_{\text{True}}, \mathscr{n})}$, which we refer to as the *Relative Hardness Ratio*.

Two observations are particularly important:

- **Monotonic trend across hardness levels.** For both PCA and IForest, $d(\mathscr{A}_{\text{Syn}_{\text{High}}}, \mathscr{n})$ is consistently smaller than or comparable to the Medium and Low settings, and the distances increase from High $\rightarrow$ Medium $\rightarrow$ Low. This confirms that the hardness control interface is effective: High hardness produces anomalies closer to normal data, while Low hardness produces more easily separable anomalies.

- **Failure cases correspond to "overly hard" synthetic anomalies.** In the failure cases highlighted in red in Table 20, the ratio $\frac{d(\mathscr{A}_{\text{Syn}}, \mathscr{n})}{d(\mathscr{A}_{\text{True}}, \mathscr{n})}$ in Table 21 tends to be comparatively small. This means that the synthesized anomalies are much closer to normal samples than the true anomalies, even though the true anomalies are already far from normal (large $d(\mathscr{A}_{\text{True}}, \mathscr{n})$). In these failure cases, the problem is inherently easy, and generating very hard synthetic anomalies can over-complicate the learned boundary and hurt performance.

**Practical takeaway.** Taken together, Table 20 and Table 21 show that:

- LLM-DAS can synthesize anomalies at controllable hardness levels;

- the original implementation naturally produces hard anomalies (small Wasserstein distance);

- rare side effects arise when the dataset is inherently easy but the synthetic anomalies are "too hard";

- these side effects can be mitigated in practice by lowering the hardness level, which increases $d(\mathscr{A}_{\text{Syn}}, \mathscr{n})$ and restores performance to baseline or better.

**Prompt used in LLM-DAS variant with controlled hardness.** To further analyze the impact of anomalous samples with varying difficulty levels, we modified the prompt in the "Requirements" section by introducing a variable, the difficulty coefficient "degree", enabling LLMs to generate anomalies with varying degrees of difficulty. The specific prompt is as follows.

---

**Example Prompt for Generating Varying Difficulty Synthesis Code**

You are an expert in anomaly detection systems. The training set contains only normal samples. We use a IForest detector, where the anomaly score is computed using model.predict_score(). The higher the score, the more anomalous the sample.

**The description of IForest.**
*XXX.
*XXX.
*XXX.
...(Same as original prompt)

**Your task** is ...
*XXX.
*XXX.
*XXX.
...(Same as original prompt)

**Requirements:** Your should strictly follow below requirements:

...

Thus the function format is generate_hard_anomalies(n_samples: int, model, X_train: np.ndarray, degree)

**Supplementary explanation:** The parameter degree is used to generate anomalies with varying levels of detection difficulty, featuring three optional values:

- `low` (low difficulty)
- `med` (medium difficulty)
- `high` (high difficulty)

These correspond to generating anomalies that are hard, moderately hard, and extremely hard to detect, respectively. You need to generate anomalies of the corresponding difficulty level based on the specific value of degree.

Return only the complete Python function generate_hard_anomalies(...), with policy you used for genenrating anomalies and clear comments explaining key steps.

---

## 6.17 ANALYSIS OF GENERATING DETECTOR-SPECIFIC BINARY CLASSIFIERS

To further explore the potential of LLMs within the proposed framework, we redesigned the prompts in the "Objective" and "Requirements" sections to enable LLMs to develop detector-specific binary classifiers. These classifiers are then directly integrated with the original ones to enhance the anomaly detection performance of the baseline models. The specific prompt can be found in prompt 6.17.

---

**Example Prompt for Generating Detector-Specific Binary Classifiers**

You are an expert in anomaly detection systems. The training set contains only normal samples. We use a IForest detector, where the anomaly score is computed using model.predict_score(). The higher the score, the more anomalous the sample.

**The description of IForest.**

*XXX.

*XXX.

*XXX.

...(Same as original prompt)

**Your task** is to write a Python function detector_specific_binary_classifiers(...) tailored for PCA detectors. This function must complement PCA by focusing on capturing the aforementioned anomaly types that PCA fails to detect. Once the function is implemented, users will be able to provide it with the following inputs:

* A trained PCA model (model) that exposes predict_score(),

* The training samples (X_train)

**Requirements**: Your should strictly follow below requirements:

1. You must use your expertise to design a binary classifier that is specifically tailored for PCA detectors, not a model-agnostic classifier. The design should leverage the inherent characteristics and limitations of PCA to achieve complementarity.

2. The classifier should focus on capturing anomaly types that PCA fails to detect, particularly those that result in low scores from model.predict_score() (i.e., anomalies with low reconstruction error in PCA). To achieve this, you can first identify "borderline" normal training samples based on professional understanding of PCA's inherent characteristics (not solely relying on anomaly scores), then design the classifier to be sensitive to transformations of these samples that PCA cannot detect. The classifier's decision logic must be specific to PCA's weaknesses, making it ineffective as a general classifier for other detectors.

3. For the PCA model, you can only use the function model.predict_score() to interact with it.

4. The classifier must be trained on the provided training data and should expose standard fit(X, y) and predict_score(X) methods.

5. The function should allow setting:

* the trained PCA model (model),

* training samples (X_train).

Thus the function format is detector_specific_binary_classifiers(model, X_train: np.ndarray)

6. All package imports must be done inside the function.

Return only the complete Python function detector_specific_binary_classifiers(...), with the policy used for designing the classifier and clear comments explaining key steps.

---

## 6.18 ANALYSIS OF GENERATING SYNTHETIC SAMPLES TARGETING MULTIPLE WEAKNESSES OF BASE DETECTOR

To explore whether LLMs can identify multiple weaknesses of detectors and whether a mixture of synthetic anomalies targeting all these weaknesses enhances robustness, we modified the prompt in the "Objective" and "Requirements" sections by instructing LLMs to first identify several distinct logical weaknesses of the detector and then generate a mixture of synthetic anomalies targeting each weakness. The specific prompt can be found in prompt 6.18. And the example of the generation policy generated by the LLMs is available in Code2. The results are provided in Table 22.

Table 22: Performance comparison between the original LLM-DAS and the multi-weakness variant LLM-DAS (M-W), which explicitly prompts the LLM to identify and target several challenging anomaly patterns per detector.

|  | PCA | +LLM-DAS (Ours) | +LLM-DAS (M-W) | IForest | +LLM-DAS (Ours) | +LLM-DAS (M-W) |
|---|---|---|---|---|---|---|
| abalone | 0.839 | 0.839 | 0.848 | 0.848 | 0.855 | 0.850 |
| annthyroid | 0.566 | 0.660 | 0.582 | 0.615 | 0.620 | 0.623 |
| imgseg | 0.772 | 0.824 | 0.799 | 0.756 | 0.859 | 0.822 |
| lympho | 1.000 | 0.989 | 1.000 | 0.959 | 0.979 | 0.972 |
| mammography | 0.417 | 0.579 | 0.431 | 0.333 | 0.571 | 0.453 |
| pendigits | 0.386 | 0.485 | 0.397 | 0.513 | 0.602 | 0.613 |
| pima | 0.701 | 0.751 | 0.749 | 0.666 | 0.740 | 0.758 |
| satellite | 0.778 | 0.792 | 0.794 | 0.858 | 0.808 | 0.867 |
| satimage-2 | 0.919 | 0.931 | 0.951 | 0.885 | 0.871 | 0.950 |
| shuttle | 0.963 | 0.991 | 0.991 | 0.917 | 0.995 | 0.995 |
| vertebral | 0.138 | 0.143 | 0.141 | 0.134 | 0.269 | 0.138 |
| vowels | 0.105 | 0.109 | 0.228 | 0.098 | 0.178 | 0.190 |
| wilt | 0.064 | 0.097 | 0.090 | 0.085 | 0.278 | 0.088 |
| WPBC | 0.394 | 0.408 | 0.400 | 0.376 | 0.399 | 0.405 |
| breastw | 0.993 | 0.996 | 0.994 | 0.945 | 0.995 | 0.994 |
| cardio | 0.863 | 0.824 | 0.873 | 0.702 | 0.622 | 0.834 |
| WDBC | 0.983 | 0.980 | 0.982 | 0.975 | 0.994 | 0.980 |
| Average | 0.640 | 0.670 | 0.662 | 0.627 | 0.684 | 0.678 |

---

**Example Prompt for Generating Synthetic Samples Targeting Multiple Weaknesses**

You are an expert in anomaly detection systems. The training set contains only normal samples. We use a IForest detector, where the anomaly score is computed using model.predict_score(). The higher the score, the more anomalous the sample.
**The description of IForest.**
*XXX.
*XXX.
*XXX.
...(Same as original prompt)
**Your task** is to write a Python function generate_hard_anomalies(...) that generates anomalies which are the most difficult for the IForest detector to detect. This means that you the generated anomalies should have relatively low anomaly score, thus they are hard to be detected. But these anomalies are helpful to build a more robust detector. **Critically, you must first identify multiple distinct logical weaknesses of IForest (not just one), then design targeted anomaly generation policies for each weakness. The final output should be a mixture of synthetic anomalies targeting all identified weaknesses to maximize robustness.** After the function is completed, users can provide it with:
* A trained IForest model (model) that exposes predict_score(),
* The training samples (X_train)
**Requirements**: Your should strictly follow below requirements:
1. **You must first identify more than one distinct, logical weaknesses of IForest.** For each weakness, clearly explain the logical basis (how it arises from IForest's design) in the function's comments.
2. **For each identified weakness, design a unique, IForest-specific anomaly generation policy (not model-agnostic).** For each policy , you can first find the set of 'borderline' normal training samples based on your unique and professional understanding to IForest, not only based on the anomaly score. Then transform them to anomalies that is tailor-designed for IForest. Please note that the transformation function should be specific for IForest, which means that it is not a general transformation for other detectors.
*XXX.
*XXX.
*XXX.
...(Same as original prompt)

Code 2: Generated Code for Synthesizing Anomalies Targeting Multiple Weaknesses of IForest

```python
import numpy as np

def generate_hard_anomalies(n_samples: int, model, X_train: np.
    ndarray):
    """
    Generates anomalies that are difficult for an Isolation Forest
        (IForest) model to detect.

    This function operates by first identifying three distinct
        logical weaknesses of the IForest
    algorithm, then implementing a targeted generation policy for
        each weakness. The final
    output is a mixture of synthetic anomalies from all policies.

    Args:
        n_samples (int): The total number of hard anomalies to
            generate.
        model: A trained IForest-like model object that exposes a
            'predict_score' method.
                The method should return a score where a higher
                    value indicates a higher
                likelihood of being an anomaly.
        X_train (np.ndarray): The training data used to train the
            model, containing only
                                normal samples. Shape (
                                    n_train_samples, n_features).

    Returns:
        np.ndarray: An array of generated hard anomalies, with
            shape (n_samples, n_features).
    """
    # 1. Package Imports
    import numpy as np

    # 2. Pre-computation and Helper Functions
    if n_samples == 0:
        return np.array([])

    n_train, d = X_train.shape
    if d == 0:
        return np.zeros((n_samples, 0))

    # The user-provided 'model.predict_score()' is assumed to
        return higher scores for
    # more anomalous samples. We use these scores to stratify the
        training data into
    # topological regions as perceived by the IForest model itself
        .
    train_scores = model.predict_score(X_train)

    # Define quantiles to identify different types of normal
        points. This is a nuanced
    # way to select 'borderline' samples based on the model's own
        perception of the data.
    # - Central points: Deep inside the data cloud (low scores).
    # - Borderline points: On the fringes of dense regions (medium
         scores).
    # - Edge points: Most isolated of the normal points (high
        scores for a normal sample).
    score_quantiles = np.percentile(train_scores, [25, 75, 95])
    central_mask = train_scores <= score_quantiles[0]
    borderline_mask = (train_scores > score_quantiles[0]) & (
        train_scores <= score_quantiles[1])
    edge_mask = train_scores >= score_quantiles[2]
```

```
47
48       # Ensure we have points in each category to draw from.
49       X_central = X_train[central_mask] if np.any(central_mask) else
             X_train
50       X_borderline = X_train[borderline_mask] if np.any(
             borderline_mask) else X_train
51       X_edge = X_train[edge_mask] if np.any(edge_mask) else X_train
52
53       # Global statistics for transformations
54       data_min = np.min(X_train, axis=0)
55       data_max = np.max(X_train, axis=0)
56       data_range = data_max - data_min
57       data_std = np.std(X_train, axis=0)
58       data_centroid = np.mean(X_train, axis=0)
59
60       # Avoid division by zero for constant features
61       data_range[data_range == 0] = 1.0
62       data_std[data_std == 0] = 1.0
63
64       # 3. IForest Weaknesses and Corresponding Anomaly Generation
             Policies
65
66       generated_anomalies = []
67
68       # Allocate samples to each policy
69       n_policy1 = n_samples // 3
70       n_policy2 = n_samples // 3
71       n_policy3 = n_samples - n_policy1 - n_policy2
72       policy_allocations = [n_policy1, n_policy2, n_policy3]
73
74       # ---
75       # Policy 1: Clustered Anomalies
76       # ---
77       # Weakness ID: "Masking" or "Crowding" Effect.
78       # Logical Basis: IForest isolates points. If several anomalies
             are positioned closely
79       # together, they form a dense cluster. The algorithm will then
             require many splits to
80       # isolate any single point within this new cluster, making
             them appear "normal" (i.e.,
81       # giving them longer path lengths and thus lower anomaly
             scores). This policy creates
82       # such a dense micro-cluster in a sparse area of the feature
             space.
83       if policy_allocations[0] > 0:
84           # Step 1: Find a 'borderline' region by selecting a normal
                 point from a sparse area.
85           # We use 'edge' points as seeds because they are already
                 in less dense regions,
86           # making them good candidates for locating our new anomaly
                 cluster far from the main data.
87           seed_point = X_edge[np.random.randint(0, len(X_edge))]
88
89           # Step 2: Define the center of the anomaly cluster by
                 pushing the seed point
90           # further away from the data's centroid.
91           cluster_center = data_centroid + 2.0 * (seed_point -
                 data_centroid)
92
93           # Step 3: Transform by generating a very tight cluster
                 around this new center.
94           # The small covariance ensures the points are very close,
                 maximizing the crowding effect.
95           cluster_cov = np.diag((data_std / 50.0)**2) # Very small
                 variance
```

```
96          anomalies_p1 = np.random.multivariate_normal(
97              mean=cluster_center, cov=cluster_cov, size=
                    policy_allocations[0]
98          )
99          generated_anomalies.append(anomalies_p1)
100
101     # ---
102     # Policy 2: Diagonal Anomalies
103     # ---
104     # Weakness ID: Inefficiency of Axis-Parallel Splits for
           Diagonal Patterns.
105     # Logical Basis: IForest builds trees by splitting on a single
            feature at a time, creating
106     # axis-parallel decision boundaries. It is inefficient at
           isolating anomalies that only
107     # become apparent through a combination of features (e.g.,
           lying on a diagonal).
108     # Such points require many "stair-step" splits to be isolated,
            increasing their path
109     # length and lowering their anomaly score.
110     if policy_allocations[1] > 0:
111         anomalies_p2 = []
112         # Step 1: Select 'borderline' normal samples. These are
                points on the edge of the main
113         # data cloud, making them good candidates to be pushed
                slightly further out in a
114         # non-obvious, diagonal direction to become hard-to-detect
                anomalies.
115         base_points_indices = np.random.randint(0, len(
                X_borderline), size=policy_allocations[1])
116         base_points = X_borderline[base_points_indices]
117
118         for point in base_points:
119             anomaly = point.copy()
120             # Step 2: IForest-specific transformation. Choose two
                    random features and push the point's
121             # values along both dimensions simultaneously,
                    creating a diagonal shift.
122             if d > 1:
123                 f1, f2 = np.random.choice(d, 2, replace=False)
124
125                 # Perturb by a magnitude related to the standard
                        deviation of each feature
126                 perturbation1 = np.random.choice([-1, 1]) * 2.5 *
                        data_std[f1]
127                 perturbation2 = np.random.choice([-1, 1]) * 2.5 *
                        data_std[f2]
128
129                 anomaly[f1] += perturbation1
130                 anomaly[f2] += perturbation2
131             else: # Handle 1D case gracefully
132                 anomaly[0] += np.random.choice([-1, 1]) * 2.5 *
                        data_std[0]
133
134             anomalies_p2.append(anomaly)
135         generated_anomalies.append(np.array(anomalies_p2))
136
137     # ---
138     # Policy 3: Subspace Anomalies
139     # ---
140     # Weakness ID: High-Dimensional Feature Sparsity.
141     # Logical Basis: In high-dimensional spaces, IForest randomly
           selects a feature to split on
142     # at each node. If a point is anomalous only within a small
           subset of features (a subspace),
```

```
143    # the algorithm is unlikely to select these specific features
           for splitting early on. It will
144    # waste many splits on irrelevant dimensions where the point
           looks normal, again increasing
145    # its path length and making it appear less anomalous.
146    if policy_allocations[2] > 0:
147        anomalies_p3 = []
148        # Step 1: Select 'central' normal samples. These points
               have the lowest anomaly scores and
149        # are deep within the normal data distribution. Modifying
               them subtly makes them hard to
150        # distinguish from their many legitimate neighbors.
151        base_points_indices = np.random.randint(0, len(X_central),
               size=policy_allocations[2])
152        base_points = X_central[base_points_indices]
153
154        # Step 2: Define a small subspace (a few features) to
               perturb.
155        num_perturbed_dims = max(1, int(d * 0.1)) # Perturb ~10%
               of features
156
157        for point in base_points:
158            anomaly = point.copy()
159            # Step 3: Apply the IForest-specific subspace
                   transformation.
160            perturbed_dims = np.random.choice(d,
                   num_perturbed_dims, replace=False)
161            for dim in perturbed_dims:
162                # Push the value just outside the observed range
                       of normal data for that feature.
163                epsilon = 0.1 * data_range[dim]
164                if np.random.rand() > 0.5:
165                    anomaly[dim] = data_max[dim] + epsilon
166                else:
167                    anomaly[dim] = data_min[dim] - epsilon
168            anomalies_p3.append(anomaly)
169        generated_anomalies.append(np.array(anomalies_p3))
170
171    # 4. Final Combination and Output
172    if not generated_anomalies:
173        # Fallback in case n_samples was small and all policies
               got 0
174        return np.empty((0, d))
175
176    final_anomalies = np.vstack(generated_anomalies)
177
178    # Shuffle to mix anomalies from different policies
179    np.random.shuffle(final_anomalies)
180
181    return final_anomalies[:n_samples] # Ensure exact number is
           returned
```

## 6.19    HOW WE ENSURE THE ROBUSTNESS OF LLM-DAS?

LLM-DAS relies on the LLM's ability to interpret a detector's algorithmic description. However, the framework is not brittle and is intentionally designed to remain robust even when the LLM's analysis is imperfect.

**(1) The LLM analyzes a clean, parameter-free, logical description—not raw tabular data.** The LLM's task is restricted to summarizing the detector's high-level mechanism and reasoning about its structural weaknesses. These descriptions (e.g., PCA steps) are symbolic and algorithmic rather than data-dependent, making them significantly easier and more reliable for LLMs to understand.

**(2) The prompt imposes strict structural and functional constraints.** As detailed in Eq.3 of Section 3.2, the LLM is required to produce: (i) a detector-aware synthesis policy, (ii) an executable, data-agnostic program, (iii) an explanation of its reasoning. These constraints substantially reduce the risk of superficial reasoning by enforcing explicit, verifiable logic. For example, in Case Study of Section 3.2.2, the LLM's policy for IForest describes two core steps: (i) selecting borderline normal samples using the detector's own score function, and (ii) performing controlled extrapolation that preserves long path lengths. The accompanying explanation clarifies why these operations align with the algorithmic weakness of IForest (i.e., difficulty isolating deep points), making the LLM's intentions transparent and allowing us to verify that the generated code faithfully implements this logic. This level of specificity prevents arbitrary code generation.

**(3) The final detector does not rely solely on the LLM-generated component.** After generating synthetic anomalies, we train an enhancement classifier $\tilde{f}_t$ and combine it with the original detector $f_t$ using a normalized score-level ensemble. This fusion (Eq.7 of Section 3.2.2) ensures that even if the LLM's synthesis is occasionally suboptimal, the original detector's behavior remains preserved, preventing degradation.

**(4) Extensive empirical evidence demonstrates robustness across detectors, datasets, and LLMs.** The results in Table 1 of manuscript indicate that the improvements over the original source detectors across 36 benchmark datasets are statistically significant. The detailed results in Table 4 and Table 5 of Appendix 6.7 also show that the suboptimal results are occasional. In addition, our experiments in Table 8 of Appendix 6.9 further verify that LLM-DAS remains effective across different LLMs, including GPT-4o, Gemini-2.5-Pro, and Qwen3.

**(5) Boundary refinement after enhancement.** Fig. 5(b) of manuscript shows that the synthetic anomalies lie close to the normal manifold and receive scores similar to borderline normal samples under the original detector $f_t$. This confirms that the generated samples match the detector-aware policy and are indeed "hard" for the original detector. Crucially, Fig. 5(c) of manuscript demonstrates that after training with LLM-DAS, the enhanced detector exhibits a clear rightward shift of both synthetic and real anomaly score distributions relative to normal data. This tightening of the decision boundary shows that the synthesized anomalies not only challenge the detector but meaningfully reshape the classifier to produce better separation. This is direct evidence that the LLM-generated samples are of sufficiently high quality to improve the detector's generalization.

## 6.20 ON ISOLATING THE CONTRIBUTION OF LLM REASONING

What knowledge is LLM actually contributed? Our claim is not that the LLM invents entirely new mathematics, but that it performs a contextual, detector-specific reasoning and synthesis task that goes far beyond retrieving a single, static rule.

**The Ablations Test Specific, Necessary Components:** Our ablation studies (Generic/Simple/Random in Fig. 4(b)) are designed to remove specific, instructed capabilities from the LLM:

- "Generic": Removes detector-awareness. The LLM cannot perform reasoning conditioned on the target detector's algorithm.
- "Simple": Removes the capability to generate hard anomalies by denying access to the score function.
- "Random": Removes the borderline heuristic, a key strategic element.

The significant performance drop in each variant confirms that each of these instructed reasoning components is critical. This shows that the LLM is not just executing a single built-in rule but is dynamically combining these elements based on the prompt.

**The Cross-Detector Experiment is the Key Evidence (Connecting to Table 2):** The most compelling evidence against a "universal rule" hypothesis comes from our cross-detector experiment (Table 2).

- If the LLM were simply applying a generic, detector-agnostic heuristic (e.g., "add small noise" or "extrapolate"), then the synthesis policy $\mathcal{S}_{\text{program}}^A$ for Detector A should work reasonably well when transferred to Detector B.

- The results clearly show the opposite: Transferred strategies often perform poorly. This demonstrates that each LLM-generated policy is highly specialized and its effectiveness is contingent on a precise match between the synthesis logic and the target detector's internal mechanics. This specialization is the hallmark of genuine, detector-specific reasoning.

In summary, the ablations prove the necessity of key reasoning components, and the cross-detector experiment proves the specificity of the resulting strategies. Together, they robustly isolate and validate the contribution of the LLM's contextual reasoning capabilities.

