# OpenReview forum: "LLM as an Algorithmist: Enhancing Anomaly Detectors via Programmatic Synthesis"
_ICLR.cc/2026/Conference — ICLR 2026 Poster_

### Official Review · Reviewer_JZ9t · 2025-10-28

**Soundness:** 2
**Presentation:** 2
**Contribution:** 2
**Rating:** 2
**Confidence:** 4

**Summary:**

This paper introduces the use of large language models (LLMs) to augment training datasets, claiming that LLMs can synthesize additional anomalous samples. However, traditional methods for tabular anomaly detection primarily follow the one-class classification paradigm, where detectors are trained solely on normal data. By training detectors on a combination of normal samples and LLM-generated anomalous samples, the authors report significant performance improvements across various datasets and detection methods.

**Strengths:**

1. The figures, pipelines, and writing are easy to understand.
2. The paper tackles an interesting direction: using LLMs to augment the training dataset of anomaly detection in the tabular dataset, potentially as a way to alleviate the strong assumption posed by the existing detectors.

**Weaknesses:**

- The paper only considers general anomaly detection. However, there are multiple types of anomalies—such as point anomalies, contextual anomalies, and others—that differ in nature and characteristics. The proposed method does not take this diversity into account.

- Delegating the anomaly generation process to LLMs raises concerns. How can we ensure that the generated samples are indeed anomalous? More importantly, how is an anomaly formally defined in this context? Although the authors provide some visualizations in Figure 5, a more thorough discussion of what constitutes an anomaly point is necessary.

- It remains unclear how to evaluate the hardness of the generated anomaly samples. Can the LLMs generate anomalies with varying degrees of difficulty?

- The potential side effects of these synthetic samples are not discussed. For instance, could they negatively affect the generalizability of the detector, given that LLMs might exhibit biases toward certain types of anomalies and thus overrepresent them?

- Experiments: The ratio of blended synthetic anomaly samples is insufficiently studied. An ablation study examining the optimal proportion of anomaly samples needed to improve robustness would strengthen the empirical analysis.

**Questions:**

> ...This leads to tabular anomaly detection (TAD) usually being studied under a one-class classification paradigm, where the detector is trained solely on normal samples to distinguish anomalies at test time...

I have great concern about this argument. Other well-defined AD methods in the general domain, such as Self-Inf [1] can be easily applied to the tabular dataset. But Self-Inf is apparently not a one-class classification method. Following this concern, I would doubt whether the challenge of 'one-class classification' can well support your motivation for introducing LLMs to augment the training dataset.

---

> ### Author Response · Authors · 2025-11-20
> **Response (Part 1/4)**
>
> **R to W1:**
> We thank the reviewer for this insightful comment regarding the diversity of anomaly types. We agree that this is a crucial aspect of anomaly detection. Our core contribution is a paradigm that dynamically addresses this diversity by targeting the specific weaknesses of each detector, rather than relying on a static, one-size-fits-all notion of ''anomaly''.
> We kindly refer the reviewer to Appendix 6.15 for full experimental details, analyses, and extended results.
>
> **(1) Different datasets contain different anomaly types; different detectors excel at different anomaly types.**
> As highlighted in ADBench [1], real tabular datasets rarely come with a standardized anomaly taxonomy—datasets may contain local, global, dependency-breaking, or clustered anomalies in unknown proportions.  Meanwhile, classical detectors have well-documented biases: e.g., Isolation Forest and OCSVM struggle with dependency/local anomalies, while performing well on clustered anomalies. Thus, anomaly-type diversity is inherently detector-dependent.
>
> **(2) Core mechanism of LLM-DAS: detector-aware anomaly synthesis.**
> LLM-DAS does not assume any predefined taxonomy of anomaly types.  Instead, for each detector $t$, the LLM receives a high-level description of the detector’s algorithmic principles and produces a piece of synthesis code $\text{Code}^t$ intended to generate samples that are ''hard'' for that specific detector.  When instantiated on a dataset $D$ and a trained detector $f_t$, this code produces a synthetic anomaly set $A(D,t)$ that reflects the detector’s algorithmic blind spots.
> This design ensures that anomaly-type diversity is handled through a detector-centric lens.
>
> **(3) Controlled evaluation: constructing four testbeds with ground-truth anomaly types.**
> To empirically validate what types of anomalies are synthesized by $\text{Code}^t$, we adopt the canonical ADBench protocol.
> Starting from a normal-only base dataset $D$, we construct four benchmark datasets by injecting one specific anomaly type (Local, Global, Dependency, Clustered).  Each benchmark consists of training data $D_{\mathrm{train}}^{(\tau)}$ (normal only) and test data $D_{\mathrm{test}}^{(\tau)}$ containing the injected anomaly type $\tau$.  This setup enables controlled and fair comparison between synthesized anomalies by LLM-DAS and each canonical anomaly type from benchmark.
>
> **(4) Key findings: LLM-DAS automatically targets the anomaly types that each detector performs worst on.**
> For each detector $t$, we instantiate $\text{Code}^t$ on $D_{\mathrm{train}}^{(\tau)}$ to obtain $A(D_{\mathrm{train}}^{(\tau)}, t)$ and compare it against each of the four anomaly types using Wasserstein distance. The results (Table 19 in Appendix 6.15) show a clear and consistent pattern: LLM-DAS does not generate a fixed or arbitrary type of anomaly; rather, it adaptively synthesizes samples that align with the weaknesses of each base detector.  As shown in Table 19, the synthesized anomalies for IForest and OCSVM are distributionally closer to dependency- and local-type anomalies—precisely the types where these detectors
> perform poorly according to ADBench.  Similarly, for LOF, which struggles with clustered-type anomalies, LLM-DAS produces anomalies most similar to clustered structures.
>
> Correspondingly, Table 19 and Fig. 11 - 13 demonstrate that LLM-DAS yields the largest performance gains and the most significant score-separation improvements on these hard anomaly types.  This evidence supports that the framework indeed learns to target and reinforce the most challenging decision regions for each detector, rather than fitting or imitating any single pre-defined anomaly taxonomy.
>
> Meanwhile, for anomaly types where the base detector already performs well (e.g., IForest on Cluster anomaly type in Table 19),
> LLM-DAS maintains stable or slightly improved results, indicating that it enhances robustness without overfitting to particular anomaly categories.  The ensemble design further ensures that the original detector continues to dominate in regions it already models accurately, preserving its discriminative ability for easy anomalies while extending its coverage toward harder cases.
>
> **(5) Summary.**
> LLM-DAS inherently accounts for anomaly-type diversity through a detector-aware synthesis mechanism.
> It does not rely on fixed anomaly-type assumptions but instead adapts to each detector’s algorithmic weaknesses.
> This explains why the method achieves consistent gains across heterogeneous datasets with unknown anomaly structures.
> Appendix 6.15 presents this logic and findings more clearly and we thank the reviewer for prompting this valuable clarification.
> In the future, we plan to extend this line of analysis by developing a controllable synthesis module that explicitly conditions the LLM on user-specified anomaly types (e.g., emphasizing dependency or clustered patterns.
>
> [1] ADBench: Anomaly Detection Benchmark. NeurIPS 2022.

---

> > ### Author Response · Authors · 2025-11-20
> > **Response (Part 2/4)**
> >
> > **R to W2:** Thanks for your insightful suggestion! We address your concerns through two perspectives, (1) the constitution of anomalies and (2) the guarantee of LLM generation.
> >
> > **(1) The constitution of anomalies.**
> > As discussed in our response to W1, the notion of “anomaly” in tabular data is inherently complex and not standardized across datasets.  Following ADBench[1] and other recent works[2,3], anomalies may manifest as local, global, dependency, or clustered deviations from the normal data manifold.  In practice, real datasets often contain a mixture of such mechanisms without explicit labels.  Therefore, consistent with the tabular anomaly detection literature, we define anomalies operationally as samples that deviate from the detector’s learned normal distribution.  Our proposed LLM-DAS framework does not aim to generate anomalies of a particular pre-defined type, but rather generates detector-specific ''hard'' samples that expose each algorithm’s structural weakness—that is, the samples that the detector $f_t$ itself finds most difficult to classify as anomalous. The details in Appendix 6.15 and response to W1 could verify it.
> >
> > **(2) Guarantee of the LLM-generated anomalies by design.**
> >
> > - **(a) Detector-aware prompt design.**
> > For detector $f_t$, the LLM is provided with its parameter-free algorithmic description and functional pseudocode (e.g., the steps of PCA, OCSVM, or IForest).  The prompt explicitly asks the LLM to analyze the detector’s logic, identify potential blind spots in its detection behavior, and design a corresponding anomaly synthesis policy.  This ensures that the generated strategy is grounded in the detector’s own inductive bias rather than arbitrary data manipulation.
> >
> > - **(b) Structured and constrained generation.**
> > The LLM’s output is strictly structured into three components:  (1) a synthesis policy $S^{t} _ {policy}$, (2) executable Python code $S^{t} _ {program}$, and  (3) a textual explanation $S^{t} _ {explanation}$ describing the reasoning behind each step (Eq. 3, Sec. 3.2.1).  This structured format provides direct interpretability and verifiability.  Moreover, the generated code is executed under a controlled interface that grants access only to $(i)$ the detector’s scoring function $f_t(x)$, $(ii)$ the normal training set $D_{\text{train}}$, and $(iii)$ the desired number of synthetic samples $N_{\text{syn}}$ (Eq. 4–5, Sec. 3.2.2).  It cannot access internal model states or dataset-specific variables, ensuring consistent and safe execution across detectors.
> >
> > - **(c) Boundary-based synthesis mechanism.**
> > Instead of generating anomalies from scratch, the LLM learns to first locate borderline normal samples—those with scores close to the decision threshold—and then apply controlled transformations to push them just beyond the normal region.  This design guarantees that the synthesized samples are close to, but outside, the learned normal manifold, corresponding precisely to ''hard anomalies'' that are difficult but meaningful.  The accompanying explanation $S^{t}_{explanation}$ makes this process transparent and easily inspectable.
> >
> > - **(d) Empirical confirmation.**
> > As illustrated in Fig. 5(b–c), the synthetic anomalies lie near the decision boundary and, after training with LLM-DAS, the enhanced detector exhibits a rightward shift in the KDE distribution of both synthetic and real anomaly scores.  This confirms that the generated anomalies effectively refine the decision boundary and improve anomaly separation, demonstrating that the generation process produces valid and detector-aware anomalies. In addition, as demonstrated in Table 1 of manuscript, LLM-DAS improves performance across 36 datasets and 5 detectors with statistically significant consistency.  As illustrated in Table 8 of Appendix 6.9, LLM-DAS achieves consistent gains across multiple LLMs, including Gemini-2.5-Pro, GPT-4o, and the open-source Qwen3 model.
> > The consistent and substantial performance boost serves as the strongest possible empirical guarantee that the LLM-generated code is robustly reliable and effectively captures fundamental algorithmic weaknesses.
> >
> > Together, these design principles and empirical results show that the LLM-generated anomalies are not arbitrary noise, but are purposefully synthesized to exploit the structural weaknesses of each detector—thereby enhancing robustness in a transparent and verifiable way.
> >
> > [1] ADBench: Anomaly Detection Benchmark. NeurIPS 2022.
> >
> > [2] DRL: Decomposed Representation Learning for Tabular Anomaly Detection. ICLR 2025.
> >
> > [3] MCM: Masked Cell Modeling for Anomaly Detection in Tabular Data. ICLR 2024.

---

> > ### Author Response · Authors · 2025-11-20
> > **Response (Part 3/4)**
> >
> > **R to W3 \& W4:**
> > We thank the reviewer for these insightful questions. In Appendix 6.16 of the revised version, we provide (i) a controllable mechanism for generating anomalies with varying hardness, and (ii) a principled explanation---together with a mitigation strategy---for the rare failure cases where overly hard synthetic anomalies may harm generalization. We provide here a high-level overview of our new analysis, and kindly refer the reviewer to Appendix 6.16 for detailed definitions, tables, and additional discussion.
> >
> > **(1) Can LLM-DAS generate anomalies with controllable difficulty? (W3)** We extend the synthesis prompt with a hardness-level interface (High/Medium/Low), allowing the LLM's synthesis code to adjust how close the synthetic anomalies lie to the normal-data manifold. Empirically (see Table 20 and Table 21 in Appendix 6.16), across multiple datasets and two detectors (PCA and IForest), switching the hardness level leads to consistent and interpretable changes in detection performance:
> > - The original LLM-DAS (used throughout the main paper) closely corresponds to the High-hardness setting.
> > - Medium and Low hardness produce anomalies progressively farther from the normal manifold.
> > - For almost every dataset, at least one hardness setting improves over the base detector, demonstrating robustness of the framework.
> > Thus, LLM-DAS supports a continuous spectrum of anomaly difficulty rather than a single fixed level.
> >
> > **(2) How do we evaluate anomaly hardness? (W3)** To formally quantify hardness, we compute a multivariate Wasserstein distance between:
> > - the distribution of synthetic anomalies and
> > - the distribution of normal samples,
> > treating both as empirical distributions in the original feature space. A smaller Wasserstein distance indicates that synthetic anomalies lie closer to the normal manifold and are therefore harder to distinguish.
> >
> > To account for dataset-specific inherent difficulty, we further compare this quantity with the distance between true anomalies and normal samples via the Relative Hardness Ratio: $\mathcal{R} = \frac{d(\mathcal{A} _ {\text{Syn}}, \mathcal{N})}{d(\mathcal{A} _ {\text{True}}, \mathcal{N})}$, where $d(\cdot,\cdot)$ is the Wasserstein distance, $\mathcal{A} _ {\text{Syn}}$ and $\mathcal{A}_{\text{True}}$ denote the synthetic and true anomalies, and $\mathcal{N}$ denotes the normal data.
> > A smaller $\mathcal{R}$ means that synthetic anomalies are much closer to normal samples than true anomalies (i.e., they are much harder), whereas larger $\mathcal{R}$ indicates comparable or easier difficulty. This metric allows us to compare hardness across datasets with different intrinsic anomaly–normal separations.
> >
> > **(3) Do overly hard synthetic anomalies cause negative side effects? (W4)** We agree with the reviewer that synthetic anomalies could, in principle, harm generalization if they are not properly controlled. Our new experiments confirm that such effects can occur and can be understood in terms of relative hardness. Specifically, we identify several ''fail cases'' (e.g., PCA + LLM-DAS on lympho, musk, cardio) where the performance of original LLM-DAS drops below the base detector. These cases share the following pattern: they are characterized by a low relative hardness ratio $\mathcal{R}$. This quantity compares the Wasserstein distance of synthetic anomalies to that of true anomalies, and a small $\mathcal{R}$ indicates that:
> > - the dataset is inherently easy---true anomalies are already far from the normal manifold (large $d(\mathcal{A}_{\text{True}}, \mathcal{N})$), and
> > - the synthesized anomalies are much harder than necessary---they lie disproportionately close to normal samples (small $d(\mathcal{A}_{\text{Syn}}, \mathcal{N})$).
> > In such situations, training on extremely hard synthetic anomalies may over-complicate the decision boundary near the normal manifold and slightly hurt generalization. In short, fail cases occur when synthetic anomalies are too hard relative to the dataset's inherent difficulty, and this is exactly what a small $\mathcal{R}$ quantifies.
> >
> > **(4) Mitigation via hardness-level control (W3 + W4)** The hardness interface offers a direct practical mechanism to mitigate these side effects:
> > - Lowering hardness from High to Medium/Low increases $d(\mathcal{A}_{\text{Syn}}, \mathcal{N})$ and thus increases $\mathcal{R}$, making synthetic anomalies better aligned with the dataset’s inherent difficulty.
> > - In all identified fail cases, switching to Medium or Low hardness restores the performance to the baseline level or even exceeds it (see Table 20 in Appendix 6.16).
> > This demonstrates that LLM-DAS not only allows hardness to be quantified, but also provides a tunable knob to prevent synthetic anomalies from becoming overly challenging for inherently easy datasets.

---

> ### Author Response · Authors · 2025-11-20
> **Response (Part 4/4)**
>
> **R to W5:**
> Thanks for your insightful question regarding the ratio of synthetic pseudo-anomalies!
>
> **(1) Why we initially used a 10\% ratio.**
> In our original LLM-DAS framework, we intentionally use only a small fraction of challenging synthetic anomalies when training the auxiliary binary classifier. This design prevents the classifier from being dominated by synthesized samples and helps preserve the inductive bias and decision structure of the base detector within the ensemble. Nonetheless, we fully agree that
> analyzing alternative ratios is important.
>
> **(2) New comprehensive experiments across all datasets and ratios 0.1-1.0.**
> To directly address your concern, we conducted an extensive analysis on all datasets with synthesis ratios from 0.1 to 1.0 (LLM-DAS uses 0.1 by default, and 1.0 indicates balanced ratio), replacing the original small-scale ratio study in the Appendix 6.12
> with a full-spectrum evaluation. Table 10-11 in the revised Appendix 6.12 summarize the results.
> Our extended experiments reveal that the overall performance gains of LLM-DAS over the base detector are robust across a wide range of synthesis ratios.
> Across many datasets, ratios within the range of 0.1–0.4 already achieve near-peak improvement, indicating that only a modest number of synthesized anomalies is sufficient to refine the decision boundary.
> More balanced or larger ratios do not yield significant additional benefits, and in some datasets even result in slight degradations, likely because excessive synthetic anomalies begin to distort the classifier’s decision landscape rather than complement the base detector.
> These findings suggest that LLM-DAS does not rely on a finely tuned anomaly ratio; instead, the
> synthetic anomalies primarily serve to sharpen the boundary near the detector’s weak regions,
> and this effect saturates early.
>
> We suggest to set the ratio to 0.1 by default, which balances performance gain and computational cost.
> These expanded results are included in the revised Appendix 6.12.
>
> **R to Q1:**
> We appreciate the your thoughtful observation.
> We agree that anomaly detection (AD) in general is not limited to the one-class classification paradigm:
> methods such as Self-Inf can also be applied to tabular data.
> Our statement that “TAD is usually studied under a one-class classification paradigm”
> was intended to describe the highly popular and practically significant paradigm in the
> tabular anomaly detection (TAD) literature, rather than to exclude other possible formulations.
> Specifically, recent representative TAD studies—including DRL[1], MCM[2], AnoLLM[3], NPT-AD[4] and ICL[5]—
> all adopt the one-class classification setting where the detector is trained purely on normal samples
> and distinguishes anomalies at test time. Hence, our focus on one-class TAD follows this widely adopted paradigm,
> which remains a key challenge in practice due to the absence of labeled anomalies.
> We have clarified this point in the revision and revise our phrasing to:  ''In tabular anomaly detection, many recent works adopt a one-class paradigm,
> where only normal samples are available for training, and our method is developed to enhance detectors in such setting.''
>
>
> We appreciate your valuable suggestion and will explore, in future work,
> how the proposed LLM-DAS framework can be extended beyond the one-class setting.  In particular, we plan to investigate its integration with other AD methods in the general domain, such as Self-Inf, examining whether detector-aware synthetic anomalies can also improve AD performance when labeled or proxy anomalies are available.
>
>
> [1] DRL: Decomposed Representation Learning for Tabular Anomaly Detection. ICLR 2025.
>
> [2] MCM: Masked Cell Modeling for Anomaly Detection in Tabular Data. ICLR 2024.
>
> [3] AnoLLM: Large Language Models for Tabular Anomaly Detection. ICLR 2025.
>
> [4] Beyond Individual Input for Deep Anomaly Detection on Tabular Data. ICML 2024.
>
> [5] Anomaly Detection for Tabular Data with Internal Contrastive Learning. ICLR 2022.

---

> ### Author Response · Authors · 2025-11-27
> **Response Summary and Opportunity for Further Discussion**
>
> Dear Reviewer JZ9t,
>
> Thank you once again for your time and thoughtful feedback. To respectfully follow up, we would like to briefly summarize the concrete steps we have taken to address each of your comments:
>
> **For W1:** Diversity of Anomaly Types. We expanded Appendix 6.15 to show that LLM-DAS inherently targets detector-specific weaknesses across anomaly types (local, global, dependency, clustered), without assuming a fixed taxonomy.
>
> **For W2:** Definition and Validity of Anomalies. We clarified the constitution of anomalies, and we implemented structured LLM generation with boundary-based synthesis to ensure meaningful and challenging samples.
>
> **For W3 & W4:** Hardness Control and Side Effects. We introduced a controllable hardness-level interface (High/Medium/Low) and a failure-mode analysis using Wasserstein distance and Relative Hardness Ratio (Appendix 6.16). In addition, we provided a principled mitigation strategy via controlled hardness-level adjustment.
>
> **For W5:** Synthetic Anomaly Ratio. We conducted a full-scale ablation across ratios 0.1–1.0 (Table 10–11, Appendix 6.12), confirming that LLM-DAS is robust to ratio choice and saturates early with minimal synthetic data.
>
> **For Q1:** One-Class Paradigm Justification. We revised the text to clarify that our work aligns with the widely adopted one-class setting in recent TAD literature, while acknowledging other paradigms like Self-Inf as promising future directions.
>
> We sincerely appreciate your detailed reviews, which have helped us significantly improve the clarity, rigor, and validation of our work. If you have any further questions or suggestions, we would be truly grateful to hear them.

---

> > ### Comment · Reviewer_JZ9t · 2025-11-27
> > **Response to Authors**
> >
> > Thanks for the authors' detailed response. These rebuttal messages successfully addressed my concerns. I increased my score from 2 to 6.

---

> ### Author Response · Authors · 2025-11-27
>
> Thank you for kindly increasing the score from 2 to 6 and for taking the time to review our response!

---

### Official Review · Reviewer_N2i1 · 2025-10-29

**Soundness:** 3
**Presentation:** 3
**Contribution:** 3
**Rating:** 4
**Confidence:** 4

**Summary:**

This paper tackles the task of tabular anomaly detection. The authors propose leveraging a large language model to analyze existing detectors and generate code that synthesizes hard-to-detect anomalies. A binary classification model is then trained and integrated with the existing detectors to perform anomaly detection. Extensive experimental results demonstrate the superiority of the proposed method.

**Strengths:**

1.	This paper is well written and easy to read.
2.	The proposed method sounds reasonable.
3.	The authors conduct extensive experiments to demonstrate the effectiveness of the proposed method.

**Weaknesses:**

1.	The paper employs an LLM to generate code for synthesizing pseudo-anomalies. However, it is unclear how the authors ensure the validity and reliability of the LLM’s outputs. The paper should include an explanation or verification procedure for this aspect.
2.	The proposed method utilizes the RandomForest as the binary classification model, the authors are expected to conduct experiments to analysis the robustness of the proposed method to various binary classifiers.
3.	The authors generate pseudo-anomalies for only 10% of the training set. It is unclear why this specific ratio was chosen. Moreover, the imbalance between normal and anomalous samples might affect the training of the binary classifier. Would using a balanced dataset improve the classifier’s performance?

**Questions:**

1.	How does the binary classifier perform if it is used alone for anomaly detection?
2.	Since the LLM is used to analyze the characteristics of existing anomaly detectors and generate “hard-to-detect” anomalies, it would be interesting to investigate whether the LLM’s analysis could also be leveraged to design detector-specific binary classifiers. This might further enhance the adaptability and effectiveness of the proposed framework.

---

> ### Author Response · Authors · 2025-11-20
> **Response (Part 1/3)**
>
> **R to W1:**
> We appreciate the reviewer’s concern regarding the correctness and reliability of the
> LLM-generated anomaly synthesis code. This is indeed an essential aspect of evaluating
> any framework that leverages LLMs for code generation. LLM-DAS incorporates several
> mechanisms that ensure both validity (the code can be executed safely and
> consistently) and reliability (the code captures the intended detector-aware
> behavior and produces meaningful hard anomalies):
>
> **(1) Controlled and restricted programming interface.**
> The generated code by LLM is only allowed to access three stable, abstract interfaces:
> $(i)$ the fitted detector $f_t$, $(ii)$ the normal training set $D_{\text{train}}$, and
> $(iii)$ the desired number of samples $N_{\text{syn}}$ (Eq. 4 and Eq. 5 of Section 3.2.2).  It cannot access internal model states or dataset-specific assumptions. This drastically limits the surface for invalid outputs and ensures universal executability.
>
> **(2) Structured outputs with explicit reasoning and verifiable intent.** The prompt requires the LLM to output: (a) a synthesis policy $S^{t} _ {policy}$, (b) executable Python code $S^{t} _ {program}$, and (c) an explanation $S^{t} _ {explanation}$ justifying key steps (Eq. 3 of Section 3.2.1). This structured requirement ensures that the code is not produced heuristically
> but is grounded in an explicit and inspectable line of reasoning.  For example, in Case Study of Section 3.2.2, the LLM’s policy for IForest describes two core steps: (1) selecting borderline normal samples using the detector’s own score function, and  (2) performing controlled extrapolation that preserves long path lengths.  The accompanying explanation clarifies why these operations align with the algorithmic weakness of IForest (i.e., difficulty isolating deep points), making the  LLM’s intentions transparent and allowing us to verify that the generated code faithfully  implements this logic.   This level of specificity prevents arbitrary or superficial code generation.
>
>
> **(3) Empirical reliability of the synthesized anomalies.**
> The reliability of the LLM-generated code is demonstrated through multiple
> consistent empirical indicators:
>
> - **Hardness of synthesized anomalies.**
> Fig. 5(b) of manuscript shows that the synthetic anomalies lie close to the normal manifold and
> receive scores similar to borderline normal samples under the original detector $f_t$.  This confirms that the generated samples match the detector-aware policy and are indeed “hard” for the original detector.
>
> - **Boundary refinement after enhancement.**
> Crucially, Fig. 5(c) of manuscript demonstrates that after training with LLM-DAS, the enhanced  detector exhibits a clear rightward shift of both synthetic and real anomaly score distributions relative to normal data.  This tightening of the decision boundary shows that the synthesized anomalies not only challenge the detector but meaningfully reshape the classifier to produce better separation. This is direct evidence that the LLM-generated samples are of sufficiently high quality to improve the detector's generalization.
>
> - **Consistent performance gains across datasets, detectors and LLMs.**
> As demonstrated in Table 1 of manuscript, LLM-DAS improves performance across 36 datasets and 5 detectors with statistically significant consistency.  In addition, as illustrated in Table 8 of Appendix 6.9, LLM-DAS achieves consistent gains across multiple LLMs, including Gemini-2.5-Pro, GPT-4o, and the open-source Qwen3 model.  The consistent and substantial performance boost serves as the strongest possible empirical guarantee that the LLM-generated code is robustly reliable and effectively captures fundamental algorithmic weaknesses.
>
> We have added a short subsection elaborating on these verification procedures in Appendix 6.19.

---

> > ### Author Response · Authors · 2025-11-20
> > **Response (Part 2/3)**
> >
> > **R to W2:** Thanks for your valuable comment! We have already verify the robustness of the proposed method to various binary classifiers in Fig. 9 of Appendix 6.12. The results show that the phenomenon of performance improvements are consistent across different classifiers, and the method is robust to different binary classifiers.
> > This confirms that our framework is not tied to RandomForest and is in fact robust to the choice of downstream binary classifier—the key performance gains stem from the synthesized anomalies rather than the specific classifier used.
> >
> > **R to W3:**
> > We appreciate the your insightful question regarding the ratio of synthetic pseudo-anomalies and the potential impact of class imbalance on the training of the
> > binary classifier.
> >
> > **(1) Why we initially used a 10\% ratio.**
> > In our original LLM-DAS framework, we intentionally use only a small fraction of challenging synthetic anomalies when training the auxiliary binary classifier. This design prevents the classifier from being dominated by synthesized samples and helps preserve the inductive bias and decision structure of the base detector within the ensemble. Nonetheless, we fully agree that
> > analyzing alternative ratios is important.
> >
> > **(2) New comprehensive experiments across all datasets and ratios 0.1-1.0.**
> > To directly address your concern, we conducted an extensive analysis on all datasets with synthesis ratios from 0.1 to 1.0 (LLM-DAS uses 0.1 by default, and 1.0 indicates balanced ratio), replacing the original small-scale ratio study in the Appendix 6.12
> > with a full-spectrum evaluation. Table 10-11 in the revised Appendix 6.12 summarize the results.
> > Our extended experiments reveal that the overall performance gains of LLM-DAS over the base detector are robust across a wide range of synthesis ratios.
> > Across many datasets, ratios within the range of 0.1–0.4 already achieve near-peak improvement, indicating that only a modest number of synthesized anomalies is sufficient to refine the decision boundary.
> > More balanced or larger ratios do not yield significant additional benefits, and in some datasets even result in slight degradations, likely because excessive synthetic anomalies begin to distort the classifier’s decision landscape rather than complement the base detector.
> > These findings suggest that LLM-DAS does not rely on a finely tuned anomaly ratio; instead, the
> > synthetic anomalies primarily serve to sharpen the boundary near the detector’s weak regions,
> > and this effect saturates early.
> > We suggest to set the ratio to 0.1 by default, which balances performance gain and computational cost.
> >
> > **R to Q1:**  Thanks for your valuable question! We have already included the results of only use binary classifier in Fig. 10 of Appendix 6.12. The results show that the standalone binary classifier performs better than the base detector, indicating that the classifier indeed captures useful complementary patterns learned from LLM-synthesized anomalies. However, the full LLM-DAS framework (fusing binary classifier and base detector) is still better than the standalone binary classifier.
> > This aligns with our design intuition: the binary classifier alone lacks access to the base detector’s global inductive biases and decision structure, whereas the fused model benefits from both the detector’s original strengths and the fine-grained refinements introduced by the classifier.
> > Since the additional binary classifier is lightweight, introduces negligible computational overhead (the computational cost is provided in Table 9 of Appendix 6.10), and enhances—rather than replaces—the base detector’s behavior, we recommend using the fused LLM-DAS model, which achieves the best overall performance.

---

> > ### Author Response · Authors · 2025-11-20
> > **Response (Part 3/3)**
> >
> > **R to Q2:**
> > Thanks for the insightful suggestion of leveraging the LLM’s analysis not only to generate hard anomalies, but also to design detector-specific binary classifiers. This idea is highly aligned with the spirit of LLM-DAS, and we conducted additional experiments to investigate it.
> >
> > **(1) How we instantiate detector-specific classifiers.**
> > In our original framework, the enhancement classifier $\tilde{f}_t$ is a standard discriminative model (e.g., Random Forest) trained on normal samples and LLM-DAS–generated hard anomalies. Although the classifier architecture itself is model-agnostic, its decision function becomes detector-specific through the training data, since the synthetic anomalies are explicitly tailored to the weaknesses of a particular detector.
> >
> > Following the reviewer’s suggestion, we further modified the prompt so that the LLM is asked to design an explicit detector-specific binary classifier rather than only a synthesis strategy. Concretely, for a given detector (e.g., PCA), the prompt instructs the LLM to:
> > (i) analyze the inherent characteristics and limitations of that detector,
> > (ii) identify anomaly patterns that the detector tends to miss, and
> > (iii) implement a Python function detector\_specific\_binary\_classifiers(model, X\_train) that constructs a binary classifier whose decision logic is explicitly tailored to these detector-specific behaviors, while interacting with the detector only through its score interface (e.g., model.predict\_score()).
> > The resulting classifier exposes standard fit() and predict\_score() methods and is trained on the given training data (the prompt is provided in Appendix 6.17).
> >
> > **(2) Experimental findings.**
> > We denote this variant as LLM-DAS (Classifier) and compare it against the base detector and our original LLM-DAS (Ours) on multiple datasets. The AUC-PR results are summarized as follows. We observe that:
> > (1) LLM-DAS (Classifier) consistently improves over the base detector on average (from 0.536 to 0.553), confirming that detector-specific classifier design via LLM reasoning is indeed beneficial;
> > however,
> > (2) it remains slightly worse and less stable than our original LLM-DAS (Ours).
> >
> > These results suggest that tightly encoding detector-specific logic directly into the classifier architecture and code can increase reasoning and implementation complexity for the LLM, and may reduce the flexibility of the learned decision boundary. In contrast, our original design—where the LLM focuses on generating detector-aware hard anomalies and a standard classifier learns from these samples—appears to strike a better balance between detector-specific adaptation and robust generalization.
> >
> > **(3) Takeaways and future directions.**
> > In summary, our experiments show that the reviewer’s idea is promising: LLM-designed detector-specific classifiers can improve over the base detector, but the current instantiation can be further improved with more refined and detector-specific design choices. In the future work, we plan to more systematically explore how the LLM can be guided to directly synthesize high-quality detector-specific classifiers.
> >
> > | Datasets       | PCA   | +LLM-DAS (Ours) | +LLM-DAS (Classifier) |
> > |-----------|-------|-----------------|-----------------------|
> > | abalone   | 0.839 | 0.841           | 0.840                 |
> > | imgseg    | 0.772 | 0.824           | 0.807                 |
> > | lympho    | 1.000 | 0.989           | 0.864                 |
> > | pendigits | 0.386 | 0.485           | 0.455                 |
> > | pima      | 0.701 | 0.751           | 0.712                 |
> > | shuttle   | 0.963 | 0.991           | 0.992                 |
> > | vertebral | 0.138 | 0.143           | 0.149                 |
> > | vowels    | 0.105 | 0.109           | 0.118                 |
> > | wilt      | 0.064 | 0.097           | 0.187                 |
> > | WPBC      | 0.394 | 0.408           | 0.401                 |
> > | Average   | 0.536 | 0.564           | 0.553                 |

---

> > ### Author Response · Authors · 2025-11-27
> > **Response Summary and Opportunity for Further Discussion**
> >
> > Dear Reviewer N2i1,
> >
> > Thank you once again for your time and thoughtful feedback. To respectfully follow up, we would like to briefly summarize the concrete steps we have taken to address each of your comments:
> >
> > **For W1:** Ensuring Validity and Reliability of LLM-Generated Code. We have discussed our verification mechanisms, including controlled programming interfaces, structured output requirements, and empirical validation of anomaly hardness and boundary refinement.
> >
> > **For W2:** Robustness to Binary Classifier Choice. We expanded experiments in Fig. 9 (Appendix 6.12) to show consistent performance gains across multiple classifiers, confirming that improvements stem from the synthesized anomalies, not the classifier itself.
> >
> > **For W3:** Synthetic Anomaly Ratio and Class Imbalance. We conducted a comprehensive ratio analysis (0.1–1.0) across all datasets (Table 10–11, Appendix 6.12), demonstrating that LLM-DAS is robust to ratio changes and that a small fraction (e.g., 10%) of challenging anomalies is sufficient for effective refinement.
> >
> > **For Q1:** Standalone Binary Classifier Performance. We included results in Fig. 10 (Appendix 6.12) showing that while the classifier alone improves over the base detector, the full ensemble (classifier + base detector) achieves the best performance, leveraging both refined boundaries and original inductive biases.
> >
> > **For Q2:** LLM-Designed Detector-Specific Classifiers. Motivated by your suggestion, we experimented with a variant where the LLM designs a custom classifier (prompt is in Appendix 6.17). The performance highlights that it is a promising direction for future work.
> >
> > We sincerely appreciate your detailed reviews, which have helped us significantly improve the clarity, rigor, and validation of our work. If you have any further questions or suggestions, we would be truly grateful to hear them.

---

### Official Review · Reviewer_v399 · 2025-10-31

**Soundness:** 3
**Presentation:** 3
**Contribution:** 3
**Rating:** 6
**Confidence:** 2

**Summary:**

This paper introduces LLM-DAS, a framework that leverages Large Language Models to enhance tabular anomaly detection methods by generating detector-specific synthetic anomalies. Rather than using LLMs to directly process data, the approach positions LLMs as "algorithmists" that analyze detector mechanisms and generate reusable Python code for synthesizing "hard-to-detect" anomalies. These synthetic anomalies are used to augment training data, transforming the one-class classification problem into a more discriminative two-class setting. The method is evaluated on 36 benchmark datasets with five different base detectors.

**Strengths:**

1 - The repositioning of LLMs from data processors to algorithm strategists is novel and well-motivated, effectively addressing both privacy concerns and the inherent difficulties LLMs face with heterogeneous tabular data while leveraging their strengths in code generation and algorithmic reasoning.

2 - The framework demonstrates strong empirical results across 36 diverse datasets, showing consistent improvements over base detectors and outperforming recent LLM-based approaches like AnoLLM while being more computationally efficient.

3 - The approach is practical and scalable - the detector-specific synthesis code is generated only once per detector type and can be reused across multiple datasets, making it cost-effective compared to approaches requiring LLM fine-tuning or repeated querying.

**Weaknesses:**

1 - The paper lacks theoretical analysis or formal guarantees about why the generated "hard" anomalies should improve detector performance, relying primarily on empirical validation without providing theoretical bounds or convergence properties of the enhanced detectors.

2 - The synthesis strategies appear relatively simplistic (e.g., extrapolation along vectors for IForest), and it's unclear whether the LLM genuinely discovers novel algorithmic insights or simply implements predetermined heuristics that could have been hand-coded without LLM involvement.

3 - The evaluation is limited to classical anomaly detectors and doesn't explore integration with modern deep learning-based methods, nor does it investigate failure modes where the synthetic anomalies might degrade performance or introduce unintended biases.

4 - The paper doesn't adequately address the potential for adversarial exploitation - if the synthesis code is reusable and potentially accessible, malicious actors could use knowledge of the augmentation strategy to craft anomalies that evade the enhanced detectors.

5 - The ablation studies, while comprehensive in number, don't sufficiently isolate the contribution of LLM reasoning versus simpler rule-based approaches - a comparison with expert-designed synthesis strategies for each detector would better validate the value of LLM involvement.

**Questions:**

Can you provide examples (such as performing a casae study) where the LLM-generated code discovers non-obvious or counterintuitive synthesis strategies that human experts might not have designed?

How sensitive is the approach to the choice of LLM and prompt engineering - would the method still work with smaller, open-source models?

---

> ### Author Response · Authors · 2025-11-20
> **Response (Part 1/4)**
>
> **R to W1:**
> We thank the reviewer for raising the question of theoretical understanding.
> While our main contribution is empirical, we agree that providing conceptual intuition
> is valuable. We clarify the mechanism at a high level and include a formal proposition
> with proof in the Appendix 6.14.
>
> **(A) Easy–Hard anomaly decomposition.**
> We distinguish between two types of anomalies:
> (i) easy anomalies, which lie far from the decision boundary of the base detector
> $f_t$ and are already correctly detected, and
> (ii) hard anomalies, which lie close to the boundary and account for most errors.
> LLM-DAS is designed to generate detector-aware hard synthetic anomalies by
> transforming borderline normal samples. These hard synthetic samples provide precisely
> the near-boundary supervision that $f_t$ lacks. Thus, the enhancement classifier
> $\tilde f_t$ improves detection specifically in the region where $f_t$ struggles, while $f_t$
> continues to handle easy anomalies well. The ensemble combines these complementary
> strengths.
>
> **(B) A formal proposition added to the Appendix.**
> To formalize this intuition, we introduce in the Appendix 6.14 a simple proposition that shows:
> if synthetic hard anomalies geometrically cover true hard anomalies within a small
> neighborhood, then any margin learned on the synthetic anomalies transfers to the true
> hard anomalies under mild regularity. This illustrates how detector-aware synthetic
> hard samples can tighten the decision boundary around the difficult region.
> The full statement and proof are provided in the Appendix 6.14.
>
> **(C) Convergence and stability of the enhancement procedure.**
> The final fused detector is $F_t(x)
> = \mathrm{Norm}(f_t(x)) + \mathrm{Norm}(\tilde f_t(x))$.
> Both terms are normalized, so the fusion preserves the stability properties
> of the base detector $f_t$ while adding a corrective term near the boundary.
> In this sense, the enhancement step acts as a stable additive refinement that reduces
> bias around hard cases while introducing only controlled variance.
>
> **R to W2, W5, and Q1: (Part 1/2)**
>
> We thank the reviewers for raising these critical points regarding the depth of LLM reasoning and the necessity of its involvement. We address them jointly by first demonstrating the *non-obvious nature* of the LLM-generated strategies (directly answering Q1 and W2), and then presenting empirical evidence that these strategies *outperform both simple and expert-inspired heuristics* (addressing W5 and the remainder of W2).
>
> **(A) The LLM Identifies Non-Obvious, Detector-Specific Weaknesses (Addressing Q1 and the ''Simplistic'' Concern in W2)**
>
> We first emphasize a key clarification of our framework: LLM-DAS does not ask the LLM to generate anomalies directly. Instead, it asks the LLM to analyze the algorithmic weaknesses of a target detector and write a programmatic synthesis policy that exploits those weaknesses.
> This distinction—LLM as an algorithm analyst, not a data-level generator—is crucial for understanding why the resulting strategies are non-trivial and detector-specific.
> In practice, the LLM synthesizes code that exploits subtle, detector-specific failure modes in ways that are not standard practice. We highlight two compelling examples from Appendix 6.20:
>
> *   **For OCSVM (Constellation Centroid Synthesis):** The LLM identified that the kernel-induced decision boundary can form "pockets"—regions enclosed by support vectors but empty of training data. Without access to internal model parameters, the LLM's strategy approximates these pockets by forming "constellations" of high-score borderline normal samples and synthesizing anomalies at their geometric centroids. These centroid points fall inside the pockets, receiving deceptively low anomaly scores. This strategy exploits a nuanced geometric property of OCSVM's decision function that is non-trivial and not a standard data augmentation technique.
>
> *   **For PCA (Manifold Extrapolation):** The LLM reasoned that any point lying near the principal subspace will have low reconstruction error, even if semantically anomalous. Instead of naive perturbation, it generates anomalies by extrapolating along directions connecting extreme normal samples—directions that align with high-variance components. This ensures the synthesized points are far from the normal data cloud yet remain within the principal subspace, making them "adversarial" to PCA. This strategy requires an understanding of PCA's linear manifold assumption beyond simple noise injection.
>
> These examples demonstrate that the LLM performs structured reasoning about the detector's inductive bias, producing targeted strategies that are **not simple, pre-determined heuristics** but rather adaptive, algorithm-aware policies.

---

> > ### Author Response · Authors · 2025-11-20
> > **Response (Part 2/4)**
> >
> > **R to W2, W5, and Q1: (Part 2/2)**
> >
> > **(B) Empirical Validation Against Simpler Baselines and the Infeasibility of Expert-Coding at Scale (Addressing W5 and "Hand-coded" in W2)**
> >
> > We agree that comparing against hand-designed, detector-specific expert rules is an interesting point. Our experimental design and the core motivation of LLM-DAS are built upon a key observation: **manually crafting effective, detector-specific synthesis rules for every algorithm is highly non-trivial, time-consuming, and does not scale.** Our framework aims to *automate* this very process of "expert reasoning".
> >
> > 1.  **LLM-DAS vs. Common Expert Heuristics (The Best Available Proxy):** While we cannot retrospectively design a full suite of expert rules for all detectors, we have compared against the most common and intuitive *rule-based heuristics* that an expert might first consider (Fig. 4 (a)). Methods like Gaussian noise injection, random outlier generation, and SMOTE-based extrapolation represent a class of simple, model-agnostic "expert rules". The results clearly show that these generic strategies are **insufficient and often harmful**, as they fail to respect the detector's intrinsic logic. For instance, random synthesis severely degraded the performance of the sophisticated DRL detector. This demonstrates that effective augmentation requires deep detector awareness, which is non-trivial to hand-code.
> >
> > 2.  **The Cross-Detector Experiment as Proof of Specificity (Why LLM $>$ Static Heuristics):** The cross-detector experiment (Table 2) is our strongest *indirect* evidence against the "simple hand-coding" hypothesis. If the LLM were merely implementing a simplistic, universally applicable heuristic (e.g., "add small noise" or "extrapolate along a vector"), then the synthesis code for one detector (e.g., PCA) should work reasonably well for another (e.g., OCSVM). **The fact that it does not—and that mismatched strategies can cause significant performance degradation—proves that the LLM-generated code is not a generic heuristic, but is instead highly tailored to the specific mechanics of each detector.** This level of detector-specific customization is precisely what a human expert would strive for, and our results show the LLM achieves it programmatically and at scale.
> >
> > 3.  **Ablation Studies Isolate the "Reasoning" Component:** Our ablation studies (Fig. 4 (b)) further deconstruct the LLM's contribution. When we remove detector-awareness (''Generic'' variant) or the ability to generate "hard" anomalies (''Simple'' variant), performance drops significantly. This confirms that the gains are not from a generic code-generation capability, but from the LLM's success in executing the core, expert-like task we set for it: *reasoning about algorithmic weaknesses to formulate a targeted synthesis policy.*
> >
> > In summary, while a perfect comparison against a complete set of human-expert strategies is logistically prohibitive, our suite of experiments provides compelling, multi-faceted evidence:
> > *   LLM-DAS consistently outperforms common, simple heuristics.
> > *   It generates strategies that are detector-specific, not generic.
> > *   Its performance hinges on components that mimic expert reasoning (detector-awareness, hardness targeting).
> >
> > This demonstrates that LLM-DAS offers a scalable way to automate expert-level, detector-specific synthesis design—something that would be difficult and costly to hand-engineer across heterogeneous detector families.

---

> > ### Author Response · Authors · 2025-11-20
> > **Response (Part 3/4)**
> >
> > **R to W3:**
> > We appreciate the reviewer's insightful comments. Below we address both parts of the concern.
> >
> > **(1) Integration with modern deep learning-based anomaly detectors.**
> > Actually, we have already explored integration with the recent SOTA deep learning-based method DRL [1], as illustrated in Table 1 of manuscript. The results show that LLM-DAS continues to provide statistically significant performance improvements on top of DRL. And LLM-DAS coupled with DRL already achieves state-of-the-art performance on several benchmark datasets (as shown in Fig. 3 of manuscript). This demonstrates that
> >
> > - LLM-DAS is not restricted to classical detectors,
> > - its refinement mechanism is compatible with modern deep-learning architectures, and
> > - the improvements persist even when the baseline model is already strong.
> >
> > We plan to expand our evaluation to more deep learning–based detectors in future work.
> >
> > **(2) Analysis of failure modes and potential unintended biases.**
> >
> > We agree that synthetic anomalies could, in principle, introduce biases or degrade performance.
> > To investigate this, we conducted a detailed failure-mode analysis (Table 20, Table 21, Appendix 6.16), where we introduce a controllable hardness-level interface (High/Medium/Low) and quantify hardness using Wasserstein distances. We provide here a high-level overview of our new analysis, and kindly refer the reviewer to Appendix 6.16 for detailed definitions, tables, and additional discussion.
> > Our findings reveal a clear and actionable pattern:
> >
> > - Failures occur primarily in datasets that are inherently easy, where true anomalies lie far from the normal manifold.
> > - In these cases, the default LLM-DAS setting sometimes produces synthetic anomalies that are much harder than necessary, lying too close to the normal manifold and over-complicating the decision boundary.
> >
> > Importantly, the hardness interface offers a direct practical mechanism to mitigate these side effects.
> > When we reduce the hardness level (Medium/Low), performance consistently recovers to the baseline or surpasses it (Table 20).
> >
> >
> > **Conclusion.**
> > LLM-DAS integrates smoothly with deep learning–based anomaly detectors and provides statistically significant gains.
> > Furthermore, our failure-mode investigation identifies when synthetic anomalies may introduce side effects and provides a principled mitigation strategy via controlled hardness-level adjustment—ensuring that LLM-DAS remains robust and safe across diverse scenarios.
> >
> > [1] DRL: Decomposed Representation Learning for Tabular Anomaly Detection. ICLR 2025.

---

> > ### Author Response · Authors · 2025-11-20
> > **Response (Part 4/4)**
> >
> > **R to W4:**
> > We thank the reviewer for raising this important consideration regarding adversarial security. We agree that understanding the potential for misuse is critical. Our framework is fundamentally designed as a defensive augmentation technique to improve detector robustness, and we clarify below why it does not meaningfully increase adversarial risk compared to the status quo.
> > (1) The generated synthesis code is not a reusable or accessible attack vector.
> > The reviewer's concern presupposes that the synthesis code is both reusable and accessible to an attacker. In practice, both conditions fail to hold: (a) Not Reusable: The programs generated by our LLM are detector-specific and instantiation-private. They are tightly coupled to the internal state (e.g., decision function, training statistics) of a specific, already-trained detector model  $f_t$. Without direct access to this defender-owned model and its training data $D_{train}$ , the code is a template that cannot produce effective anomalies.
> > (b) Not Accessible: The synthesis occurs during the defender's private, one-time training phase. The augmented dataset is kept private during the defender's training process. Only the final model's performance is published, not the synthetic data itself, preventing any direct exploitation by malicious actors. Consequently, the generated program is a private, one-time augmentation function, not a public, inference-time attack recipe. It offers no practical leverage to a malicious actor.
> >
> > **R to Q2:**
> > Thanks for your valuable suggestion! We would like to respond to the sensitivity of LLM-DAS to the choice of LLMs and prompt engineering one by one.
> > For the choice of LLMs, we have already provided the results in Table 8 of Appendix 6.9. The results show that LLM-DAS with GPT-4o achieves performance comparable to that with Gemini-2.5-pro. To further strengthen this point, we have conducted new experiments with an even more accessible Qwen3 (open-source). The results are summarized in Table 8 of Appendix 6.9. Though the average performance of LLM-DAS on Qwen3 slightly drops compared to Gemini-2.5-pro and GPT-4o, it still surpasses the base detector by a large margin. The results clearly show that LLM-DAS maintains its strong performance across all three LLMs. The reasons are as follows: our framework positions LLM as an ''algorithmic strategist.'' LLM takes as input a high-level description of the algorithm (such as how OCSVM or Isolation Forest works), rather than complex, heterogeneous raw tabular data. LLM is required to output structured Python code. This explicit, narrow output format and task scope (generating code designed to break specific hypotheses) significantly reduces code complexity and the risk of generalization errors, improving its reliability. These operations greatly simplify the inference burden of LLM, allowing it to focus on abstract logical vulnerability analysis, enabling a range of capable models to achieve this.
> >
> > For the sensitivity of LLM-DAS's performance to the prompt engineering, we have already provided the results in Fig. 4(b) with detailed results in Appendix 6.13.2. In the experiments, we consider the following variants: First, to test our core detector-aware premise, a Generic variant was created by removing the detector's principles from the prompt. Second, to confirm the necessity of generating hard anomalies, a Simple variant was prompted without access to the model.predict\_score() function, preventing the LLM from assessing anomaly difficulty. Finally, to evaluate our borderline heuristic, a Random variant was instructed to transform randomly selected normal samples rather than borderline normal samples into anomalies. The inferior performance of each variant confirms that detector-awareness, the capability to generate challenging anomalies, and a focus on borderline samples are all critical to the success of LLM-DAS.

---

> ### Comment · Reviewer_v399 · 2025-11-21
>
> I thank the authors for their feedback including the case studies - some follow up questions about the mentioned weaknesses and questions that concern me the most below.
>
> 1 - Regarding the question why the generated "hard" anomalies should improve detector performance, I acknowledge the provided conceptual intuition is reasonable to some extend, while I am still hesitated since whether the generated hard anomalies can help with the performance or how much they can actually improve the performance seem to be determined by the shape of the decision boundary. It seems to be easy to think about boundaries that may not be tightened by the generated hard anomalies (e.g., those complicated ones with the generated hard anomalies only covering a small proportion of it). Can you help explain this to avoid making the justification purely empirical?
>
> 2 - It remains confusing that whether the several ablations (Generic/Simple/Random) truly isolate the contribution of LLM reasoning. There has been studies revealing that LLMs can involve some widely acknowledged rules or designing principles based on its built-in reasoning capabilities without explicitly pointing it out in the prompt. This also closely relates to the second question. This part thus remains vague to me and I would appreciate further clarifications.
>
> 3 - I suggest the authors to enhance the connection between appendix and the main content since the appendix is really long while most pointers in the mian content are not with enough guidance.

---

> > ### Author Response · Authors · 2025-11-21
> > **Response to additional questions (Part 1/2)**
> >
> > We sincerely thank the reviewer for the engaging follow-up questions, which allow us to clarify the core mechanisms of LLM-DAS with greater depth and precision.
> >
> > **R to additional question 1: Further Justification Beyond Empiricism:** We agree with the reviewer that a complex, global decision boundary may not be uniformly "tightened." The effectiveness of LLM-DAS does not rely on such global coverage but on a targeted, local refinement mechanism. Our conceptual justification is twofold and is strongly supported by our anomaly-type analysis.
> >
> > **(1) Local Patching of Algorithmic Weak Spots:** The LLM in LLM-DAS acts as an automatic analyst that identifies a detector's structural weaknesses from its algorithm description. It then generates a program to create anomalies that specifically target these weaknesses. The goal is not to resample the entire boundary but to patch critical, high-error regions where the base detector $f_t$ is known to fail. The performance improvement stems from precisely this targeted effort.
> >
> > **(2) Empirical Evidence via Anomaly-Type Alignment (Connecting to Appendix 6.15):** Our experiments in Appendix 6.15 (Table 19, Fig. 11-13) provide direct, empirical validation of this local-patching principle. They demonstrate that LLM-DAS automatically synthesizes anomalies whose distribution is closest to the specific anomaly types (e.g., dependency and local anomalies) that each base detector (e.g., IForest, OCSVM) struggles with most, as per established benchmarks like ADBench.
> >
> > - This is crucial: Different anomaly types challenge different segments of the decision boundary. By generating anomalies that align with a detector's known algorithmic blind spots, LLM-DAS proves it can automatically locate and target the most impactful "weak regions" of the boundary, even if they constitute only a portion of the whole.
> >
> > - The significant performance gains we observe on these specific anomaly types confirm that patching these localized regions is both necessary and sufficient for overall improvement.
> >
> > **(3) Preservation of Existing Strengths via Ensembling:** Furthermore, our framework is designed to be robust and non-destructive. As noted in our analysis, for anomaly types where the base detector already performs well (e.g., IForest on Cluster anomalies in Table 19), LLM-DAS maintains stable or slightly improved results. This indicates that our method enhances robustness without overfitting to the particular anomaly categories it uses for augmentation. This is a direct benefit of our ensemble design $F(x) = \text{Norm}(f_t(x)) + \text{Norm}(\tilde{f}_t(x))$. The original detector $f_t$ continues to dominate in regions it already models accurately, preserving its discriminative ability for "easy" anomalies, while the enhancement classifier $\tilde{f}_t$ extends the coverage toward harder cases targeted by the LLM.
> >
> > Therefore, while a single set of synthetic anomalies may not cover the entire boundary, its coverage is highly strategic. It focuses on the regions that matter most for reducing the base detector's error, which is why the method delivers consistent and significant gains beyond purely empirical observations.

---

> > > ### Author Response · Authors · 2025-11-21
> > > **Response to additional questions (Part 2/2)**
> > >
> > > **R to Question 2: On Isolating the Contribution of LLM Reasoning**
> > >
> > > We appreciate this opportunity to clarify what our ablations truly demonstrate. The reviewer correctly notes that LLMs may utilize internalized knowledge. Our claim is not that the LLM invents entirely new mathematics, but that it performs a contextual, detector-specific reasoning and synthesis task that goes far beyond retrieving a single, static rule.
> > >
> > > **(1) The Ablations Test Specific, Necessary Components:** Our ablation studies (Generic/Simple/Random in Fig. 4(b)) are designed to remove pecific, instructed capabilities from the LLM:
> > >
> > > - ''Generic'': Removes detector-awareness. The LLM cannot perform reasoning conditioned on the target detector's algorithm.
> > > - ''Simple'': Removes the capability to generate hard anomalies by denying access to the score function.
> > > - ''andom'': Removes the borderline heuristic, a key strategic element.
> > >
> > > The significant performance drop in each variant confirms that each of these instructed reasoning components is critical. This shows that the LLM is not just executing a single built-in rule but is dynamically combining these elements based on the prompt.
> > >
> > > **(2) The Cross-Detector Experiment is the Key Evidence (Connecting to Table 2):** The most compelling evidence against a "universal rule" hypothesis comes from our cross-detector experiment (Table 2).
> > >
> > > - If the LLM were simply applying a generic, detector-agnostic heuristic (e.g., ''add small noise'' or ''extrapolate''), then the synthesis policy  $S_{\text{program}}^A$  for Detector A should work reasonably well when transferred to Detector B.
> > > - The results clearly show the opposite: Transferred strategies often perform poorly. This demonstrates that each LLM-generated policy is highly specialized and its effectiveness is contingent on a precise match between the synthesis logic and the target detector's internal mechanics. This specialization is the hallmark of genuine, detector-specific reasoning.
> > >
> > > In summary, the ablations prove the necessity of key reasoning components, and the cross-detector experiment proves the specificity of the resulting strategies. Together, they robustly isolate and validate the contribution of the LLM's contextual reasoning capabilities.
> > >
> > >
> > > **R to Question 3: On Enhancing the Connection Between Main Content and Appendix**
> > >
> > > We sincerely thank the reviewer for this invaluable suggestion to improve the readability of our work. We have taken the following concrete steps to create a clearer navigational structure:
> > >
> > > **(1) Enhanced In-Text Guidance:** We have revised the main text by adding explicit pointers at the end of key sections (e.g., Methodology and Experiments) to guide readers to the relevant appendices for extended analyses and results. For instance, we now explicitly direct readers interested in theoretical formulation,  and failure-mode analysis to Appendices 6.14 and 6.16, respectively.
> > >
> > > **(2) Added a Descriptive Appendix Table of Contents:** To address the challenge of the appendix's length, we have now included a Table of Contents at the beginning of the Appendices. This  provides descriptive titles for each appendix (e.g., ''Hardness-Controlled Synthetic Anomalies and Side-Effects Analysis'' instead of just ''Appendix 6.16''), allowing readers to quickly locate the specific extended discussions they are interested in without having to browse through the entire document.
> > >
> > > We believe these modifications significantly enhance the reader's experience by seamlessly connecting the main narrative with the rich supporting evidence in the appendices.

---

> ### Comment · Reviewer_v399 · 2025-11-25
>
> I thank the authors for their detailed feedback — this clarifies the concerns that interest me the most. I suggest the authors to involve the discussion here into the appendix of this paper in its next version, and I am thus raising my score accordingly.

---

> ### Author Response · Authors · 2025-11-25
>
> Thank you for kindly increasing the score from 6 to 8 and for taking the time to review our response! Following your suggestion, we have added this discussion to Appendix (6.14.5 and 6.21).

---

### Official Review · Reviewer_mvV9 · 2025-11-01

**Soundness:** 3
**Presentation:** 4
**Contribution:** 3
**Rating:** 6
**Confidence:** 4

**Summary:**

This paper proposes LLM-DAS, a novel framework to enhance tabular anomaly detectors (TAD). It addresses the "fragile assumptions" of existing detectors and the privacy/usability issues of applying LLMs to tables. The core idea is to reposition the LLM from a "data processor" to an "algorithmist". In a data-agnostic phase, the LLM reasons about a detector's high-level logic to identify weaknesses and generates reusable Python code to synthesize "hard-to-detect" anomalies that exploit these flaws. In a data-specific phase, this code is executed on a dataset to augment the normal-only training data . This converts the one-class problem into a more robust two-class classification task. Extensive experiments on 36 benchmarks show significant performance boosts for mainstream detectors.

**Strengths:**

1. The central idea of using the LLM for abstract algorithmic reasoning and code generation—rather than data processing—is highly innovative. It leverages the LLM's core strengths and opens a new path for applying LLMs to non-linguistic domains.
2. The data-agnostic code generation phase fundamentally solves the data privacy issue. The resulting code is reusable, making the approach highly scalable: the LLM is queried only once per detector type, not once per dataset.
3. The framework's effectiveness is validated across 36 datasets and 5 diverse detectors. Crucially, ablation studies (Fig. 4b) and cross-detector experiments (Table 2) confirm that the "detector-aware" strategy is the key to success, significantly outperforming generic synthesis methods .

**Weaknesses:**

1. The framework's success is entirely contingent on the LLM's ability to correctly understand an algorithm's description and identify its true weaknesses . A superficial or incorrect logical analysis by the LLM would lead to a suboptimal synthesis policy.

2. The enhanced classifier is trained to detect anomalies from the LLM's specific synthesis strategy. It is not guaranteed to generalize to all types of "hard" anomalies, but may simply learn to spot the artifacts of those specific generated samples.


3. The final detector is an ensemble (a sum) of the original detector and a new classifier (e.g., RandomForest). This adds complexity and inference cost rather than directly "patching" or retraining the original model itself.

**Questions:**

1. For trainable detectors (like DRL or Autoencoders), did you consider using the augmented data to fine-tune the original detector itself, rather than adding an external binary classifier?
2. The framework seems to generate one primary synthesis policy per detector. What if a detector has multiple, distinct logical weaknesses? Could the LLM be prompted to find several weaknesses, and would a mixture of synthetic anomalies targeting all of them provide even more robustness?

---

> ### Author Response · Authors · 2025-11-20
> **Response (Part 1/3)**
>
> R to W1: We thank the reviewer for this insightful observation. Indeed, LLM-DAS relies on the LLM’s
> ability to interpret a detector’s algorithmic description. However, the framework is
> not brittle and is intentionally designed to remain robust even when the LLM’s
> analysis is imperfect.
>
> **(1) The LLM analyzes a clean, parameter-free, logical description—not raw tabular data.**
> The LLM’s task is restricted to summarizing the detector’s high-level mechanism
> and reasoning about its structural weaknesses. These descriptions (e.g., PCA steps) are symbolic and algorithmic rather than data-dependent,
> making them significantly easier and more reliable for LLMs to understand.
>
> **(2) The prompt imposes strict structural and functional constraints.**
> As detailed in Eq. 3 of Section 3.2, the LLM is required to produce:
> (i) a detector-aware synthesis policy,
> (ii) an executable, data-agnostic program,
> (iii) an explanation of its reasoning.
> These constraints substantially reduce the risk of superficial reasoning by enforcing
> explicit, verifiable logic.
> For example, in Case Study of Section 3.2.2, the LLM’s policy for IForest describes two core steps:
> (i) selecting borderline normal samples using the detector’s own score function, and
> (ii) performing controlled extrapolation that preserves long path lengths.  The accompanying explanation clarifies why these operations align with the algorithmic weakness of IForest (i.e., difficulty isolating deep points), making the
> LLM’s intentions transparent and allowing us to verify that the generated code faithfully implements this logic.  This level of specificity prevents arbitrary code generation.
>
> **(3) The final detector does not rely solely on the LLM-generated component.**
> After generating synthetic anomalies, we train an enhancement classifier $\tilde{f}_t$ and
> combine it with the original detector $f_t$ using a normalized score-level ensemble.
> This fusion (Eq. 7 of Section 3.2.2) ensures that even if the LLM’s synthesis is occasionally
> suboptimal, the original detector’s behavior remains preserved, preventing degradation.
>
> **(4) Extensive empirical evidence demonstrates robustness across detectors, datasets, and LLMs.**
> The results in Table 1 of manuscript indicate that the improvements over the original source detectors across 36 benchmark datasets are statistically significant.
> The detailed results in Table 4 and Table 5 of Appendix 6.7 also show that the suboptimal results are occasional.
> In addition, our experiments in Table 8 of Appendix 6.9 further verify that LLM-DAS remains effective across different LLMs, including GPT-4o, Gemini-2.5-Pro, and Qwen3.
>
> We have clarified these points in the Appendix 6.19.

---

> ### Author Response · Authors · 2025-11-20
> **Response (Part 2/3)**
>
> R to W2 \& Q2:
>
> We thank the reviewer for the thoughtful questions regarding
> (1) whether the enhanced classifier truly generalizes beyond the specific LLM-generated anomalies,
> and (2) how our framework behaves when a detector is challenged by several different anomaly patterns.
> We kindly refer the reviewer to Appendix 6.15 for detailed analyses on anomaly synthesis across different anomaly types.
>
> **(1) Generalization beyond a specific synthesis strategy (W2).**
> We agree that a single synthesis strategy cannot cover all possible forms of ''hard'' anomalies.  However, LLM-DAS does not generate arbitrary artifacts tied to a particular code snippet;  instead, its synthesis is detector-aware and explicitly driven by the detector’s scoring behavior.  As shown in our anomaly-type analysis (Appendix 6.15), different detectors exhibit distinct sensitivity patterns: for example, IForest and OCSVM show relatively poor performance on dependency and local anomalies.
> LLM-DAS is able to synthesize anomalies whose distributions are closest to the anomaly types that are most challenging for each detector (Table 19 of Appendix 6.15), and these synthesized anomalies lead to the largest performance gains and the most pronounced score-separation improvements on those challenging types (Table 19, Fig. 11-13 of Appendix 6.15).
> This indicates that the enhanced detector is not merely memorizing synthetic artifacts; it is refining its decision boundary in precisely those regions where the base detector struggles.
>
> At the same time, for anomaly types where the base detector already performs well, LLM-DAS maintains stable or improved performance (Table 19).  This shows that the framework enhances robustness without overfitting to a specific anomaly category.  Moreover, because we use a score-level ensemble between the original detector $f_t$ and the enhancement classifier $\tilde{f}_t$, the original detector continues to dominate in regions it already models accurately, preserving its discriminative ability on easier anomalies while extending coverage toward harder cases.
>
> **(2) Detectors that are challenged by multiple anomaly types.**
> Some detectors are not only weak on a single anomaly type, but perform poorly on several related types.  For instance, ADBench [1] and our experiments show that IForest and OCSVM have difficulty with both dependency and local anomalies.
> In Appendix 6.15, we observe that for such detectors, the anomalies synthesized by LLM-DAS are not restricted to a single category: they exhibit mixed structural characteristics and are distributionally close to both dependency and local anomalies (Table 19), and the performance gains are substantial on both of these challenging types (Table 19, Fig. 11-13).
> This suggests that even though we ask the LLM to produce one synthesis policy per detector, the resulting anomalies can naturally cover multiple challenging anomaly patterns, rather than being confined to a single, narrowly defined anomaly type.
>
> **(3) Explicitly prompting for several challenging patterns (Q2).**
> Motivated by the reviewer’s suggestion, we further experimented with a variant where the prompt explicitly requests the LLM to consider several challenging behaviors of a detector and generate a unified synthesis program that targets multiple kinds of hard patterns (prompt is provided in Appendix 6.18).  We denote this variant as LLM-DAS (M-W) (Multiple Weakness), and the results are provided in Table 22 of Appendix 6.18.  The results show that this variant still consistently improves over the base detector, and it surpasses the original LLM-DAS in some cases, but on average it is slightly inferior and less stable than our original LLM-DAS. The experimental results indicate that the reviewer’s intuition is indeed promising: LLMs are capable of reasoning about multiple detector weaknesses and generating synthesis code that improves performance across several challenging patterns. However, directly requiring the LLM to simultaneously account for several heterogeneous behaviors
> within a single synthesis program may increase both the reasoning burden and the code-generation complexity. This tends to blur the detector-specific logic that makes LLM-DAS effective, leading to slightly reduced stability compared to our original design.
>
> **(4) Future directions.**
> We greatly appreciate the reviewer’s suggestion and believe it opens a fruitful direction for future work. Motivated by the reviewer’s valuable suggestion, we plan to develop a more refined multi-policy extension of LLM-DAS in the future work, such as generating a set of disentangled synthesis sub-policies—each tailored to one weakness.  We can also extend this line of analysis by developing a controllable synthesis module that explicitly conditions the LLM on user-specified anomaly types.
>
> [1] ADBench: Anomaly Detection Benchmark. NeurIPS 2022.

---

> ### Author Response · Authors · 2025-11-20
> **Response (Part 3/3)**
>
> R to W3 \& Q1.
> Thank you for the insightful questions regarding the using of external binary classifier. We address your concern from the following perspectives:
>
> **(1) Why not fine-tune trainable detectors (Q1)?**
>
> We agree that, in principle, one could attempt to fine-tune trainable detectors using  synthetic anomalies. However, we adopt using an external binary classifier for the following reasons:
>
> - ***(a) Trainable detectors (like DRL or Autoencoders) are not parameterized for modeling anomalies.***
> Our work follows the widely adopted one-class tabular anomaly detection (TAD) paradigm,  where only normal samples are available during training and labeled anomalies are absent.  Methods such as DRL or Autoencoders are trained with a
> one-class objective focused solely on reconstructing or compactly encoding the normal
> distribution. Their architectures and losses are not designed to incorporate or discriminate anomaly samples. Fine-tuning them with synthetic anomalies risks breaking their original training objective and destabilizing the learned normal manifold.
>
> - ***(b) A binary enhancement layer provides universal compatibility.***
> Different trainable detectors use incompatible objectives and architectures; updating
> each of them would require detector-specific fine-tuning procedures. The proposed
> binary classifier offers a unified, detector-agnostic refinement layer that can exploit the
> synthesized hard samples without modifying the underlying model. This ensures that
> LLM-DAS remains broadly applicable across different detector families.
>
> - ***(c) Stability and empirical support.*** In addition, the external binary enhancement classifier guarantees a safety
> property: this fusion of binary classifier and original detector (Eq. 7 of Section 3.2.2) ensures that even if the LLM’s synthesis is occasionally suboptimal, the original detector’s behavior remains preserved, preventing degradation. Across 36 datasets, the simple enhancement layer combined with the original detector consistently delivers significant improvements (Table 1). This suggests that the proposed combination is both effective and reliable without requiring  task-specific retraining of the underlying detector.
>
>
> **(2) Inference cost of the ensemble (W3).**
>
> We agree that LLM-DAS introduces a lightweight
> enhancement classifier on top of the original detector. However, the additional
> computational cost is small in practice. As already reported in Appendix 6.10 (Table 9) of
> the original manuscript, the inference-time overhead contributed by the enhancement
> classifier (e.g., a lightweight RandomForest) is small compared to
> the source detector.
> Therefore, the overall inference cost of LLM-DAS remains effective in practice.
>
> **(3) A promising direction for future work.** We appreciate the reviewer's suggestion and agree that exploring how to incorporate
> LLM-DAS into the training dynamics of the original detector—without introducing the
> external binary classifier—is a valuable future direction. This could involve designing
> objectives that safely integrate hard synthetic anomalies while preserving the
> one-class structure, or developing detector-specific ways to "patch" the original model
> directly.

---

> ### Author Response · Authors · 2025-11-27
> **Response Summary and Opportunity for Further Discussion**
>
> Dear Reviewer mvV9,
>
> Thank you once again for your time and thoughtful feedback. To respectfully follow up, we would like to briefly summarize the concrete steps we have taken to address each of your comments:
>
> **For W1:** LLM’s Understanding of Algorithmic Weaknesses. We clarified in Appendix 6.19 that LLM-DAS is designed to be robust even with imperfect LLM reasoning, through structured prompts, executable code constraints, and score-level ensembling that preserves the original detector’s performance.
>
> **For W2:** Generalization Beyond Synthetic Artifacts. We added extensive analysis in Appendix 6.15 showing that LLM-DAS synthesizes anomalies aligned with a detector’s challenging anomaly types, leading to improved performance on real hard cases without overfitting.
>
> **For Q2:** Handling Multiple Detector Weaknesses. We introduced a ''Multiple Weakness'' variant (LLM-DAS (M-W)) in Appendix 6.18. While it shows promise, it is slightly less stable than our original design, suggesting future work on multi-policy or disentangled synthesis.
>
> **For W3 & Q1:** Use of External Classifier vs. Fine-Tuning. We justified the use of an external classifier due to the one-class training paradigm of many detectors, which are not designed to incorporate anomalies. The ensemble ensures stability, universal compatibility, and minimal inference cost (Table 9 of Appendix 6.10).
>
> We sincerely appreciate your detailed reviews, which have helped us significantly improve the clarity, rigor, and validation of our work. If you have any further questions or suggestions, we would be truly grateful to hear them.

---

### Author Response · Authors · 2025-11-20
**Global response**

We would like to sincerely thank all the reviewers and area chairs for the dedicated efforts invested in reviewing our paper. Their insightful comments have been instrumental in helping us significantly improve the rigor, clarity, and overall quality of our paper. We have thoroughly revised the manuscript and appendices to address all concerns, with key changes highlighted in blue.

**Summary of Revisions \& Additions:**

1.  **Anomaly-Type Analysis:** Expanded **Appendix 6.15** to demonstrate how LLM-DAS inherently targets detector-specific weaknesses across diverse anomaly types.
2.  **Hardness Control \& Failure Analysis:** Introduced a controllable hardness-level prompting interface and a comprehensive failure-mode analysis in **Appendix 6.16**, quantifying anomaly difficulty via Wasserstein distance and the Relative Hardness Ratio, providing a principled mitigation strategy via controlled hardness-level adjustment.
3.  **Theoretical Foundation:** Added a formal proposition with proof in **Appendix 6.14** to provide conceptual and theoretical intuition.
4.  **Extended Experiments \& Ablations:**
- Conducted a full-scale analysis on synthetic anomaly ratios (0.1 to 1.0) across all datasets (**Appendix 6.12**).
- Added experiments with an open-source LLM (Qwen3) to verify broad applicability (**Table 8, Appendix 6.9**).
- Explored reviewer-suggested variants (multi-weakness synthesis and LLM-designed classifiers) in **Appendices 6.18 and 6.17**.
5.  **Validation \& Robustness:** Added a new subsection in **Appendix 6.19** detailing mechanisms that ensure the validity and reliability of LLM-generated code.

We believe these revisions have significantly strengthened the paper and thank the reviewers for their invaluable contributions. Itemized responses (denoted as ''R'') to each reviewer's comments (weakness/question is denoted as ''W'' / ''Q'') are appended below.

---

### Comment · Area_Chair_vopw · 2025-11-26

Dear reviewers,

The authors have provided updates and clarifications during the rebuttal, and they are awaiting your follow-up comments.

Please take a moment to review the updates if you haven't already.

Thank you for your efforts.

---

### Author Response · Authors · 2025-11-28
**Summary of Rebuttal and Revisions for AC**

Dear AC,

We sincerely thank all reviewers and area chairs for their thoughtful and constructive feedback.  Before the rollback of scores:

- Reviewer v399 and JZ9t **confirmed that their concerns were fully resolved** and raised scores from **6 to 8** and **2 to 6**.
- Reviewer mvV9 and N2i1 did not post a follow-up comment and remained at **6** and **4**. We confirm that we have made our best effort to respond to all their raised weaknesses and questions.

Below, we summarize how we addressed the concerns raised by each reviewer.

**Reviewer v399 (score raised from 6 to 8, confidence raised from 2 to 3),**
- **Theory \& boundary intuition:** We added a new formal proposition with proof (Appendix 6.14) that explains why synthetic hard anomalies transfer margin improvements to true hard anomalies and how the decision boundary is locally refined.
- **LLM reasoning vs. simple heuristics:** We added detailed case studies on LLM generated policies (Appendix 6.20), and further clarified ablation and cross-detector experiments (Fig. 4, Table 2), showing the LLM performs a contextual, detector-specific reasoning rather than retrieving heuristics, static rules.
- **Integration with deep detectors \& failure modes:**
We highlighted compatibility with the SOTA deep method and added an extensive failure-mode and hardness-level analysis (Appendix 6.16). We also provide a direct practical mechanism to mitigate these side effects.
- **Adversarial risk:** We clarified that synthesis programs are detector-specific, private, and not reusable as public attack recipes.
- **LLM / prompt robustness:** We provided new experiments showing robustness across various LLMs (Table 8 of Appendix 6.9) and clarified prompt-sensitivity analyses (Fig. 4).
- **Better navigation:** We included a Table of Contents to improve the connection between main text and appendix.

**Reviewer JZ9t (score raised from 2 to 6),**
- **Diversity of anomaly types:** We expanded Appendix 6.15 to show that LLM-DAS inherently targets detector-specific weaknesses across different anomaly types, without assuming a fixed taxonomy.
- **Definition and validity of anomalies:** We clarified the constitution of anomalies, and we implemented structured LLM generation with boundary-based synthesis to ensure meaningful and challenging samples.
- **Hardness control and side effects:** We introduced a controllable hardness-level interface and a failure-mode analysis (Appendix 6.16). We also provided a principled mitigation strategy via controlled hardness-level adjustment.
- **Synthetic anomaly ratio:** We conducted a full-scale ablation across ratios 0.1–1.0 (Table 10–11, Appendix 6.12), confirming that LLM-DAS is robust to ratio choice.
- **One-class paradigm justification:** We revised the text to clarify that our work aligns with the widely adopted one-class setting in recent TAD literature.

**Reviewer mvV9 (score remained at 6),**
- **Robustness to imperfect LLLM analysis:** We clarified in Appendix 6.19 that LLM-DAS is designed to be robust even with imperfect LLM reasoning.
- **Generalization beyond synthetic artifacts:** We added analysis (Appendix 6.15) showing that LLM-DAS synthesizes anomalies aligned with a detector's challenging anomaly types, leading to improved performance on real hard cases without overfitting.
- **Handling multiple detector weaknesses:** We added a new multi-weakness synthesis variant in Appendix 6.18, which shows promise.
- **Use of external classifier vs. fine-tuning:** We justified the use of an external classifier due to the one-class training paradigm of many detectors. And the ensemble ensures stability, universal compatibility, and minimal inference cost (Table 9 of Appendix 6.10).

**Reviewer N2i1 (score remained at 4),**
- **Ensuring validity and reliability of LLM-generated code:** We have discussed our verification mechanisms, and added the details to Appendix 6.19.
- **Robustness to binary classifier choice:** We expanded experiments in Fig. 9 (Appendix 6.12) to show consistent performance gains across multiple classifiers.
- **Synthetic anomaly ratio and class imbalance:** We conducted a comprehensive ratio analysis (0.1–1.0) across all datasets (Table 10–11, Appendix 6.12), demonstrating that LLM-DAS is robust to ratio changes.
- **Standalone binary classifier performance:** We included results in Fig. 10 (Appendix 6.12) showing that while the classifier alone improves over the base detector, the full ensemble (classifier + base detector) achieves the best performance.
- **LLM-designed detector-specific classifiers:** Motivated by reviewer's suggestion, we experimented with a variant where the LLM designs a custom classifier (Appendix 6.17), which shows promise.

In addition, we have thoroughly revised the manuscript and appendices, with key changes highlighted in blue. In the revised PDF, the main updates are summarized in the comment named ''Global response'' below.

Best regards,

The authors

---

### Meta-Review · Area_Chair_gofc · 2026-01-04

**Summary:**

Given that most reviewers (except one) find the core idea novel and promising, and the main issues are primarily about clarification and additional analysis (theory/definitions, code validation, robustness to classifier/ratio choices, and broader failure-case discussion), I think this paper is suitable for acceptance as a poster, contingent on the authors addressing these concerns in the revision/rebuttal.

The suggested poster decision was informed by a consistent set of reviewer concerns about completeness and validation rather than the core novelty. Reviewers questioned the lack of theoretical grounding (no formal guarantees or clear definition of what constitutes an anomaly and what “hardness” means), and asked for a clearer procedure to verify the correctness/reliability of LLM-generated synthesis code. They also flagged limited robustness analysis around key design choices (reliance on RandomForest, the fixed 10% pseudo-anomaly ratio, and the impact of class imbalance), and requested stronger ablations on the optimal amount/mix of synthetic anomalies. On evaluation, reviewers noted the focus on classical detectors only, with insufficient exploration of deep methods, failure modes (including cases where synthetic anomalies hurt), and potential bias/side effects from LLM-generated samples. Finally, some raised security/adversarial concerns that reusable synthesis code could be exploited to adapt attacks or evade the enhanced detectors.

**Reviewer Concerns:**

Reviewers raised concerns about limited theoretical grounding and unclear definitions (what constitutes an anomaly and how “hardness” is measured), the validity and reliability of LLM-generated synthesis code, and the robustness of key design choices (binary classifier selection, synthetic-anomaly ratio, and class-imbalance effects). They also questioned whether the LLM adds real detector-specific reasoning beyond simple heuristics, and asked for broader evaluation (including deep methods), clearer analysis of failure modes, side effects/bias, hardness control, and potential adversarial implications. The authors responded by adding a formal proposition with proof, clarifying anomaly/hardness concepts, introducing structured generation and explicit code-verification mechanisms, and expanding experiments across multiple LLMs, prompts, classifiers, and synthesis ratios (0.1–1.0). They also added detailed case studies and stronger ablations to isolate the LLM’s contribution, improved discussion/compatibility with deep detectors, and provided extensive failure-mode and side-effect analyses with practical mitigation strategies.

**Reviewer Scores:**

I think the initial scores (except one) were fair, and the rebuttal largely supports that assessment by directly addressing the key concerns with added theory, stronger validation, and broader robustness analyses. In particular, the additional results and clarifications substantially reduce the main uncertainties, so the remaining low score would reasonably be expected to increase if the reviewer accounts for the rebuttal updates.

---

### Decision · Program_Chairs · 2026-01-26

Accept (Poster)